# Pooled CRISPR interference screening enables genome-scale functional genomics study in bacteria with superior performance

Tianmin Wang[1], Changge Guan[1], Jiahui Guo[1], Bing Liu[2], Yinan Wu[1], Zhen Xie[3,4], Chong Zhang [1,4] & Xin-Hui Xing[1,4]

To fully exploit the microbial genome resources, a high-throughput experimental platform is needed to associate genes with phenotypes at the genome level. We present here a novel method that enables investigation of the cellular consequences of repressing individual transcripts based on the CRISPR interference (CRISPRi) pooled screening in bacteria. We identify rules for guide RNA library design to handle the unique structure of prokaryotic genomes by tiling screening and construct an *E. coli* genome-scale guide RNA library (~60,000 members) accordingly. We show that CRISPRi outperforms transposon sequencing, the benchmark method in the microbial functional genomics field, when similar library sizes are used or gene length is short. This tool is also effective for mapping phenotypes to non-coding RNAs (ncRNAs), as elucidated by a comprehensive tRNA-fitness map constructed here. Our results establish CRISPRi pooled screening as a powerful tool for mapping complex prokaryotic genetic networks in a precise and high-throughput manner.

[1] MOE Key Laboratory for Industrial Biocatalysis, Institute of Biochemical Engineering, Department of Chemical Engineering, Tsinghua University, Beijing 100084, China. [2] Beijing Syngentech Co., Ltd., Beijing 102206, China. [3] MOE Key Laboratory of Bioinformatics and Bioinformatics Division, Center for Synthetic and System Biology, Department of Automation, Tsinghua National Lab for Information Science and Technology, Tsinghua University, Beijing 100084, China. [4] Center for Synthetic and Systems Biology,  Tsinghua University, Beijing 100084, China. These authors contributed equally: Tianmin Wang, Changge Guan.  Correspondence and requests for materials should be addressed to C.Z. (email: chongzhang@tsinghua.edu.cn)

To fully exploit the boosting accumulation of bacterial genomes to provide valuable insights in microbiology and engineering of microbial cells, it is important to develop experimental approaches for gene-phenotype mapping at the genome level. Considering the thousands of genes in diverse microbes, such experimental methods need to be cost-effective and high-throughput, enabling profiling of genome-wide gene sets under multiple conditions in parallel.

Three categories of methods have been established for such purposes. The first makes use of the arrayed collection of single-gene deletion strains, which has been constructed for several model microorganisms, such as *Escherichia coli*[1] and *Bacillus subtilis*[2]. This approach provides a gold standard for understanding the genomics of these microorganisms. However, their applicability to a wide range of species is problematic, because arrayed collections of single-gene knockouts are available for only a handful of species and the manipulation of such libraries requires expensive automation system. An alternative approach depends on recombination and subsequent pooled screening, such as trackable multiplex recombineering (TRMR)[3] or CRISPR-enabled trackable genome engineering (CREATE)[4]. Such strategies also face similar problems when applied to a wider range of microorganisms, as the recombineering system has been established in only a limited number of strains. The most widely applied approach in this field depends on random transposon insertion-derived gene knockout libraries (Tn-seq)[5]. Tn-seq achieves quantitative gene-phenotype mapping by mixing a large number of transposon mutants and monitoring their abundance within a competitive growth culture with next-generation sequencing (NGS). Despite many successful examples[6–8], Tn-seq suffers from the random insertion of transposons into the chromosome, resulting in a bias toward genes with long coding regions and hence poor statistical robustness when assigning functions to short genes, such as genes encoding small but functionally important ncRNAs[9,10]. In addition, Tn-seq is only applicable to genome-wide rather than a more specific library, giving rise to the increased labor and cost when only a subset of genes is of interest.

To address the limitations of established methods in microbial functional genomics, we turned to recently developed CRISPR-Cas9 technology, which can be used for versatile genome editing or expression modulation guided via a programmed single guide RNA (sgRNA) in many organisms[11–13]. This system provides several advantages. First, CRISPR-Cas9 activity has been confirmed in diverse bacterial species[14–17], as well as archaea[18] in only 5 years since its first introduction as a genome editing tool in prokaryotes[13], thereby providing a broadly adoptable platform for theoretically any prokaryote. Second, the target-coding region in sgRNAs consists of only ~20 nucleotides, compatible with massively parallel microarray oligonucleotide synthesis (MOS) and NGS, which simplifies the procedure for constructing either an sgRNA library using MOS or preparing the NGS library using PCR reactions. Finally, compared with Tn-seq, the sgRNA library can be designed uniformly across the bacterial chromosome with minimal bias towards longer genes.

Previously, CRISPR-Cas9-based functional genomics screening in a pooled format has only been demonstrated in mammalian cell lines[19]. To the best of our knowledge, whether a similar approach works as well in prokaryotic organisms has not been determined. In addition, considering the fundamental differences of prokaryotic and eukaryotic genomes[20], it is difficult to directly apply the rules for sgRNA library design and pooled screening learned from previous studies performed in eukaryotic cells to microorganisms. Here we describe a pooled functional genomics study platform in *E. coli* as a proof-of-concept for prokaryotic organisms based on the CRISPR interference (CRISPRi) system, a CRISPR-Cas9 derivative using a nuclease-activity-free Cas9 mutant protein (dCas9) to repress transcription. We first perform a tiling screening where we test the activity of 2,281 sgRNAs targeting 44 genes with known phenotypes. Based on this experiment, we indirectly (at the functional level) learn the activity positioning of sgRNAs in coding regions where CRISPRi maximally changes the expression of endogenous genes, as well as rules for a minimal sgRNA number per gene to maintain reliable hit gene calling. This results in an algorithm to further design a genome-scale sgRNA library targeting each protein-coding or RNA-coding gene in the *E. coli* genome. To make use of this library (~60,000 sgRNAs), we screen for essential, auxotrophic, as well as chemical tolerance-related genes. Notably, we find the CRISPRi functional genomics method to be superior to Tn-seq in terms of essential gene identification, especially when the length of genes is short or the library size is similar. We also elucidate the power of this method to generate hypotheses regarding the functions of ncRNAs. These experiments demonstrate that CRISPRi screening platform represents a transformative tool for defining gene function in a cost-effective and high-throughput manner for potentially any prokaryotic organism.

## Results

**Tiling library screening defines rules for sgRNA design.** The fundamental differences between eukaryotic and prokaryotic genomes, as well as their transcriptional regulation[21], hampers the direct application of the rules drawn from previous eukaryotic library design to prokaryotes. For example, chromatin accessibility and nucleosome occupancy, which are unique structures in eukaryotic genomes[20], have a substantial impact on sgRNA activity[22,23]. Moreover, with current CRISPRi sgRNA library design guidelines in eukaryotic cells, a target site is selected around the transcription start site (TSS)[23]. However, many genes in prokaryotic genomes are organized in operons co-transcribed as polycistronic mRNA, where a common promoter drives the transcription of all genes. In these cases, directing the CRISPRi complex to the promoter region is expected to repress the transcription of the whole operon, rendering the method unable to identify individual genes responsible for the investigated phenotype.

Considering the abovementioned factors, to establish rules for sgRNA design in prokaryotic (*E. coli* here) functional genomics pooled screening, in the first step, we sought to construct an sgRNA library targeting genes for which a knockout produces a known and easily selectable phenotype to perform a tiling screening. To this end, we checked the Keio library[1] for auxotrophic genes in minimal medium. All genes thus identified were cross-checked to verify normal growth in Luria-Bertani (LB) broth[1], and their operon structures were determined. We selected genes transcribed as monocistronic mRNAs with impaired growth in MOPS medium to generate Library I, which consisted of 22 candidates. We also selected a series of genes residing in operons transcribed as polycistronic mRNAs that confer auxotrophy in MOPS medium, as well as all their co-transcribed genes without relevant phenotypes, to generate Library II, which consisted of 22 genes from nine operons with one auxotrophic gene each. Genes in these libraries are listed in Supplementary Table 1. Up to 50 sgRNAs targeting the non-template strand of the open reading frame (ORF) were designed for each gene. We also included 400 negative control sgRNAs without any off-target hit in *E. coli* genome as Library NC. All designed sgRNAs are listed in Supplementary Data 1. We subsequently prepared this library with 2281 members by MOS and incorporated the PCR-amplified library into an optimized sgRNA expression vector that has sustainable gene repression activity (see Methods).

We transformed the sgRNA library by electroporation into *E. coli* strain MCm (a K12 MG1655 derivative with an integrated chloramphenicol-resistance cassette) carrying pdCas9-J23111. We performed screening with MOPS medium as the selective condition and LB broth as the control condition throughout a period of ten cell doublings. The change (selective vs. control conditions) in the relative abundance of each sgRNA (sgRNA fitness) in the final culture was profiled via NGS. Based on the fitness of sgRNAs for each individual gene, the quantitative estimation of genotype-phenotype association (median sgRNA fitness) was calculated and the statistical significance was determined by comparison with negative control sgRNAs (Fig. 1a). An ~80% mapping ratio and good biological replicate agreement (Supplementary Fig. 1) suggested that the screening procedure was sufficiently reliable. We found that the majority (21/31) of genes known to be auxotrophic could be recovered (Supplementary Fig. 2). Meanwhile, only one gene unrelated to auxotrophy was identified as a false positive (Supplementary Fig. 2). These results indicate that our approach is not only highly sensitive but also highly specific for high-throughput genotype-phenotype association. The specificity issue is especially important considering the potential polar effect on neighboring genes of CRISPRi[17] when targeting genes transcribed as polycistronic mRNA. The gene fitness profile of each operon in Library II is shown in Supplementary Fig. 3.

With the dataset produced from the screening, we looked for an effect of sgRNA location within an ORF on their activities. We combined sgRNAs from Library I whose corresponding genes were shown to be true positives, thus constructing a "functional" sgRNA set (16/22 genes, 468 sgRNAs). The absolute values of sgRNA $Z$ scores (see Methods) are a reasonable metric to evaluate their activities. We categorized these sgRNAs into subgroups according to their relative position along the ORF and observed that only the subgroup of sgRNAs located within the first 5% of the ORF proximal to the start codon exhibited enhanced activity (Fig. 1b). This was consistent with previous reports[24] but provided better resolution.

To define the minimal number of sgRNAs needed per gene for reliable hit-gene calling, we used a computational sampling approach. Considering the position-dependent sgRNA activity observed above, for each gene, we included X sgRNAs most proximal to the start codon, giving rise to 6 sgRNA subsets (X = 3, 5, 10, 15, 20, 30). We then recalculated the fitness for each gene based on the sampled sgRNA subset. This sampling strategy is termed as "position", in contrast to "random" method to select sgRNA subset. Applying these two methods to 16 true positive genes in Library I, we determined that 10 sgRNAs/gene is sufficient to pick out auxotrophic genes in competitive growth over 10 doublings (Fig. 1c). In this process, we also observed that the "position" method generally outperformed random sgRNA

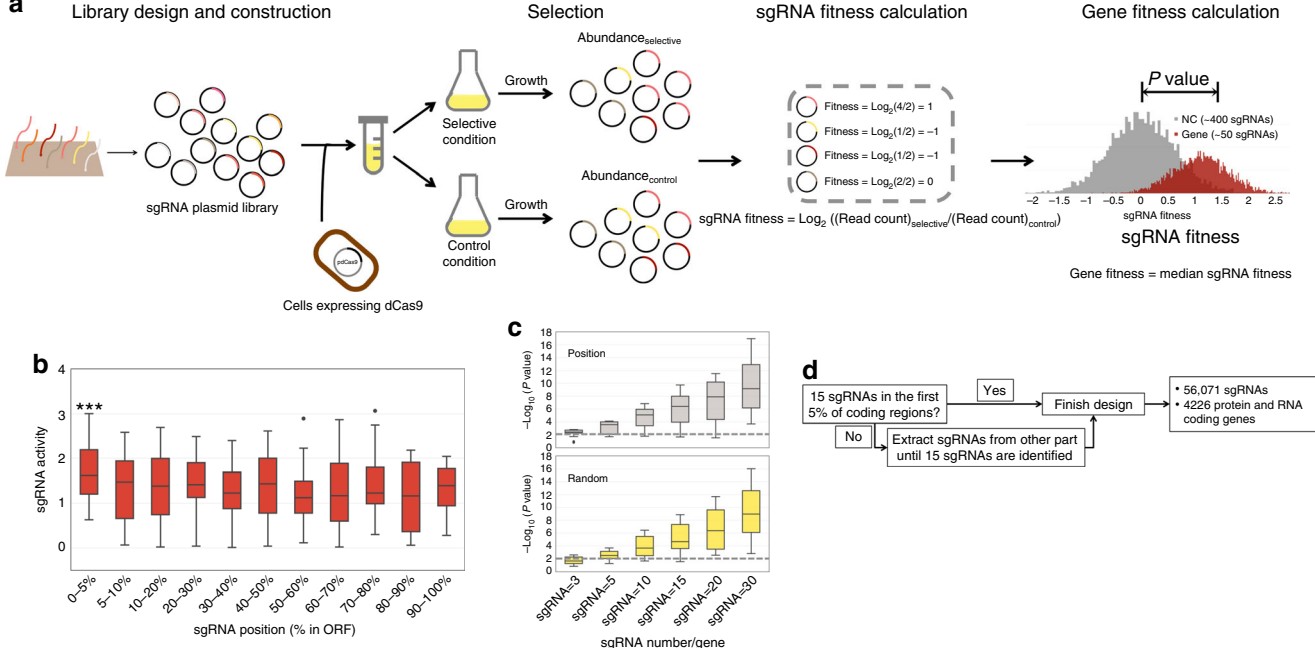

**Fig. 1** Overview of this work. **a** Proof-of-concept demonstration of the CRISPRi pooled screening for high-throughput functional genomics in *E. coli*. An sgRNA library targeting genes of interest is synthesized on a DNA microarray. Oligonucleotides are amplified and cloned into expression plasmids, transformed into *E. coli* expressing dCas9 protein, resulting in cell libraries. The cell libraries are grown under selective and control conditions. NGS libraries are constructed based on the extracted plasmids to determine the log$_2$ change of each sgRNA between the selective and control conditions (sgRNA fitness). The sgRNA fitness distribution (red histogram) of each gene is compared with that of control sgRNAs (no target site in the *E. coli* genome; gray histogram) to evaluate the extent to which this gene is associated with relevant phenotypes (selective conditions). In the first part of this work, we designed a tiling sgRNA library composed of 2281 members targeting 44 genes with known phenotypes to evaluate the activities of these sgRNAs. **b** The absolute values of the $Z$ scores for each sgRNA targeting the true positive genes were extracted, and the distribution of each group (categorized via position in the ORFs) against all 468 sgRNAs was quantified by a two-tailed MWU test. Triple asterisks indicate $P < 0.01$. For clarity, only sgRNAs with absolute $Z$ scores of 0–4 were plotted. **c** The minimal number of sgRNAs per gene for reliable hit-gene calling. Results are shown for sampling of 3, 5, 10, 15, 20, and 30 sgRNAs for each gene (16 true positive genes in Library I). Two algorithms (position (Supplementary Fig. 4, see Methods) and random) were applied to determine the priority of sgRNA selection during sampling. Results are presented as box plots of the −Log$_{10}$ ($P$ value) (MWU test) for the genes recalculated with the sampled sgRNA subset. Dashed line refers to $P = 0.01$. **d** In the second part of this work, we designed a genome-scale sgRNA library for *E. coli* based on the rules learned from the tiling experiment and performed screening experiments to test our methods

selection (Fig. 1c) when the available sgRNA subset was small. Therefore, we further optimized our hit-gene calling algorithm by searching for the sgRNA subset with the best statistical significance based on position priority (Supplementary Fig. 4).

**Design and preparation of the genome-wide sgRNA library.** The tiling CRISPRi screening provided a set of rules enabling us to design a high-quality genome-scale sgRNA library (Fig. 1d). Based on the result that 10 sgRNAs/gene were sufficient for reliable hit-gene calling and there was an increase in statistical significance if more sgRNAs were available, we chose a library size of 15 sgRNAs/gene to ensure robust performance for genes with moderate phenotypes. Moreover, the result of active sgRNA positioning resulted in our selection of as many sgRNAs as possible from within the first 5% of the ORF. In addition, for genes with multiple copies in the genome, we used the BLASTN program to categorize genes with highly similar sequences into clusters (Supplementary Data 2) and designed sgRNAs to target all members of a cluster. Hence, genes in one cluster are regarded as functionally identical. We designated this approach as the "cluster" strategy. Following these rules, we designed a genome-scale CRISPRi sgRNA library consisting of 55,671 sgRNAs (Supplementary Data 3 and 4), as well as 400 negative control sgRNAs. Based on a strict off-target quality control threshold, we successfully designed at least one sgRNA for 98.6% of 4140 protein-coding genes and 79.8% of 178 RNA-coding genes. For genes with at least three sgRNAs, these percentages were 96.8% and 48.9%, respectively (Table 1). A histogram of the number of sgRNAs per gene is shown in Supplementary Fig. 5. The coverage of genes at the genome level in our CRISPRi method significantly exceeds those reported for Tn-seq with similar library size, which is generally around 80% for protein-coding genes[25,26]. This comparison is still true if we assume that there are ~300 essential protein-coding genes (~7% of all protein-coding genes) that cannot be covered by transposon insertion. This difference is mainly derived from the random and even biased insertion of transposons into the chromosome in Tn-seq, in contrast to the uniform distribution of a synthetic sgRNA

library across the chromosome (Fig. 2, using the Tn-seq dataset from[25]). As reported previously[25], the transposon insertions are more densely distributed near the origin of replication but are sparse near the terminus (Fig. 2). Moreover, we noted the absence of transposon insertions within several chromosomal regions of up to 10 kb, whereas CRISPRi libraries can be designed to cover such regions that Tn-mutagenesis hits at low frequency (Fig. 2).

It is important to minimize the overrepresented or over-diluted members during library preparation, because the former might skew the population, wasting the sequencing capacity, and the latter might lead to an increase in statistical noise. To this end, we profiled the synthetic plasmid sgRNA library after transformation into the host cell (two biological replicates) via NGS. The results indicated that transformation of plasmids into the host cell with roughly 20-fold coverage was sufficient, as the library profile differed minimally before and after transformation (Table 1, Supplementary Fig. 6). Moreover, the relative abundance of the majority of sgRNAs (>80%) and genes (median sgRNA read count, >93%) is kept within 10-fold range (Supplementary Fig. 6), supporting the distribution uniformity of the synthetic library.

**Screening experiments using the genome-wide sgRNA library.** We used the genome-wide sgRNA library to perform screening experiments to test the reliability of our method (Table 2). The experimental set-up is schematically demonstrated in Supplementary Fig. 7. Briefly, we transformed the library into *E. coli* cells with (selective) and without (control) dCas9 expression to ascertain the essential gene set, using this as a benchmark to compare method performance with Tn-seq. The dCas9 expression group was then cultivated to $OD_{600} \sim 1$ and used as the seed to carry out additional screenings in minimal medium and under several stress conditions. All experiments were carried out with two biological replicates. The mapping ratio of sequencing reads to the in silico library (Supplementary Table 2), as well as the good agreement between replicates (Supplementary Fig. 8) confirmed the reliability of these experiments. The comparison of auxotrophic gene fitness between this experiment and the tiling library screening showed that our method is highly reproducible

**Table 1 Summary of genome-wide sgRNA library metrics**

| 4140 protein-coding genes | In silico | Plasmid | NC1[a] | NC2[a] |
|---|---|---|---|---|
| Genes with at least 1 sgRNA ≥20 reads (ratio vs. all genes %) | 4084 (98.6) | 4079 (98.5) | 4080 (98.6) | 4081 (98.6) |
| Genes with at least 3 sgRNAs ≥20 reads (ratio vs. all genes %) | 4006 (96.8) | 3999 (96.6) | 3994 (96.5) | 3996 (96.5) |
| *178 RNA-coding genes* | *In silico* | *Plasmid* | *NC1* | *NC2* |
| Genes with at least 1 sgRNA ≥20 reads (ratio vs. all genes %) | 142 (79.8) | 142 (79.8) | 139 (78.1) | 139 (78.1) |
| Genes with at least 3 sgRNAs ≥20 reads (ratio vs. all genes %) | 87 (48.9) | 86 (48.3) | 86 (48.3) | 85 (47.8) |

[a] Transformation with 20-fold coverage results in two biological replicates (NC1 and NC2)

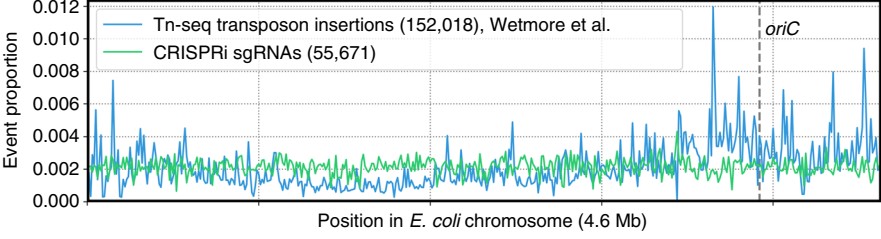

**Fig. 2** Distribution uniformity of the genome-wide sgRNA library. Comparison of distribution uniformity between the genome-scale sgRNA library and a benchmark Tn-seq transposon insertion dataset along the *E. coli* chromosome. The proportion of event number (either sgRNA or transposon insertion densities) within each window over all events across the entire *E. coli* MG1655 chromosome is shown per 10-kb sliding window

(Supplementary Fig. 9). The gene and sgRNA fitness datasets for each phenotype are shown in Supplementary Data 5–10. The sgRNA activity analysis based on the essential gene dataset supported the conclusion that only the sgRNA subgroup located within the first 5% of the coding region exhibited enhanced activity (Supplementary Fig. 10). Moreover, 91.1% of sgRNAs showed significant activity (using $Z$ score = 2 as the threshold), indicating that the majority of sgRNAs in the library are highly active.

**CRISPRi screening to assign gene essentiality.** We first investigated the protein-coding genes that, when repressed, caused a reduction in the number of *E. coli* cells over the course of CRISPRi screening in LB broth and compared the result with the essential gene set of the Keio collection[1,27] (Fig. 3a). Among 313 genes determined to be essential gene candidates during the construction of the Keio collection (Supplementary Data 11, derived from the EcoCyc database), 62.0% have a fitness value

below –6 (the threshold at which cells show no doubling in our experiment, see Methods; these genes are tagged in Supplementary Data 11), and 93.0% represent hit genes that impair cell growth significantly when their expression is knocked down (FDR = 5%), indicating the method is highly sensitive. The observed weaker phenotype found in our CRISPRi experiments in contrast to the lethal phenotype after knockout may be derived from the non-uniform distribution of sgRNA activities (Fig. 1b, Supplementary Fig. 10). The sgRNAs with poor activities give rise to residual expression of target gene, thereafter resulting in overall weaker phenotype in contrast to gene knockout method. GO analysis suggested that the hit genes identified by CRISPRi screening are mainly related to fundamental biological processes, such as translation, transcription, cell membrane or wall biosynthesis (Supplementary Fig. 11), further indicating the reliability of this method.

To assess the false positive issue that may occur because of polycistronic operons (Supplementary Fig. 12 presents the

**Table 2 The phenotypes studied in this work**

| Phenotype | Selective condition | Control condition |
|---|---|---|
| Essentiality | dCas9, LB | Empty plasmid, LB |
| Auxotrophy | MOPS | LB |
| L-Trp biosynthesis | 0.5 g/L casamino acid, MOPS | LB |
| Furfural tolerance | 0.4 g/L furfural, MOPS | initial, see Supplementary Fig. 7 |
| Isobutanol tolerance | 4 g/L isobutanol, MOPS | initial, see Supplementary Fig. 7 |

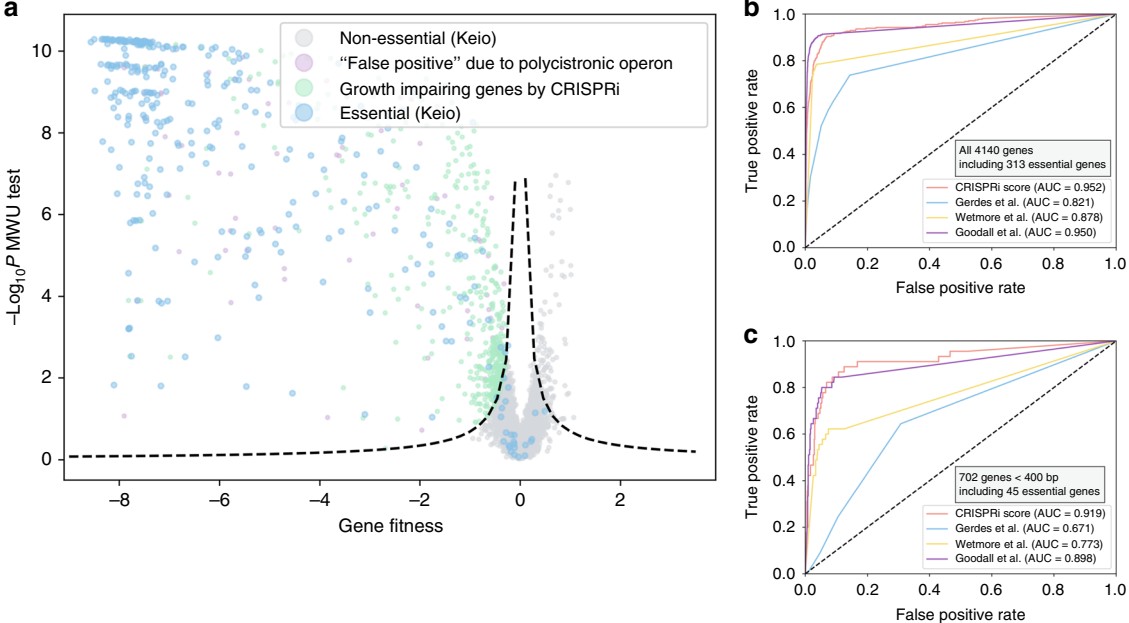

**Fig. 3** Predicting *E. coli* essential genes from CRISPRi screening. **a** Volcano plot of gene fitness and $-\mathrm{Log}_{10}P$ value of two-tailed MWU test. Dashed lines represent a threshold (FDR = 0.05) for calling hits based on the screening score (see Methods). Four groups of genes are shown: blue, 313 essential genes from the Keio collection (Supplementary Data 11); gray, non-essential genes from the Keio collection and assigned as true negatives by CRISPRi screening; purple, false-positive "required" genes assigned by CRISPRi screening, potentially because of downstream Keio collection essential genes in operons transcribed as polycistronic mRNA; green, other genes found by CRISPRi screening to significantly inhibit cell growth. **b** and **c** ROC curves indicate the performances of different methods in identifying essential genes when considering all 4140 protein-coding genes (**b**) and 702 protein-coding genes with smaller coding regions (<400 bp) (**c**). True-positive rates and false-positive rates were calculated using the gold-standard set of essential and nonessential genes from the Keio collection[1]. Shown are ROC curves for CRISPRi screening (55,671 sgRNAs) (red, CRISPRi score), a benchmark Tn-seq experiment (152,018 unique transposon insertions) (yellow, Wetmore et al.[25]), a recently reported Tn-seq dataset with unprecedented-density transposon library size (901,383 members) (purple, Goodall et al.[29]) and a widely used transposon insertion—based genetic footprinting dataset (blue, Gerdes et al.[28]). The dashed line represents a random guess of the essential genes

rationale for the forward polarity), we checked the operon structure of all growth-impairing genes identified via CRISPRi and tagged those with a downstream Keio essential gene(s) in one polycistronic operon as a false positive (Fig. 3a), which was the case for 8.0% of all hits. This result suggests that polycistronic operons result in moderate interference with our method at the genome level, at least in the essential gene case. Even though, cautions are still needed to cope with polycistronic operons via CRISPRi method. Orthogonal methods are important to provide additional information. Another interesting observation is that, the forward polar effect of CRISPRi in polycistronic operons does not always hold true. Using all essential genes suggested by Keio collection with significant growth defect phenotype in CRISPRi screening (FDR<0.05 and fitness<0) as probes, we collected 47 polycistronic operons. The non-essential counterparts upstream of these probing essential genes in the same operon were expected to exhibit growth defect phenotype due to the forward polar effect (Supplementary Fig. 12). However, 35% of them presented no phenotype in CRISPRi screening (Supplementary Fig. 13, Supplementary Table 3). This result is consistent with some cases observed in our initial tiling library screening (Supplementary Fig. 3, *aroF-tyrA*; *nirBDC-cysG*). We suggested that unknown promoters in these operons might contribute to this phenomenon. Our results indicate the poor understanding about the transcription and regulation in these operons, and further experiments at molecular level are needed to dissect them.

We also checked the more moderate reverse polarity of CRISPRi using our data, which was first described by Peters et al.[17] that repressing the downstream gene by CRISPRi perturbs the expression of its upstream counterpart in one operon. Consistent with previous reports, our result (Supplementary Fig. 14) suggests a position-dependent reverse polarity effect, which is within the window of the first 50 bp of downstream genes. This observation cannot be simply explained by the current model of CRISPRi to shutdown RNAP transcription, indicating more works are needed to understand the mechanism of CRISPRi better at the molecular level.

**Performance comparison of different methods**. To compare the performance of our approach with the state-of-the-art method in the microbial high-throughput functional genomics field. We adopted our method, a dataset from a benchmark Tn-seq study representing the largest application of this method so far[25] and a widely used *E. coli* essential gene set constructed by the transposon insertion footprinting method[28] as three binary classifiers (see Methods). During the peer review process of this paper, an elegant work was reported by Goodall et al.[29] using Tn-seq to dissect essential genes in *E. coli* based on a transposon library with an unprecedented size (901,383 unique transposon insertions, 16-fold of our sgRNA library and 6-fold of the transposon library reported by Wetmore et al. used above). We also trained their dataset as another binary classifier similarly. Taking advantage of the essential gene set of the Keio collection as the gold standard, we studied the performances of these four classifiers based on the receiver operating characteristic curve (ROC) approach (Fig. 3b). The results indicated that the CRISPRi screening achieved performance generally comparable to that of Tn-seq method with a 16-fold larger library size (area under the curve (AUC)-ROC value: CRISPRi, 0.952; Goodall et al., 0.950), despite moderately but significantly poorer performance in the low false positive rate range. The performance of Tn-seq decreased significantly as the decrease of library size (Wetmore et al. AUC-ROC, 0.878), followed by genetic footprinting strategy (AUC-ROC, 0.821). We hypothesize that CRISPRi should perform better coping with shorter genes as compared with the

transposon insertion-based methods, as transposon insertion suffers from more severe bias problem. To test this, we recruited 702 protein-coding genes shorter than 400 bp, including 45 Keio essential genes, and retested the method performances (Fig. 3c). Indeed, the AUC of CRISPRi was maintained better (0.919) whereas those for Tn-seq decreased to 0.898 (Goodall et al.[29]) and 0.773 (Wetmore et al.[25]). Note that the strain used by Keio collection[1] and all Tn-seq studies referred here[25,29] is *E. coli* K12 BW25113, while K12 MG1655 strain is used in this work. The genetic differences between these two strains are minimal[1], which should does not influence the comparison performed here.

CRISPRi screening also identified some "essential genes" missed by the gene knockout method. For example, there are two genes coding for translation elongation factors Tu 1 and Tu 2 (*tufA* and *tufB*, respectively, which share 99% nucleotide identity), the most abundant proteins in *E. coli*, and the knockout of either does not lead to lethality[30]. In contrast, a recent paper suggested that *tufA* cannot be replaced by foreign homologs in the context of *tufB* deletion[31], showing that the translation elongation factor is "essential" to the survival of *E. coli*. This phenotype was also observed in our screening results (gene fitness = −7.50), because we designed sgRNAs targeting both these two genes based on the cluster strategy in library design. We assume that by applying a more relaxed cutoff (>95% nucleotide identity in this work), it is possible to use cluster-level gene repression via CRISPRi screening to explore prokaryotic genetic interactions. Moreover, although loss of information by using clustering strategy is inevitable facing gene duplicates with identical sequences, considering the fact that two mismatches between DNA target and sgRNA are enough to abolish CRISPRi activity[32], it is still possible to study genes with highly similar (not completely identical) sequences individually. Hence, in practice, the threshold of clustering can be regarded as a customized parameter in CRISPRi screening to fulfill the requirements of research. More generally, CRISPRi screening should enable comprehensive analyses of prokaryotic genetic interactions based on the established multiplex sgRNA cloning technology[33].

We are also very interested in 22 essential genes from the Keio collection that showed no significant phenotypes in this experiment. We firstly checked the number of sgRNAs belonging to these 22 genes and found that most of them have >10 sgRNAs (Supplementary Table 4). The distributions of sgRNA number and gene length both exhibit no significant difference compared with that of all protein-coding genes ($P = 0.649$ and 0.142 by two-tailed Mann–Whitney U-test (MWU test)), suggesting no bias derived from sgRNA design is introduced when performing hit gene calling. None of them are duplicated in the *E. coli* chromosome. Moreover, functional analysis indicates that most of them are not related to paramount biological processes (Supplementary Table 4). For instance, *entD* encodes a phosphopantetheinyl transferase that is responsible for enterobactin biosynthesis and is annotated as essential in the Keio collection, in contrast to other nonessential genes in this pathway (*entABCEFH*). Indeed, 13 of these 22 genes were reported to be non-essential by an independent work using complementation assisted gene deletion in K12 MG1655 strain[34] (Supplementary Table 4). This analysis indicates that there may be some misannotations among these genes because of the recombineering efficiency or polar effect problems when trying to knock them out. To test this, we used CRISPR-Cas9 facilitated recombination (see Methods) to introduce 2 bp indel frameshift mutations to five (*chpS*, *folk*, *gpsA*, *grpE*, and *yhhQ*) of these genes. Sanger sequencing confirmed that we obtained mutant strains successfully for three of them (*chpS*, *gpsA*, and *yhhQ*) (Supplementary Fig. 15) and their viabilities were confirmed in liquid culture.

During the peer review process of this paper, Rousset et al.[35] posted a nice preprint paper, reporting a very similar strategy as our work. They constructed a random sgRNA library of roughly 90,000 members and used pooled CRISPRi screening to identify essential or phage-resistance-related genes. Interestingly, our results (essential gene identification) are highly consistent with theirs. They mentioned that there existed totally 53 genes previously regarded as essential but exhibited no growth phenotype in CRISPRi screening. Among seven potentially false negative essential genes they mentioned in the main text (*alsK*, *bcsB*, *chpS*, *entD*, *mazE*, *yafN* and *yefM*; *alsK*, *bcsB*, and *entD* were confirmed by knockout; no supplementary data was provided, disabling systematic comparison), six of them are also identified by our work (Supplementary Table 4) except for *yafN*, while this gene is not annotated as essential by Keio collection data. These results suggest that CRISPRi screening is highly reproducible.

**CRISPRi screening maps phenotypes to ncRNA-coding genes.** ncRNAs tend to be considerably shorter than their protein-coding counterparts, which may cause a statistical noise problem when mapping phenotypes to these genes using Tn-seq. It is also worth noting that some ncRNAs have multiple copies in the prokaryotic genome (such as tRNAs), thereby increasing the difficulty in applying Tn-seq or simple gene knockout methods to investigate their functions. Hence, despite growing knowledge about their importance in microbial physiology[9,10], a high-throughput method is lacking studying ncRNA functions. To test whether CRISPRi could be an alternative for this purpose, we checked the performance of our method assigning essentiality to ncRNAs. Our cluster strategy enabled handling ncRNAs with multiple copies. For example, there are seven, seven and eight copies of 23S, 16S, and 5S ribosomal RNA-coding genes, respectively, in the genome of *E. coli* K12 MG1655 (organized in seven operons). The knockdown of each cluster by CRISPRi screening leads to lethal effects (Fig. 4), as expected.

As another common class of ncRNAs, the tRNA pool within a microorganism consists of various tRNA isoacceptor families (Supplementary Fig. 16). Each family has a unique anticodon that

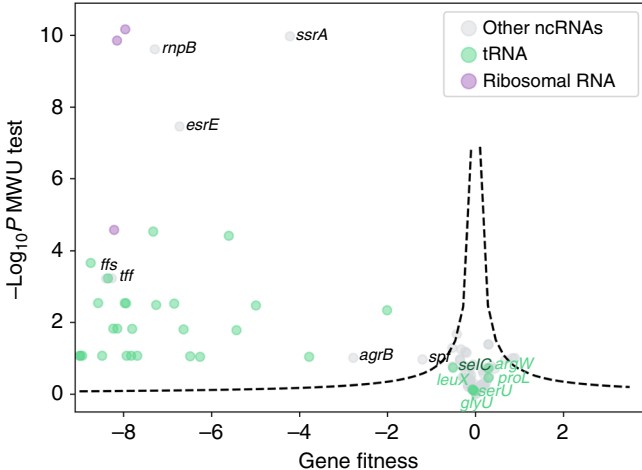

**Fig. 4** CRISPRi screening maps ncRNA contributions to fitness. Volcano plot of ncRNA-coding gene fitness and $-\text{Log}_{10}P$ value of two-tailed MWU test. Dashed lines represent a threshold (FDR = 0.05) for calling hits based on the screening score (see Methods). Three groups of gene (each dot denotes a gene cluster with one or multiple genes, see Methods) are shown: purple, ribosomal RNAs; green, tRNAs; gray, other ncRNAs. Gene names for tRNAs that were not essential to *E. coli* identified by CRISPRi in rich medium and other ncRNAs reported to confer significant growth defects are given

decodes the corresponding codon by Watson–Crick base pairing or via non-perfect base pairing at the third nucleotide with the wobble interaction[36]. Each tRNA family has a single or multiple gene copies. The *E. coli* K12 genome contains 86 tRNA genes, corresponding to 43 families, including 23 singleton families with only one gene copy (Table 3). We notice that tRNAs belonging to one family have very similar (usually identical) sequences, more conserved than tRNAs from other families. Hence, in spite of overall sequence similarity (Supplementary Fig. 16), it is possible to design sgRNAs to discriminate tRNA-coding genes at isoacceptor family level. The cluster strategy in sgRNA library design thus enables us to study the phenotype of totally 32 isoacceptor families and 61 tRNA-coding genes (Table 3). We found that most tRNA families are essential to cellular survival, except for six singleton families (Fig. 4), suggesting that the *E. coli* genome tends to have redundancy (multiple copies) for tRNA families with more important functions related to cellular survival. Among these non-essential families, *selC* encodes a special tRNA that inserts selenocysteine, an unnatural amino acid at certain in-frame TGA codons. The non-essentiality of other five tRNAs are proposed to be related to wobble interactions. For example, we hypothesize that tRNA-Arg(TCT) can also recognize codon AGG based on the non-essentiality of tRNA-Arg(CCT). Similar reasoning applies to the required tRNA-Leu(CAA) and its counterpart tRNA-Leu(TAA) (Table 3). For tRNA-Pro(GGG), it is reported that tRNA-Pro(TGG) can read all other proline codons in *Salmonella* strains[37]. For tRNA-Gly(CCC) and tRNA-Ser(CGA), the tRNA responsible for the wobble interaction could not be ascertained because of the availability of multiple candidates. It is also interesting to find that repression of tRNA-Cys(GCA), the only tRNA responsible for decoding the codon to incorporate cysteine, impaired cell growth only moderately. Although tRNA utilization governs the efficiency and accuracy of translation, the multiple-copy and sequence homology issues make it very hard to systematically study their functions by conventional mutagenesis. As the first comprehensive study of this topic in *E. coli* as we know, our experiment shows that CRISPRi screening could act as a powerful tool to better understand the biological roles played by different tRNAs (families).

Finally, we turned to other ncRNAs, seven of which showed impaired growth in our screening (Fig. 4), including the well-known RNA component (*ffs*) of the signal recognition particle and a subunit (*rnpB*) of RNase P involved in RNA processing. Previous studies are consistent, in that the known lethality or growth defect phenotypes of these seven ncRNAs upon knockout correlate well with the fitness values we obtained (Supplementary Table 5). These results suggest that CRISPRi pooled screening should be a better, or at least complementary method relative to hypersaturated transposon mutagenesis-based Tn-seq[6,7,38] for assigning phenotypes to ncRNAs in a high-throughput manner.

**Dissecting metabolic network via CRISPRi screening.** We next used this method to profile the auxotrophic genes in MOPS medium (10 g/L glucose). Our results correlated well with dataset suggested by the Keio collection[1] (Fig. 5a). GO analysis revealed that the screening results were enriched for auxotrophic genes involved in many fundamental biosynthetic processes (Fig. 5b). To cope with the inconsistency of auxotrophic genes among different reports, we also compared our results with an experimentally identified auxotrophic gene dataset in *E. coli* on agar plates[39]. By using this dataset as a reference, we found that the binary classifier to assign auxotrophic genes trained based on our results had very good performance (AUC-ROC = 0.897, Supplementary Fig. 17), comparable to that of essential gene

**Table 3 Summary of the fitness scores of *E. coli* tRNA family**

| Cluster | Anticodon | Fitness |
|---|---|---|
| *alaV, alaT, alaU* | TGC | -7.93 |
| *alaX, alaW* | GGC | UD |
| *argQ, argV, argY, argZ* | ACG | -6.49 |
| *argX* | CCG | -5.44 |
| *argW* | CCT | 0.29 |
| *argU* | TCT | -8.99 |
| *asnT, asnV, asnU, asnW* | GTT | -8.14 |
| *aspU, aspT, aspV* | GTC | -7.83 |
| *cysT* | GCA | -3.78 |
| *glnW, glnU* | TTG | -8.94 |
| *glnX, glnV* | CTG | -9.00 |
| *gltW, gltV, gltT, gltU* | TTC | -7.33 |
| *glyT* | TCC | -7.80 |
| *glyU* | CCC | -0.03 |
| *glyW, glyY, glyX, glyV* | GCC | -8.23 |
| *hisR* | GTG | -7.97 |
| *ileV, ileT, ileU* | GAT | UD |
| *ileY, ileX* | CAT* | UD |
| *leuT, leuQ, leuV, leuP* | CAG | UD |
| *leuU* | GAG | -4.99 |
| *leuW* | TAG | -6.64 |
| *leuX* | CAA | -0.08 |
| *leuZ* | TAA | -7.26 |
| *lysT, lysV, lysQ, lysZ, lysY, lysW* | TTT | UD |
| *metU, metT#* | CAT | UD |
| *metZ, metV, metW, metY#* | CAT | -8.49 |
| *pheV, pheU* | GAA | -8.75 |
| *proK* | CGG | UD |
| *proL* | GGG | 0.29 |
| *proM* | TGG | -6.26 |
| *selC* | TCA | -0.51 |
| *serT* | TGA | -5.61 |
| *serU* | CGA | -0.06 |
| *serW, serX* | GGA | -7.69 |
| *serV* | GCT | -8.36 |
| *thrT* | GGT | UD |
| *thrU* | TGT | UD |
| *thrV* | GGT | -2.01 |
| *thrW* | CGT | UD |
| *trpT* | CCA | -6.85 |
| *tyrV, tyrT, tyrU* | GTA | -8.58 |
| *valT, valY, valX, valU, valZ* | TAC | -7.95 |
| *valW, valV* | GAC | UD |

Highlighted are tRNAs not essential to *E. coli* in rich medium (light green for common tRNAs, deep green for a special tRNA encoded by *selC* to incorporate an unnatural amino acid) and tRNAs that we failed to design any sgRNAs for (gray). In addition, * the CAT anticodon is modified with a lysidine at C34 to recognize ATA codon of isoleucine, rather than methionine, # these two clusters of tRNA genes belongs to one tRNA isotype with CAT anticodon, *UD* undetermined (failure to design sgRNA)

identification (Fig. 3b). To better understand this method, we studied the reason for false negative hits in this experiment (Supplementary Note 1). Briefly, we suggest that pooled format of screening, rather than sgRNA activity issue, should be responsible to these false negatives.

Microbial genomes contain redundant isoenzymes in the metabolic network to react to dynamic environmental changes. We explored whether our method can quantitatively measure the contribution of each isoenzyme. We first focused on the chorismate pathway, the upstream module for aromatic amino acid biosynthesis. There are three genes (*aroF*, *aroG*, and *aroH*) coding isoenzymes carrying out the first step of this pathway. Despite the knowledge that the *aroG* is responsible to most of the enzymatic activity in this reaction[40], our results showed that knockdown of each individual gene had no growth phenotype (Supplementary Fig. 18). In contrast, knockdown of *aroK* resulted in a sustainable growth defect as compared with *aroL*, both of which encode isoenzyme of shikimate kinase, catalyzing the fifth reaction in this pathway (Supplementary Fig. 18). We also identified auxotrophic *metE*, which encodes the vitamin $B_{12}$-independent (*E. coli* has no de novo vitamin $B_{12}$ biosynthesis pathway) homocysteine transmethylase as the major activity contributor to the first step in methionine biosynthesis, as compared with the vitamin $B_{12}$-dependent isoenzyme encoded by *metH* without fitness defect upon repression (Supplementary Fig. 19). Similar results can be found for cystathionine β-lyase and cysteine synthase (Supplementary Fig. 19).

Chemical genomics screening is a powerful method for dissecting metabolic network organization. As a simple proof-of-concept, we performed another screening with MOPS medium supplemented with 0.5 g/L casamino acid, which is composed of all amino acids except for tryptophan. According to the amino acid biosynthetic network of *E. coli* (Fig. 5d), we expected that most of the auxotrophic genes involved in amino acid biosynthesis should be rescued by casamino acid addition with the exception of the five genes on the network branch that leads toward tryptophan biosynthesis (*trpABCDE*). The result was consistent with our hypothesis (Fig. 5c). These results suggested that our method is a reliable quantification tool to decipher the structure of the metabolic network.

**Identification of genes carrying toxic chemical tolerance**. We performed screening in MOPS medium with 0.4 g/L furfural or 4 g/L isobutanol to profile the chemical-tolerance profile in *E. coli* at the genome level (Fig. 6a, b). Tolerance to both chemicals is of industrial interest, as furfural is a common toxic byproduct formed during the pretreatment of lignocellulose and isobutanol is regarded as a promising biofuel molecule. Among hit genes, we observed previously reported candidates whose disruptions lead to tolerance (Supplementary Table 6), suggesting the reliability of the experiment. Moreover, we also identified several ncRNAs involved in the responses to these two chemicals (Fig. 6c; for all significant genes, Supplementary Fig. 20). Among them, the knockdown of *rdlD* and *eyeA* enhanced tolerance under both of these conditions. *rdlD* encodes an antitoxin and its knockdown may result in decreased metabolic activity, a general mechanism for microorganisms to cope with stress[41]. We also found that knockdown of non-essential tRNA-Arg(CCT) (*argW*) resulted in a growth defect either in MOPS medium or under furfural stress. A similar observation was previously made that the deletion of some non-essential tRNAs results in growth phenotypes under stressed conditions[42]. In addition, we identified by GO analysis the important role of indole molecule for these two stresses. Indole is known to modulate persistence towards aminoglycoside antibiotics[43,44]. It is converted from tryptophan (reversible reaction) via lysase encoded by *tnaA*, whose knockdown

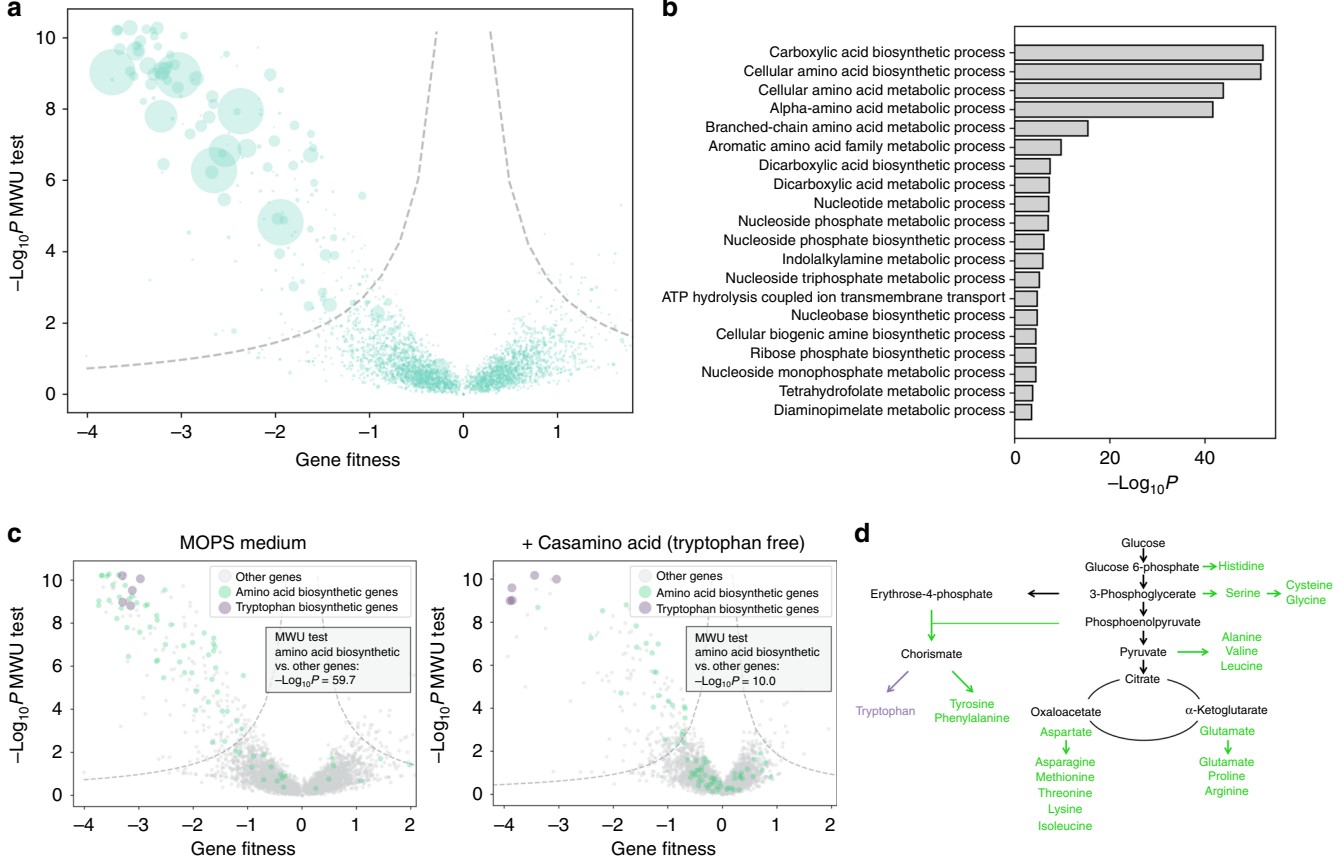

**Fig. 5** CRISPRi screening dissects the *E. coli* metabolic network. **a** Volcano plot of gene fitness in MOPS medium relative to $-\mathrm{Log}_{10}P$ value from the two-tailed MWU test. Dashed lines represent the threshold (FDR = 0.05) for calling hits based on the screening score (see Methods). The size of the scatter is proportional to the $1/\mathrm{OD}_{600}$ value of the relevant gene knockout reported with the Keio collection[1]. **b** GO enrichment analysis of auxotrophic genes identified by CRISPRi screening in MOPS medium. *P* values are derived from two-tailed Fisher exact test. **c** Comparison of gene fitness in MOPS medium with casamino acids (right) or a single carbon source (left) for *E. coli*. 78 genes responsible for amino acid biosynthesis are highlighted in green, whereas the five genes (*trpABCDE*) forming a branched pathway leading to tryptophan biosynthesis are in purple. The differences between fitness of amino acid biosynthesis genes and all other genes were quantified via the two-tailed MWU test (*P*, $10^{-10.0}$ with vs. $10^{-59.7}$ without casamino acid addition). **d** Schematic of *E. coli* amino acid biosynthetic network. Green and purple elements denote amino acid biosynthetic genes and tryptophan biosynthetic genes (*trpABCDE*), respectively, corresponding to those highlighted in (**c**)

here results in change of tolerance facing either furfural or iso-butanol. Such effect is also observed for indole transporter, TnaB and upstream tryptophan biosynthesis genes. Based on previous reports and the results here, we anticipate that indole is a general signaling molecule for bacteria to cope with diverse environmental stresses, whose molecular mechanism needs further study. Another interesting target is *pcnB*, which encodes a Poly(A) polymerase I responsible for the polyadenylation of 3′ ends of RNA molecules, thus reshaping the stability (lifetime) of bacterial transcriptome. We observed strong tolerance towards these two chemicals upon the knockdown of *pcnB*. This is consistent with our recent work[45] that the mutations in this gene contribute to the Trp production of engineered strain at the stressed conditions. We suggest that this gene is a potential global regulator for bacterial stress response and may serve as a novel target for engineering effort.

**Software package**. To facilitate the use of this method by experimental biologists, we developed an integrated Python software package including sgRNA library design and post-NGS data processing functions (Supplementary Fig. 21). The users only need to edit a configure file and provide other standard files to complete the whole pipeline on a laptop. Extensive schematic illustration is included for procedure quality control. The package is available at https://github.com/zhangchonglab/CRISPRi-functional-genomics-in-prokaryotes with a detailed user manual. We used this package to design genome-level sgRNA libraries for several model microbes (Supplementary Data 12, 13) as a demonstration.

## Discussion

Here we established CRISPRi as a robust tool to rapidly screen for loss-of-function phenotypes in a pooled format in *E. coli* at the genome level. Considering that the CRISPR-Cas system is broadly applicable in many prokaryotic organisms[13–18], we thus expect that our method will open up the possibility to screen for gene-phenotype associations or genetic interactions across many different environments and species based on either genome-wide or focused sgRNA libraries, especially for relatively rarely explored ncRNAs. The lessons we learned here about sgRNA library design in the context of prokaryotic hosts and software package should fuel the utilization of this method by more microbiologists.

As a convenient and high-throughput method, CRISPRi method can either capture a big fraction of known essential genes or suggest the potent false positive (negative) targets in other arrayed gene knockout library for further investigation. Different methods such as CRISPRi, Tn-seq and arrayed knockout probably capture different aspects of biology regarding gene essentiality or other functionalities. In this line, a combinatorial approach of different methods may further increase the reliability of prokaryotic functional genomics

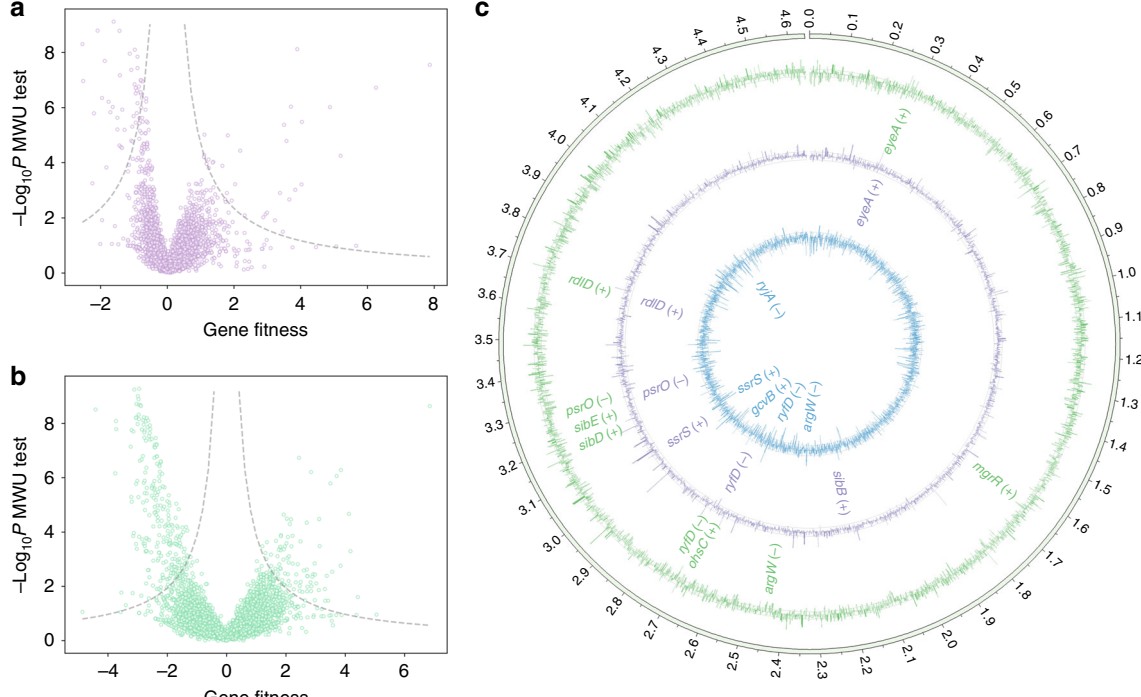

**Fig. 6** Genes conferring tolerance phenotypes with respect to the toxic chemicals isobutanol (purple) and furfural (green). **a**, **b** Volcano plot of gene fitness in MOPS medium supplemented with 4 g/L isobutanol (**a**; FDR = 0.1, dashed line) and 0.4 g/L furfural (**b**; FDR = 0.05, dashed line). **c** Genomic plots of fitness for library variants in the presence of furfural (outer, green), isobutanol (middle, purple) and no supplementation (MOPS medium alone; inner, blue). The bars of each circle indicate the fitness for genes of relevant phenotypes. ncRNAs conferring significant phenotypes ('+' for tolerance and '−' for sensitivity) (FDR < 0.05 or |fitness|>3 for auxotrophy in MOPS media and furfural tolerance, FDR < 0.1 or |fitness|>3 for isobutanol tolerance) under these conditions are highlighted

study, akin to a recent demonstration[46] where CRISPR-Cas9 and RNAi screenings were integrated in a human cell line. Besides trans-acting genetic elements like coding genes, random sgRNA library akin to the design of Rousset et al.[35], or customized library targeting intergenic regions of interest, may contribute to investigation of functional cis-acting genetic elements in bacteria, which were only addressed by tedious arrayed deletion approach previously[34].

The potential limitations of this method (which are shared with Tn-seq to a certain extent) include, first, possible false positives occurring for genes transcribed as polycistronic mRNAs because of the polar effect of CRISPRi and, second, failure to design enough sgRNAs for some genes with short coding length or microbial genomes with an extreme GC content because of the protospacer-adjacent motif (PAM) requirement. The second problem is especially of concern to cause potential biases when assigning moderate phenotypes such as tolerance to shorter genes, where only several sgRNAs are available. We suggest here to solve the first problem by adopting the recently described RNA-targeting CRISPR-Cas system[47] to modulate gene expression at the RNA level. A reengineered CRISPR-Cas9[48] or other CRISPR-Cas system[49] with divergent PAM preferences is a potential solution for the second problem. Developing a CRISPR-Cas-based gene activation system to facilitate gain-of-function screening is also important, which has been extensively demonstrated in mammalian cell lines[50] but only preliminarily in prokaryotic organisms[51].

## Methods

**DNA manipulations and reagents**. DNA purification and isolation of high-quality plasmids were performed using reagents from Omega Bio-Tek (U.S.). DNA restriction and amplification enzymes were from New England Biolabs. During plasmid construction, *E. coli* DH10B (BioMed) served as the host and was cultured in LB broth or on LB agar plates at 37 °C. Plasmids were constructed by Gibson assembly[52]. Antibiotic concentrations for kanamycin and ampicillin were 50 and

100 mg/L, respectively. MOPS medium was prepared according to standard laboratory techniques[53] (10 g/L glucose). All cultures were carried out at 37 °C.

**Strain and plasmid construction**. All strains, plasmids and primers are listed in Supplementary Tables 7 and 8. *E. coli* MG1655 (wild type) was obtained from the ATCC (700926). *E. coli* s17-1 *sfGFP* (super fold GFP) was a kind gift of the George Guoqiang Chen laboratory at Tsinghua University[54]. *E. coli* strain MCm, which was used in the screening experiments, was constructed by inserting a chloramphenicol expression cassette cloned from pKM154[55] (Addgene plasmid #13036) into the *smf* locus of wild-type *E. coli* K12 MG1655 by λ/RED recombineering[56]. *E. coli* Msac was constructed by inserting a *sacB* expression cassette (in which the J23105 promoter drives expression of *sacB* cloned from pKM154[55]; Addgene plasmid #13036) by CRISPR-Cas9 recombineering[57]. *E. coli* lyc001 is a lycopene-overproducing strain created by integrating a heterologously overexpressing *crtEIB* cluster (cloned from pTrc99a-*crt*-M[58]) into the chromosome. The dCas9 expression plasmid was constructed by replacing the promoter and resistance marker region of Addgene plasmid #44249[24] with a constitutive promoter (wild-type promoter for Cas9 from *Streptococcus pyogenes*) and the kanamycin marker cloned from pCas (Addgene plasmid # 62225)[57], resulting in pdCas9-cons. The promoter was replaced by well-characterized iGEM Anderson promoters, giving rise to plasmids pdCas9-J23109, 111, 112, 113, and 116. The empty plasmid (pKanaNC) without dCas9 that was used as the negative control in the essential gene identification experiment was constructed by amplifying pdCas9-J23111 by PCR to remove the dCas9 expression cassette followed by self-ligation via Gibson assembly. The vector for sgRNA expression was derived from pTargetF (Addgene plasmid #62226)[57] by replacing the spectinomycin marker with an ampicillin expression cassette (pTrc99a[59]) lacking the *BsaI* restriction site. The promoter region was substituted with a synthetic inducible promoter ($P_{LlacO-1}$[60]) together with the corresponding repressor expression cassette *lacI* (pTrc99a[59]), leading to pTarget-F_lac. To facilitate library insertion (which was amplified from oligonucleotides synthesized on a DNA microarray) into pTargetF_lac, pTargetF_lac_preLib was constructed by introducing two *BsaI* sites in opposite orientations between the promoter and Cas9-binding site region of pTargetF_lac. All the constructed plasmids were confirmed by Sanger sequencing.

**Assay of the CRISPRi system at bulk-population level**. To reduce the noise introduced during selection, we aimed to develop a constitutive dCas9 expression plasmid and inducible sgRNA expression system (Supplementary Fig. 22a). To this

end, we constructed a series of dCas9 expression plasmids under the control of constitutive iGEM Anderson promoters with a series of expression strengths as described above. We used strongly inducible $P_{LlacO-1}$ promoters with a well-defined TSS and tight regulation to drive sgRNA expression. However, we failed to observe any inducible profile in a test repressing *sfGFP* expression in *E. coli* owing to expression leakage and instead found that repression activity was generally determined by the strength of dCas9 expression (Supplementary Fig. 22b; the repression level is proportional to the strength of the Anderson promoter, such that J23111>J23116>J23109>J23113>J23112), in accordance with the assumption that in this system dCas9 is limited and sgRNAs are present in abundance. These results suggested that a moderate level of sgRNA expression was sufficient to drive sustainable CRISPRi activity, which is consistent with the fact that inducible CRISPR systems developed thus far in prokaryotes have been based on the regulation of dCas9 expression[17,24,54,61], and the vector backbone we used here for sgRNA expression has a relatively high copy number (pMB1 origin, ~15–20 copies/cell). Based on the results from dCas9 constructs with diverse expression strengths (Supplementary Fig. 22b), we used the pdCas9-J23111 (the strongest promoter among the five constructs) and pTargetF_lac plasmids for the following work, exploiting the leaky expression of sgRNA from the $P_{LlacO-1}$ promoter, as this provides sustained repression activity, enabling further tuning, if required. The plasmid map of pdCas9-J23111 as well as pTargetF_lac are deposited at following link:

 https://benchling.com/s/seq-UtUqPXinL0XxdjVCAPHG
 https://benchling.com/s/seq-ht6aO2giauphixQSGbP8

For fluorescence characterization mentioned above, overnight LB cultures (with ampicillin and kanamycin) from a single colony of *E. coli* s17-1 *sfGFP* containing the relevant dCas9 and sgRNA (or control sgRNA without a potential target in the *E. coli* genome) expression plasmids were individually incubated in 10 mL fresh LB medium in 50-mL flasks (initial $OD_{600} = 0.02$) with or without 1 mM isopropyl β-D-1-thiogalactopyranoside. Subsequently, cells were cultivated for 12 and 26 h, and fluorescence was measured with an F-2500 Hitachi Fluorescence Reader (excitation, 488 nm; emission, 510 nm). Fluorescence was normalized to the culture $OD_{600}$ value measured on an Amersham Bioscience spectrophotometer. The repression ratio was calculated by comparing the relative fluorescence with respect to the control strain expressing the non-targeting sgRNA.

We further tested the reliability of our optimized system (pdCas9-J23111, pTargetF_lac) by targeting the CRISPRi machinery to *crtE* (integrated at the *ldhA* locus) in the lycopene biosynthesis pathway (Supplementary Fig. 22c) and *sacB* (integrated at the *smf* locus) conferring cellular toxicity in the presence of sucrose (Supplementary Fig. 22d). Our results confirmed that this system could be applied to repress gene expression from diverse loci in the *E. coli* chromosome.

To characterize lycopene accumulation, overnight LB cultures (with ampicillin and kanamycin) from a single colony of *E. coli* lyc001 containing dCas9-J23111 and pTargetF_lac_crtE1/2 (or control sgRNA without a potential target in the *E. coli* genome) were individually incubated in 10 mL fresh LB medium in 50-mL flasks (initial $OD_{600} = 0.02$). Subsequently, fermentation was carried out for 24 h and lycopene was measured as reported[58] (Supplementary Fig. 22c). The titer was normalized to the culture $OD_{600}$ value.

For growth testing of *E. coli* Msac (*E. coli* K12 MG1655 derivate with a *sacB* integration), selective agar plates were prepared by adding 500 g/L filter-sterilized sucrose stock solution to autoclaved sodium chloride—free LB broth (1.8% agar) to a final concentration of 100 g/L. A single colony of *E. coli* Msac containing dCas9-J23111, pTargetF_lac_sacB1/2 (or control sgRNA without a potential target in the *E. coli* genome) was streaked and cultivated at 30 °C for 24 h, and the growth phenotype was measured (Supplementary Fig. 22d).

**Characterization of the CRISPRi system at single cell level.** To demonstrate the gene repression ability of CRISPRi system more clearly, it is important to investigate gene expression profile at the single-cell level. To this end, we used the s17-1 (Supplementary Fig. 22b) strain as the host and the relevant *sfGFP* as our target. We randomly selected 6 sgRNAs targeting the non-template strand of *sfGFP* ORF region (sgRNA_36, 83, 166, 257, 414, 631; sgRNA_36 was used in Supplementary Fig. 22b; the number in the name of each sgRNA represents the position of the first guanine within the PAM region (NGG) in the coding region). These plasmids as well as the negative control sgRNA were transformed into s17-1 strain carrying pdCas9-J23111. MCm/pdCas9 + sgRNA_control strain was used as a non-fluorescent reference. Overnight LB cultures (with ampicillin and kanamycin) from a single colony of these strains were individually incubated in 10 mL fresh LB medium (with ampicillin and kanamycin) in 50-mL flasks (initial $OD_{600} = 0.02$). Subsequently, cells were cultivated for 3.5 h and $OD_{600}$ reached around 0.6. The culture was diluted by 500-fold into autoclaved PBS buffer and subjected to flow cytometry analysis (BD LSRFortessa). In total 10,000 events within the gate for live cells defined by FSC and SSC were collected for each sample.

The result (Supplementary Fig. 23) shows that the cell population with repressed *sfGFP* expression exhibits log normal distribution of *sfGFP* fluorescence. Weak skewness is observed for all sgRNAs tested here, suggesting a sustainable gene knockdown efficacy across the cell population at the single-cell level. Moreover, the noise of *sfGFP* expression after CRISPRi treatment at single-cell level is comparable or even smaller than that of positive control group with a non-targeting sgRNA. Hence, we deduce that CRISPRi machinery does not lead to increased expression noise at the single-cell level, which is important for reliable

CRISPRi-based screening in a pooled format. It is also worthy noting that the diversity of sgRNA repression efficiency is observed here (as much as 10-fold, sgRNA_166 vs. sgRNA_414), which is consistent with our conclusions at the functional level (Fig. 1b, Supplementary Fig. 10).

**Indel mutation introduction by CRISPR-Cas9 recombineering.** We used CRISPR-Cas9 recombineering reported by Jiang et al[57] to introduce indel mutations. The relevant sgRNAs (Supplementary Table 7) were designed to target the first 200 bp of the coding region belong to target genes. Oligonucleotides as recombination DNA donor were designed as the lagging strand (Supplementary Table 8) to delete the guanine dimer ("GG") in DNA PAM region targeted by the corresponding sgRNA. Thus the introduced 2 bp indel is enriched by eliminating the lethal DNA cleavage via Cas9 nuclease and leads to the frameshift mutation of relevant ORF. Primers within the ORF flanking the indel were designed (Supplementary Table 8) to amplify the target region from survival colonies and the PCR products were subjected for Sanger sequencing. Usually tens of colonies were obtained on selective agar plates for the three successfully disrupted genes (*chpS*, *gpsA*, and *yhhQ*) after overnight cultivation following electroporation. PCR and Sanger sequencing suggested that most of these colonies contained designed frameshift mutations. *E. coli* K12 MG1655 was used in this assay and the obtained mutants were cultivated in liquid LB culture to confirm their growth.

**Design and preparation of the sgRNA library.** The design and preparation for the tiling and for the genome-wide sgRNA libraries were generally the same, and thus we mainly introduce below the protocol and parameters used to prepare the genome-wide library and highlight the main differences between these two. The *E. coli* K12 MG1655 genome sequence and relevant protein- or RNA-coding gene annotation of NC_000913.3 was used for the sgRNA library (20-mer) design. The SeqMap package[62] was used to check potential off-target sites of the designed sgRNAs by searching for N20NG(A)G 23-mers in NC_000913.3 with a tolerance setting of five mismatches. Customized scoring metrics inferred from previous reports[32,63] and illustrated in Supplementary Fig. 24 are designed to evaluate off-target sites identified by SeqMap. Briefly, the protospacer region of potential off-target sites is divided into three parts (8, 5, and 7 nt, from the 5′ end to the 3′ end as Region III, II and I, respectively) according to the distance to the PAM site. We used this scoring metric because mismatches are generally better tolerated at the 5′ end of the 20-nt targeting region of the sgRNA than at the 3′ end (proximal to PAM)[64]. If the PAM site of the off-target 23-mer is "NGG", the mismatch penalty in the abovementioned three regions was set as 2.5, 4.5, and 8, respectively, and the penalty set for "NAG" was 3, 7, and 10, respectively. The penalty was automatically set to be 100 otherwise. The off-target site was considered significant when Σ(penalty × mismatch)<threshold, where relevant sgRNAs were eliminated from further processing. We use a threshold of 11 for the tiling library design to maximize the number of sgRNAs designed for each gene and of 21 for the genome-scale library design to minimize the potential off-target effect leading to false positives in the screening experiments. According to a recent report comprehensively assessing off-target effects of the CRISPRi system via a partially degenerate library of variants[32] and from which we adopted the off-target threshold settings described above, one mismatch in Region I and another in Region III completely abolish CRISPRi activity[32]. Based on our much more strict off-target cutoff setting (21), even sgRNAs identified to have potential off-target effects with two mismatches in Region I and one in Region II are still removed from the library (4.5 + 8 × 2 < 21). In this manner, we applied an off-target threshold comparable to the benchmark[32] in the tiling library design and a more stringent one in the genome-wide library design, thereby minimizing the potential off-target effect. We also used a GC-content threshold (≥30% and ≤85%) to maximize the sgRNA activities[32]. Based on these principles, for the tiling library, every potential sgRNA—from 5′ to 3′ in the coding region and with the format 20 nucleotides—NGG (N20NGG, with NGG representing the PAM region)—targeting the non-template strand of the coding region[24] was checked accordingly until 50 sgRNAs passing this quality control step were extracted or until the stop codon was reached. For the genome-scale library, sgRNAs were designed first to target the 'active' region (the 5% of the coding region nearest to the start codon); as many were designed as possible according to the optimized rules learned from our tiling library screening (Fig. 1b). For a given gene, once 15 sgRNAs were designed, the design process stopped. If 15 sgRNAs could not be designed within that particular region, sgRNA design continued to uniformly select (15−x) sgRNAs from the remaining part of the coding region, where x denotes the sgRNA number designed in the 'active' region. The sgRNAs were named "gene_p" according to the position (p) of the first guanine within the PAM region (NGG) in the coding region (e.g., rsmE_9, N20 = GTTCAGGATGATAAATGCGG). In addition, we designed negative control sgRNAs by searching random 23-mers (with the format N20NGG) with the proper GC content to select those without any potential off-target candidates as identified by SeqMap (≤5 mismatches).

In the design of the genome-scale sgRNA library, to cope with genes with multiple copies in the genome, we used BLASTN with default parameters and a strict threshold (>95% identity, >95% hit coverage and >95% query coverage) to categorize genes with highly similar sequences into clusters and then designed sgRNAs to target every member of a cluster. Hence, genes in one cluster were regarded as functionally identical. We designated this approach as the cluster strategy (files for *E. coli* genome-wide gene clustering are available in Supplementary

Data [2]). Moreover, we divided the genome-scale sgRNA library into ten sublibraries according to the functions of the corresponding gene products (Supplementary Data [14]). Customized barcode sequences were accordingly incorporated within the region flanking the N20 variable part of the library, enabling PCR amplification of these libraries separately from pooled DNA oligomers based on customized primers. For the in silico design of tiling and genome-scale sgRNA libraries, see Supplementary Data [1] and [3]. The operon structures of genes during the tiling library design were determined by RegulonDB database[65].

The designed sgRNAs were synthesized as oligomers on a microarray, PCR amplified and used to generate a plasmid library by Golden Gate Assembly[66] with BsaI-digested pTargetF_lac_preLib as the backbone vector. We confirmed the quality of the library by Sanger sequencing of 47 colonies picked from the agar plate after transformation, 42 of which could be perfectly mapped back to the in silico design and 5 of which exhibited mismatch or indel mutations in contrast to the corresponding member in the library, which is consistent with the reported error rate of the current massively parallel DNA oligomer synthesis technology[3]. In addition, NGS was performed as described above to profile the genome-scale library, further validating its high quality (Table [1], Supplementary Fig. [6]).

**Screening experiments**. For a schematic illustration of the experimental design for the genome-scale library screening, see Supplementary Fig. [7]. The library plasmids were transformed by electroporation into E. coli MCm carrying the pdCas9-J23111 or pKanaNC (negative control) plasmid. Briefly, E. coli MCm cells containing pdCas9-J23111 or pKanaNC were grown in 100 mL LB broth at 37 °C until an $OD_{600}$ of 0.8 was reached. The flask was then placed on ice and all subsequent steps were performed on ice. The cells were collected by centrifugation, washed five times in ice-cold deionized water and resuspended in 6 mL 15% glycerol. The prepared competent cells were mixed with 500 ng library plasmid/mL competent cells, divided into 100-μL aliquots and loaded into 25-well electroporation plates. The electroporation was performed via a BTX Harvard apparatus ECM 630 High Throughput Electroporation System using an optimized parameter setting (2.1 kV, 1 kΩ, 25 μF). The library was independently transformed twice into either MCm/pdCas9-J23111 or MCm/pKanaNC, providing two biological replicates for each. Typically, with this protocol, we obtained around $10^5$ colonies per well. To achieve a proper coverage for the library (~60,000 members), we electroporated 32 wells of cells for each biological replicate. In addition, we tested the ratio that multiple plasmids were transformed into one cell, which might mislead the result analysis. To this end, we replaced the ampicillin resistance cassette of pTarget-F_lac with a chloramphenicol marker. These two plasmids were mixed (1:1) and the electroporation experiment was performed as described above. The resulting recovered culture was streaked onto three kinds of agar plates containing different antibiotics (kanamycin + ampicillin; kanamycin + chloramphenicol; kanamycin + ampicillin + chloramphenicol). The result suggested a ratio of multiple-plasmid-cotransformation less than $1/10^4$, which might be even smaller due to the detection limit of this method used here. Hence, we concluded that the majority of the cell population contained only one sgRNA plasmid, which rendered the following analysis reliable.

The transformed cells were incubated in LB broth (1:4, v/v) for 1 h at 37 °C to recover and washed with fresh LB broth (1:1, v/v) once. We then took 50 μL of the resulting culture to test the real transformation efficiency by dilution, streaking the diluted culture onto LB agar plates with kanamycin and ampicillin and counting the colonies after overnight incubation at 37 °C. The results confirmed that the coverage for each replicate was at least 20-fold. The remaining part of each sample was incubated with 100 mL LB broth (with kanamycin and ampicillin) in a 500-mL flask with shaking at 37 °C until $OD_{600}$ ~ 1.0 was reached (~9 h), allowing for around fifteen doublings ($10^5$ colonies per well × 30 wells = $3 \times 10^6$ cells of initial incubation. Cultivation to $OD_{600}$ ~1 results in around $10^9$ cells/mL × 100 mL = $10^{11}$ cells. $\log_2(10^{11}/(3 \times 10^6))$ = 15). For each of the four samples (two biological replicates of MCm/pdCas9-J23111 and pKanaNC (control) with the sgRNA library), we took 5 mL of culture to extract the plasmids. The cultures representing the two replicates of MCm/pdCas9-J23111 with the sgRNA library were further washed with fresh MOPS medium (1:1, v/v) once and used to seed cultures in 100 mL fresh medium in a 500-mL flask with shaking (LB, MOPS, MOPS + 0.5 g/L casamino acid, MOPS + 4 g/L isobutanol, MOPS + 0.4 g/L furfural) with an initial $OD_{600}$ of ~0.03, thereby constructing two biological replicates for each tested phenotype. The cultures representing the two replicates of MCm/pdCas9-J23111 with the sgRNA library were also mixed together, serving as the initial library for the following phenotypes to be tested (Supplementary Fig. [7]). Akin to the essential gene identification experiment, we cultivated these cultures to $OD_{600}$ of ~1.0, thus allowing the cells to reproduce for around five doubling times for each experiment, $\log_2(1/0.03)$ = ~5, and extracted plasmid from 5 mL of each culture. Five doublings during screening experiments were designed to reduce the risk of undetected de novo mutation in the bacterial genome, which might influence growth and lead to noise in final data. Meanwhile, five doublings is reliable for hit calling of auxotrophic genes (Supplementary Fig. [17], AUC-ROC = 0.9). In principle, five doublings can amplify the fitness advantage of 15% by 2-fold ($1.15^5$–2), which is sufficient for NGS to detect. Actually, pre-experiments suggested no significant differences between five and ten doublings screening in our case.

The resolution limit of our method is dependent on the ability to detect a cell exhibiting no division during the process after screening experiment. Here, we assume that the bulk population has the same doubling rate with that of the wild type cell. Hence, the growth of the bulk population leads to the dilution of cell with impaired growth or lethality. The factor limiting experiment resolution can be expressed as the following equation:

$$\text{Resolution limit} = \min\big(\log_2(\text{sequencing depth}), \text{cell doublings during screening}\big)$$

Generally, the median read count of one sgRNA in the genome-scale sgRNA library was ~100 (Supplementary Fig. [6]a). Considering the number of doublings was ~15 for essential gene identification, we reasoned that the resolution of our method for gene dropout screenings is ~6 to 7 in essential gene search ($2^6 < 100 < 2^7$). For example, in the essential gene screening experiment, we used a fitness threshold (≤−6) to infer gene essentiality, whose knockdown led to no cell division. We can improve this resolution by increasing the sequencing capacity applied to each NGS library (currently 10 million reads per library). In the following experiments (auxotrophy, casamino acid addition and chemical tolerance), because we only allowed the cells to grow for about five doubling times, the limiting factor controlling the resolution—in these cases, a fitness level of about −5 is expected for cells carrying a gene knockdown without any growth—is the doubling number rather than the sequencing depth of the NGS libraries. This issue should be carefully considered when designing experiments for this method. We note that when NGS depth is not the limiting issue for method resolution, normalization by generation in experiment, as suggested previously[67], is helpful to make the final data (gene fitness) comparable across different experiments with varieties of doublings. We also suggest that strictly inducible CRISPRi system is probably better for essential gene screening than protocol used in this work. By restricting the screening procedure into 5–6 doublings, 100 × NGS can reliably dissect essential genes and discriminate from those with only growth impairment. For the tiling library experiment, a similar protocol was followed except for the initial $OD_{600}$ = 0.001 and thus the cells underwent ten doublings during the selection.

**NGS library preparation and sequencing**. We first confirmed the robust maintenance of both dCas9 and sgRNA expression plasmids by gel electrophoresis after plasmid extraction. The purified plasmids were used as templates for PCR to amplify the N20 region of library sgRNAs (for the tiling library: 50-μL × 4 reactions for each library, 50 ng template per reaction, PF/R_pTargetLacNGS_SE75, Q5 polymerase (NEB), 98 °C 30 s, 20 cycles (98 °C, 10 s; 52.4 °C, 30 s; 72 °C 10 s), 72 °C for 1 min; for the genome-scale library with an optimized condition set-up: 50-μL × 4 reactions per library, 50 ng template per reaction, PF/R_pTargetLacNGS_PE150, KAPA HiFi HotStart polymerase (KAPA Biosystems), 95 °C 3 min, 20 cycles (98 °C, 20 s; 67.5 °C, 15 s; 72 °C, 30 s), 72 °C for 1 min). The sequencing library of the genome-scale sgRNA library was prepared following the manufacturer's protocol (TruSeq DNA Nano Library Prep Kit for Illumina). Briefly, the fragments were treated with End Prep Mix for end repairing, 5′ phosphorylation and purification using Sample Purification Beads (SPBs). Then fragments were treated with A-tailing Mix for adenylated 3′ ends, followed by ligation to adaptors indexed with a "T" overhang. Subsequently, the products were purified using the SPBs and amplified by PCR for eight cycles using P5 and P7 primers, cleaned up using SPBs, validated by an Agilent 2100 Bioanalyzer (Agilent Technologies, Palo Alto, CA, USA) and quantified with a Qubit 2.0 Fluorometer (Invitrogen, Carlsbad, CA, USA). Then libraries with different indexes were multiplexed and loaded on an Illumina HiSeq instrument according to the manufacturer's instructions (Illumina, San Diego, CA, USA). Sequencing was carried out using a 2 × 150 paired-end configuration; image analysis and base calling were conducted with the HiSeq Control Software (HCS) + OLB + GAPipeline-1.6 (Illumina) on the HiSeq instrument. Approximately 10 million reads were collected for each library (Supplementary Table [2]). For the tiling library experiment, a similar approach was applied and NGS was performed on an Illumina HiSeq 2500 with the single-end 75-bp (SE75) technique.

**NGS data processing**. Generally, the data processing consists of three steps: read count mapping, sgRNA fitness calculation and gene fitness calculation. The raw NGS data were first de-multiplexed and the adaptor region was removed to produce clean data for each sequencing library. For pair-end data, we merged each of the two pairs by FLASH script[68], and those reads without detected pairs were removed. Subsequently, we removed the read longer than 194 nucleotides and from each end of read trimmed those nucleotides with the base quality below 25. Lastly, we performed a filter step based on a customized cutoff (Q10 < 1, Q20% > 85%, and Q30% > 60%) to enrich the high-quality reads. Customized python scripts were then used to extract the 20-mer variable sequences from the raw NGS data via searching for the "GCACN20GTTT" 28_mer in the sequencing reads (and the reverse complementary sequence). Any of the 28-mers carrying mutations within upstream (GCAC) or downstream (GTTT) flanking regions (4 bp each) were removed (these 28 nucleotides are derived from the more error-prone MOS while other parts of the library plasmid are from BsaI-digested pTargetF_lac). We then mapped the extracted N20 sequences back to the designed sgRNA library, through which the read number count of each sgRNA in each library was determined. As the sequencing depths differed between experiments (Supplementary Table [2]), we adjusted read counts by applying the normalization step shown as equation ([1]) to all experiments supposing n sequencing libraries. Finally, sgRNAs with <20 read counts in the initial library for each experiment were removed to increase statistical

robustness (for the definition of the initial library for each experiment, see Supplementary Fig. 7). Subsequently, the read counts for each sgRNA in two biological replicates were averaged as the geometric mean.

$$\text{Normalization factor}_i = \text{Read count}_i / \left( \sum_{i=1}^{n} \text{Read count}_i / n \right) \quad (1)$$

For each phenotype to be studied (Table 2), we defined a selective condition and a control condition for it (Supplementary Fig. 7). For example, for the phenotype of auxotrophy, growth in MOPS medium was regarded as the selective condition and growth in LB broth was treated as the control. For each phenotype, the fitness of each sgRNA was calculated via equations (2) and (3). Briefly, we firstly calculated the fitness of each sgRNA for a phenotype (see Supplementary Fig. 7) by dividing the number of reads for this sgRNA in the corresponding selective condition by the number of reads in the relevant control condition and subsequently took the $\log_2$ value (equation 2). Then, the median of fitness for all negative control sgRNAs was determined and was used to normalize the fitness data for all sgRNAs, giving rise to the final sgRNA fitness data for this phenotype (equation 3). We annotated the quality of the sgRNA fitness by checking the read counts for each sgRNA in the control condition. Those sgRNAs with <20 reads were eliminated from the following analysis to calculate the gene fitness.

$$\text{Fitness}'_{\text{sgRNA}} = \text{Log}_2 \left( \frac{(\text{Read count})_{\text{selective}}}{(\text{Read count})_{\text{control}}} \right) \quad (2)$$

$$\text{Fitness}_{\text{sgRNA}} = \text{Fitness}'_{\text{sgRNA}} - \text{median}\left( \text{Fitness}'_{\text{NC sgRNA}} \right) \quad (3)$$

To calculate the $Z$ score of each individual sgRNA to evaluate its activity (Fig. 1b, Supplementary Fig. 10), we first fit the fitness for all negative control sgRNAs to a normal distribution, giving rise to the standard deviation ($\sigma$). The $Z$ score for each sgRNA was then calculated by dividing the sgRNA fitness by the $\sigma$ value (equation 4).

$$Z_{\text{sgRNA}} = \text{Fitness}_{\text{sgRNA}} / \sigma_{\text{normalized distribution of NC sgRNA}} \quad (4)$$

In the next step, we determined the fitness for each gene of the tested phenotype and calculated the statistical significance based on the fitness of all sgRNAs belonging to this gene. We applied a framework similar to that described recently[69] with an improved version of the algorithm via a sgRNA sampling approach. We first used a simulation method to evaluate the false positive rate (FPR) of a gene to be identified as positive for a particular phenotype given the fitness profiles of its sgRNAs. To accurately define the FPR for genes with diverse numbers of sgRNAs (1–15 sgRNAs per gene in the genome-scale sgRNA library), we explored 400 negative control sgRNAs to construct 15 quasi gene sets, each consisting of 10,000 members with 1–15 sgRNAs per member, which were referred to as QGi (i = 1–15) (QG₁ has only 400 members). For each quasi gene in QGi, a $-\text{Log}_{10}P$ value was calculated based on the two-tailed MWU test of sgRNA fitness belonging to this quasi gene against fitness of all negative control sgRNAs. We then defined a score for this quasi gene incorporating both the effect size and the $P$ values as equation (5) ('Abs' denotes the absolute value). The scores for all quasi genes in QGi were sorted in descending order, and the FPR value for each score was calculated via equation (6). By this definition, those quasi genes identified as positive by a particular threshold were regarded as false positives. We then used a linear interpolation method to convert the discrete data calculated in equation (6) to a function of FPR(score) (equation 7). In this manner, we obtained 15 functions, and each one defined the relationship between FPR and gene score given genes with $N$ sgRNAs ($N = 1, 2,…15$). The procedure for this treatment is schematically shown in Supplementary Fig. 25. The results show that the profile of FPR(score) for genes with 10 sgRNAs is similar to those with more sgRNAs (Supplementary Fig. 25), consistent with our argument that 10 sgRNAs per gene is generally sufficient for

robust hit-gene calling (Fig. 1b).

$$\text{Score} = \text{Abs}\left( \text{median}\left( \text{Fitness}_{\text{sgRNA}} \right) \right) \times (-\text{Log}_{10}P) \quad (5)$$

$$\text{FPR}(\lambda)_i (i = 1, 2, \dots, 15) = \left\{ \text{score}_j > \lambda, j = 1, 2, \dots, N_i \right\} / N_i \quad (6)$$

$$(N_1 = 400 \text{ and } N_{2 \sim 15} = 10000)$$

$$\text{FPR}(\text{score})_i (i = 1, 2, \dots, 15) = \text{linear interpolation} \quad (7)$$

$$(\text{data by equation 6})$$

With the functions relating the FPR with gene score for genes with a different number of sgRNAs, we calculated the fitness for each gene based on an sgRNA sampling method (schematically demonstrated as Supplementary Fig. 4), because we found that sgRNAs located within the 5′ region of the coding sequence exhibited better activities (Fig. 1b). Suppose $N$ sgRNAs belong to one particular gene. We sorted these sgRNAs based on their relative locations within the coding region (from the one most proximal to the 5′ end to that most proximal to the 3′ end). We then extracted the first $M$ sgRNAs, calculated the fitness based on median sgRNA fitness, $P$ value by two-tailed MWU test against all negative control sgRNAs, subsequently used equation (5) and equation (7) to get score as well as the FPR for this gene based on the current sgRNA subset with $M$ members ($M = 1$, 2, …, $N$). The sgRNA subset with the smallest FPR value was selected as the sgRNA set to determine all metrics for this gene. The fitness of this gene for the studied phenotype was the median of fitness of all sgRNAs belonging to the selected subset. We then took the classical Storey-Tibshirani approach[70] to convert the FPR values (a particular type of $P$ value) to Q values, which were used as the threshold for hit-gene calling (FDR < 0.05). For other screening experiments of genome-wide sgRNA library except for gene essentiality, essential genes of $E. coli$ identified by CRISPRi (fitness ≤ −4, FDR ≤ 0.01) were excluded.

It should be noted that we also tried alternative methods to calculate the FPR values and performed comparisons with the current one. For example, we tested the performance by using a simple two-tailed Student's $t$-test between sgRNA fitness belonging to the extracted subset and negative control sgRNA fitness to directly calculate the FPR value. The current algorithm outperformed the $t$-test method slightly in terms of essential gene identification ability (Supplementary Fig. 26). We assume this result might be due to the diverse activities of sgRNA (Fig. 1b, Supplementary Fig. 10), leading to the divergence from the hypothesized normal distribution of variables in a $t$-test.

**ROC-AUC to compare the performances of different methods**. To compare the performance of different methods to identify the essential genes of $E. coli$, we trained these methods as binary classifiers as described below. For the CRISPRi screening method, we used a naive approach to convert the screening results into prediction scores for the essentiality of each gene (equation 8).

$$\text{prediction}_{\text{gene}} = \left( 1 - \text{FPR}_{\text{gene}}, \text{if fitness}_{\text{gene}} < 0 \right) \quad (8)$$

$$\text{or} \left( 0, \text{if fitness}_{\text{gene}} \geq 0 \right)$$

For the Tn-seq dataset of all unique transposon insertion sites (152,018) by Wetmore et al.[25] (http://genomics.lbl.gov/supplemental/rbarseq/html/Keio/all.poolcount), we applied an optimized strategy similar to that which was recently reported[6] to give each gene a prediction score to indicate its essentiality. Briefly, for each gene annotated in the genome of $E. coli$ NC000913.2, upon which the annotation of this dataset is based, we calculated the insertion index as a unique insertion number in the coding region of this gene divided by gene length. As suggested[6], the distribution of insertion indices was bimodal, corresponding to the required (mode at 0) and non-required models (Supplementary Fig. 27). We thereby used a cutoff of insertion index < 0.00125 to call a gene as required. Then the following equation was used to assign a prediction score for each required gene

to indicate its essentiality quantitatively.

$$\text{prediction}_{\text{gene}} = \left(1 - e^{-(\text{local insertion index} \times \text{gene length})},\right.$$

$$\left.\text{if insertion index}_{\text{gene}} < 0.00125\right) \text{ or} \qquad (9)$$

$$\left(0, \text{if insertion index}_{\text{gene}} \geq 0.00125\right)$$

In this equation, we replaced the global insertion index (number of all unique insertion sites/genome length) reported in ref. [6] with an optimized local insertion index suggested by[28], because the biased insertion density at different chromosomal regions might lead to noise when calling hit genes. The local insertion index was calculated by counting the number of unique transposon insertion sites (N) within the 10-kb chromosomal region flanking the gene, and hence the local insertion index was equal to N/10000. This adjustment improved the performance of the trained Tn-seq classifier slightly (ROC-AUC: 0.878 vs. 0.876 for all genes, see Fig. 3b; 0.773 vs. 0.766 for genes of <400 bp, see Fig. 3c).

For the Tn-seq method based on an unprecedented library size by Goodall et al.[29], we adopted the insertion index data of every protein-coding gene in their paper (Table S1) to essential gene classifier similar as described in equation 9. The scores of assigned non-essential genes in Table S1 of their paper[29] were directly set to 0. This treatment is similar to insertion index threshold (0.00125 in dataset of Wetmore et al.[25]) used above.

For the dataset reported for the transposon-based genetic footprinting method[28] (https://www.genome.wisc.edu/Gerdes2003/table_s1.txt), the authors have annotated each gene with a tag of "essential" or "non-essential" and also assigned each essential gene with a P value. Hence, we simply used the following equation to produce the prediction score.

$$\text{prediction}_{\text{gene}} = \left(1 - P_{\text{gene}}, \text{if essential}_{\text{gene}} = \text{yes}\right) \qquad (10)$$

$$\text{or}\left(0, \text{if essential}_{\text{gene}} = \text{no}\right)$$

Thus, we trained three binary classifiers and the ROC-AUC analysis was performed based on the scikit-learn (0.19.0) Python package with the Keio collection essential genes as the gold standard (Supplementary Data 11).

**Overview of sgRNA activity landscape across ORF**. We combined sgRNAs from Library I in tiling library whose corresponding genes are located in monocistronic operons and shown to be true positives, thus constructing a "functional" sgRNA set (16 genes, 468 sgRNAs). The absolute values of sgRNA Z scores are a reasonable metric to evaluate their activities. We categorized sgRNAs in this set into subgroups according to their relative position along the ORF. We then examined the difference in activity between each subgroup and the whole population using two-tailed MWU test (Fig. 1b). Similar approach was deployed to verify the sgRNA activity positioning based on dataset from genome-scale library screening experiments performed to identify the essential gene in rich media (Supplementary Fig. 10, 337 essential genes, 4173 corresponding sgRNAs).

**Code availability**. The integrated software used for library design and NGS data processing can be found at https://github.com/zhangchonglab/CRISPRi-functional-genomics-in-prokaryotes.

**Data availability**. NGS raw data of CRISPR screening results for the tiling library and genome-scale library can be accessed from the NCBI Short Read Archive with BioProject ID PRJNA450392. The tiling and genome-scale sgRNA library (in a sublibrary format; Supplementary Data 14) are deposited in Addgene. Any other data or materials related to this work are available upon request from the corresponding author.

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

## Acknowledgements

We would like to thank Mr. Yafei Liu at Center of Biomedical Analysis, Tsinghua University for the support in flow cytometry experiment. This work was supported by the National Key Research and Development Program of China (2016YFF0202303), the National Key Scientific Instrument and Equipment Project of NSFC (21627812), the General Program of NSFC (21676156) and the Tsinghua University Initiative Scientific Research Program (20161080108).

## Author contributions

T.W., C.Z., and X.X. proposed the general design of this work. T.W. design the sgRNA library and B.L. prepared it. T.W. and J.G. performed all the experiments. T.W. and C.G. developed the software package and performed the data processing. Y.W. contributed to the establishment of the method. T.W., J.G., C.Z., and X.X. analyzed the results. Z.X. contributed by giving critical suggestions during the project design and manuscript writing. T.W., C.G., J.G. and C.Z. wrote the manuscript based on discussions and contributions of all authors. C.Z. and X.X. supervised the project.

## Additional information

**Competing interests:** The authors declare no competing interests.

