## [Peer Review File · Nature Communications]

Reviewers' comments:

Reviewer #1 (Remarks to the Author):

Recommendation:

Wang et al. present a pooled screen for identification of 67 gene knockdowns using CRISPR interference in three media conditions; MOPS, acidic, and LB. They demonstrate that such screens can successfully call individual genes that have previously been identified as critical for growth under particular conditions. They also provide a set of guidelines for building sgRNA libraries to knockdown individual gene expression across the genome, and extract genotype-phenotype interactions from resulting screens. However, in the current form I cannot recommend this paper for publishing

There are two major concerns that must be addressed. The first is that the authors have not adequately investigated the guidelines they set forth for universal library design of sgRNAs; characterization of the degree of perturbation is minimal, and the claim that individual genes within an operon can be perturbed without affecting other genes in the same operon is suspect. The second major concern is the relative weak novelty that such a genome-wide perturbation approach presents; the authors have only demonstrated that they are mostly successful at identifying previously identified targets. What they fail to do is set forth a convincing argument for why such an approach would be beneficial over previously established protocols – what can this approach provide that gene deletion screens cannot?

Major Comments:

1. Line 529: The discussion of your python script to determine off-targets is unclear. In this section, it is important for the reader to understand "Seed Sequences" of sgRNAs (Semenova, PNAS 2011, which should be cited) – that is, the fact that certain areas of the sgRNA are "more important" for determining whether the CRISPR complex will bind or not. It appears that this was considered this in the python script (which I applaud) by dividing into 3 regions depending on distance, and weighting each region accordingly. However, one mismatch at the beginning of the sgRNA and one at the end of the sgRNA would be sufficient for your script to disregard any off-target effects ($2.5+8$ or $3+10 > 11$). This seems surprisingly low to me, as I would still expect sgRNAs to be able to bind efficiently. This is less important for designing sgRNA for perturbation, as off-targets are already very unlikely to cause significant perturbation of other genes. However, in the way the paper is currently written, other scientists might easily mistake that these guidelines would be sufficient for designing sgRNAs for regular, not catalytically dead sgRNAs, where off-targets are much more of a concern. Furthermore, it appears that the authors did not consider distance from PAM for the negative controls (lines 557 to 560), and rather say that any 5 mismatches are enough (presumably because $2.5*5 > 11$). However, it has been demonstrated that the 8-12 nts are sufficient to drive sgRNA target recognition (Sander Nature Biotechnology, 2014 and Wiedenheft, PNAS 2011).

2. Revise the script to take into consideration a more rigorous definition of off-targets. Set the threshold to 20 ($2.5*8=20$, or a 12 nt sgRNA), and run all of the sgRNAs (including the controls) through this filter. If you find that any of the sgRNAs fail this new threshold, report where these off-targets occur in a new supplementary figure or table.

3. Make clearer in discussion exactly why you chose the various weights at various locations on the sgRNA – this is sound science and will help other researchers in designing sgRNAs. Also, clarify the importance of applying a higher threshold if the designed sgRNAs are to be used for guiding Cas9, where off-targets are significantly more of a concern.

4. The author's claim that they present "the first guideline for genome-wide sgRNA library design in prokaryotes" is misleading. More accurately, the present guidelines for the construction of

sgRNA libraries for subsequent use towards identifying genes important for particular phenotypes. There is a plethora of tools freely available for researchers to design a set of sgRNAs to target individual genes (for example, Liu et al, Bioinformatics 2015), from which the researchers could have identified multiple sgRNAs for each of the 67 genes they wanted to target, with minimal off-target effects. I fail to see how this would have resulted in anything different from what the author's code had provided. The real take-away that should be presented to the reader that a resolution of about 10 strains with CRISPRi knockdown are enough to resolve significant genotype-phenotype interactions, as this is the novel "guideline" they have developed.

5. The authors have not performed enough experiments to sufficiently convince the reader that they have confirmed the ability of their targets to perturb gene expression. In the main text they point to the fact that because they see differences (z -score >1), their library must be working as intended. However, this strikes me as circular logic – "Our set of CRISPRi works because we see these differences, and the differences we see are because the CRISPRi is working". I would like to see further characterization of at least subset of these constructs to see how well they actually behave at knocking down endogenous gene expression through an approach such as RT-qPCR.

6. The primary way the authors have attempted to address this is through supplementary figure S1; a figure which I have major reservations about. In the system the authors have created, IPTG should relieve repression of the sgRNAs, theoretically increasing repression rates. However, this does not appear to be the case, as the authors demonstrate in S1b – here, IPTG seems to have no impact on GFP expression. The lack of error bars/replicates is also highly concerning; the authors cannot claim that lycopene production was "significantly reduced" without providing such statistics. Finally, there is no discussion on the differences between each of the pdCas9s 109-116; why have the authors chosen to test these different Anderson promoters, and what is the difference between each of these?

7. Provide proof that the dCas9 plasmid was not lost during the selections. Was this tested in some of the selected pools? This can be checked easily by sequencing a few individual colonies picked from a plate of the streaked freezer stocks.

8. The authors should discuss the relevance of 5 or 10 generations for the selection experiment. Assuming the doubling time of *E. coli* is 30 min, this means the experiment was done only for 3-5 hours? The authors must explain reason for such a short selection. Also it will be useful to explain the 'k' term in the equations for Rho calculation.

9. The authors present no proof that knocking down expression of one gene (*nirB*) within an operon does not impact knocking down expression of other genes in the same operon. This is especially concerning because a major claim they make is the ability to resolve genotype-phenotype interactions of individual genes within operons. However, the mechanism through which CRISPRi works is to cause transcriptional interference – RNA polymerase hits dCas9 sitting on DNA and falls off, causing early termination. Therefore, within an operon, genes downstream of particular targets should also be repressed equally. This might not be the case if a gene within an operon has its own promoter (for instance, *cysG* in the *nirB-nirD-nirC-cysG* operon). However, not all of the operons the authors have investigated are as such. For instance, in the *aroF-tyrA* operon, there is only one known promoter that drives expression of both genes (RegulonDB). I'm therefore skeptical of the results presented in Figure 3, where they show that targeting *aroF* with dCas9 does not cause a selective disadvantage, while targeting *tyrA* does. The authors should provide a convincing argument as to why they were able to resolve this difference when the mechanism of CRISPRi dictates they should not. It could be possible that the authors have discovered a previously uncharacterized hidden promoter between these genes, or that the repression of *aroF* was too weak, or something else I have not considered.

10. Across this library of 3000+ sgRNAs, it is likely that a number of these sgRNAs do not actually exhibit any functionality. In fact, dCas9 binding to DNA can occasionally lead to the unintended

consequence of increasing mRNA production, especially if the target location offsets transcriptional regulatory machinery of the cell.

11. The general usefulness of this approach is not particularly demonstrated by the authors – they have only extracted previously identified genotype-phenotype interactions. All 67 genes have already demonstrated known knockouts, so in the end the authors have mainly demonstrated a new-approach at looking at old data, without showing how said approach is better. This greatly diminishes the impact of this paper has. A more impactful paper that would be the discovery of novel genotype-phenotype interactions based on this CRISPRi mechanism.

Minor comments:

1. Abstract needs revising. Make more informative:
 - a. Remove “the most intensely studied prokaryote”, “which are unique to prokaryotic genomes”
 - b. List conditions in abstract
 - c. What does “robust identification of targeted genes” mean?
2. 32- Change “The central goal of genetics” to “An important goal of genetics”
3. 76-80: Sentence is weird circular logic. “It’s important to do this because it hasn’t been done”. Revise sentence to say what new insights this approach can provide over other technologies – talking about insights into polycistronic operons would be good here.
4. 95 – I’ve never heard this hypothesis before, and it makes sense that trying to repress near the end of the ORF wouldn’t work as well as at the beginning. Cite where you got this.
5. 544 – for the “5’ upstream regions”, be more specific. Do you mean the Transcription Start Site on downstream to the end of the ORF?
6. Figure 1 – the sgRNA fitness calculation column is confusing. It took me a while to understand the values in the gray dotted box are calculated from the “example” on the left. This should be made more clear to the reader, perhaps by include these actual numbers (ie change fitness =1 to fitness = $\log_2(4/2)=1$).
 - a. Additionally, expand upon how the p-value differences between NC and gene in the histogram are calculated. Is the red histogram made up of the fitness calculation of the 50 or so sgRNAs targeting that one particular gene, and the grey histogram is all 400 sgRNAs from the NC library?
7. Figure 2 – a and b be should be lined up. Also, median rho score on 2b doesn’t provide an benefit to the paper - its just the box plots from 2a with less information and should be removed.
8. 213 – The Syntrophic exchange experiment
 - a. What happened to the false negative argG? Why was this not recovered due to syntrophic exchange
 - b. Discussion on icd false negative should also be included
9. Figure S2, and S9 axis labels are too small to read.

Reviewer #2 (Remarks to the Author):

SUMMARY, GENERAL REMARKS

- The authors report the first pooled CRISPRi screen in bacteria as a proof-of-principle experiment to determine “design principles” to optimize detection of significant growth phenotypes in co-culture; however, the design principles that the authors emphasize were previously reported, and their experimental conditions confirm known phenotypes rather than discover new ones.
- They develop a CRISPRi system for Escherichia coli based on plasmid-borne dCas9 and plasmid-borne sgRNA, and present a high-throughput cloning strategy to assemble complex sgRNA pools. Their experiments consist of pools of knockdown strains, targeting genes with previously known phenotypes for the given condition: auxotrophic genes in minimal media, and acidic-growth-sensitive genes in acidic media.
- The result of the authors’ comprehensive analysis of sgRNA efficacy largely confirms what was previously reported.. The screens focus on a small number of genes with known phenotypes (a total of 67 genes across the two conditions), 50 sgRNAs were designed per gene and included in

the library. This experimental design is more comprehensive than other bacterial CRISPRi studies to date, however it is unclear why—given the scalability of the authors’ strategy—sgRNAs were only designed to target the non-template strand of the ORF (line 135). A more comprehensive analysis of sgRNA efficacy on the non-template strand is valuable on its own, although the conclusion that the protospacer location within the targeted open reading frame is the most informative predictor of efficacy (Fig. S13) was mostly previously known. The fact that this effect is limited to the first 5% of the coding region (although this metric is slightly unclear) is interesting, and their comprehensive approach lends support to this being true generally.

- The authors report an expected pitfall of pooled screening, which is suppression of phenotypes when they are complemented in a mixed population.

MAJOR POINTS

- A major limitation of the manuscript is the lack of new biology. Given that their analysis of characteristics determining sgRNA efficacy also did not result in a meaningful contribution, this is a major issue.

- The second major limitation of the manuscript is misunderstanding of the mechanism of CRISPRi in bacteria; the claim that bacterial CRISPRi can give single-gene level information is a seriously misleading one, and is not reflected in the data they present. In part this is an issue with overstating their contribution: (lines 88-91) “Previously, the resolution of genotype-phenotype association [...] has been limited to the operon level. In our analysis, functional resolution could be achieved at the single-gene level [...].” In bacteria CRISPRi functions at the level of the transcriptional unit—there is no mechanism by which it could give single-gene level information as the authors claim. Rather, the cases the authors mention most likely reflect the existence multiple transcriptional units within an operon (see below). The authors themselves suggest that “[f]urther investigation is still needed to more comprehensively understand the mechanisms of polar and reverse polar effects when applying CRISPRi [...]” (lines 280-281) and we agree; given this, their strong assertions of single-gene information need to be adjusted.

- We suggest that the authors consult, in addition to RegulonDB, Conway et al. (2014 MBio) for additional transcriptional units within the operons studied here. As the authors suggest (lines 268-270), promoters within the operon can “mask” polar effects (although again, this language is misleading). Specifically, *cysG* and *glnA* are predicted to be transcribed as their own transcriptional unit, in addition to being transcribed in a polycistronic mRNA, potentially explaining the lack of forward polarity when targeting upstream of *cysG*, and the lack of reverse-polarity when targeting downstream of *glnA* (Fig. 3).

- The authors refer inconsistently to operons (eg. line 687: “monocistronic operons” and many comparisons to “polycistronic operons”). Mono- and polycistronic refer to the mRNA, whereas operons are groupings of genes that may be expressed as a polycistronic mRNA. Operons may have multiple, overlapping “transcriptional units”: this is obscured by the authors’ use of mono- and polycistronic to describe them.

MINOR POINTS

- lines 62-63 : transposon-based approaches can also introduce overexpression phenotypes if they carry outward-facing promoters (eg. Pozsgai et al. 2012, Appl. Environ. Microbiol.)

- Figure S2, S9 : comparison between these growth curves would be easier if they were overlaid (eg. all replicates overlaid with each other).

- lines 341-342 : why use a model which combines dCas9 binding and Cas9 cleavage activity? Is cleavage activity relevant for CRISPRi?

- lines 93-96 : the data does not actually “[challenge] the hypothesis that the knockdown level of CRISPRi can be reduced by simply locating the sgRNA more proximally to the stop codon of a coding gene”. The data provide further evidence for that hypothesis, and a more precise claim about location.

- line 126 : the authors should say “one auxotrophic gene” instead of “one essential gene” to avoid confusion.

- line 109 : CRISPRi-based screening is always reverse genetic screening, not forward. This is because the targeted mutation (in this case the target of knockdown) is known beforehand.

- lines 131-132 : the experiment with Library III did not identify genes with moderate phenotypes, because those genes were chosen to be included in the library because of previously-known phenotypes. This experiment confirms phenotypes for those genes.
- The authors should clarify what the MCm strain is and why it was chosen.
- lines 206-208 : the authors should not refer to the arginine biosynthesis genes as "false negatives". Experiments using pooled screens can only detect non-complemented phenotypes: therefore, their arg gene knockdowns lack of phenotype is a true negative in the pooled screen.
- (related to the point above) lines 222-224 : do the authors' data contribute to a new understanding of these genes and their phenotypes, given the paper they reference? It is unclear.
- lines 493-500 : the lack of inducibility of the CRISPRi system may not be surprising, and the data is consistent with their system being dCas9-limited and having sgRNA in abundance. This could be due to the relatively high copy sgRNA plasmid (pMB1 origin, ~15-20 copies/cell).

Reviewer #1 (Remarks to the Author):

Recommendation:

Wang et al. present a pooled screen for identification of 67 gene knockdowns using CRISPR interference in three media conditions; MOPS, acidic, and LB. They demonstrate that such screens can successfully call individual genes that have previously been identified as critical for growth under particular conditions. They also provide a set of guidelines for building sgRNA libraries to knockdown individual gene expression across the genome, and extract genotype-phenotype interactions from resulting screens. However, in the current form I cannot recommend this paper for publishing.

There are two major concerns that must be addressed. The first is that the authors have not adequately investigated the guidelines they set forth for universal library design of sgRNAs; characterization of the degree of perturbation is minimal, and the claim that individual genes within an operon can be perturbed without affecting other genes in the same operon is suspect. The second major concern is the relative weak novelty that such a genome-wide perturbation approach presents; the authors have only demonstrated that they are mostly successful at identifying previously identified targets. What they fail to do is set forth a convincing argument for why such an approach would be beneficial over previously established protocols – what can this approach provide that gene deletion screens cannot?

Response: we thank the reviewer for this point. First of all, in the current paper, we present a two-phase approach, where in the first minor part, we used a tiling library (67 genes with known phenotypes) screening method to learn some guidelines for more general sgRNA library design, as described in NCOMMS-17-00870 (Figure 1b in this paper). Using this guideline, we further designed a genome-wide sgRNA library and performed comprehensive screening experiments to show the performance of the method (Figure. 2 and 3) as well as shed light on some new biology of *E. coli* functional genomics (Figure 4~6). It is worthy noting that many of the guidelines presented in tiling library screening are confirmed by the data from genome-wide library screening (Figure S10). For other more specific concerns, we provided point-to-point responses at the following part.

Major Comments:

1. Line 529: The discussion of your python script to determine off-targets is unclear. In this section, it is important for the reader to understand “Seed Sequences” of sgRNAs

(Semenova, PNAS 2011, which should be cited) – that is, the fact that certain areas of the sgRNA are “more important” for determining whether the CRISPR complex will bind or not. It appears that this was considered in the python script (which I applaud) by dividing into 3 regions depending on distance, and weighting each region accordingly. However, one mismatch at the beginning of the sgRNA and one at the end of the sgRNA would be sufficient for your script to disregard any off-target effects ($2.5+8$ or $3+10 > 11$). This seems surprisingly low to me, as I would still expect sgRNAs to be able to bind efficiently. This is less important for designing sgRNA for perturbation, as off-targets are already very unlikely to cause significant perturbation of other genes. However, in the way the paper is currently written, other scientists might easily mistake that these guidelines would be sufficient for designing sgRNAs for regular, not catalytically dead sgRNAs, where off-targets are much more of a concern. Furthermore, it appears that the authors did not consider distance from PAM for the negative controls (lines 557 to 560), and rather say that any 5 mismatches are enough (presumably because $2.5*5 > 11$). However, it has been demonstrated that the 8-12 nts are sufficient to drive sgRNA target recognition (Sander Nature Biotechnology, 2014 and Wiedenheft, PNAS 2011).

Response: we thank the reviewer for this important point. Actually, we applied different off-target thresholds in the design of tiling (11) or genome-wide (21) library (see Methods). This choice is due to the different purposes of the two libraries. We used a relatively relaxed off-target threshold in tiling library design to maximize the number of sgRNAs designed per gene, by which we can test the activity of many sgRNAs via targeting only 67 genes. In contrast, because we learned from the tiling library screening that 10 sgRNAs per gene is enough for robust hit gene calling, we designed just 15 sgRNAs per gene in our genome-wide library, where naturally a more stringent off-target threshold was applied. Even though, we want to emphasize the fact that even the relaxed off-target threshold (11 in tiling library) is also comparable to those used in CRISPRi screening in mammalian cells (Gilbert 2014 Cell), and these previous works have comprehensively demonstrated that such threshold setting is ‘safe’ enough to avoid potential off-target effect. Moreover, considering the fact that prokaryotic organisms have much smaller genomes (~ 1,000 fold), the risk of off target is not that severe as experiments performed in mammalian cells. For the negative control sgRNAs, we confessed that it is possible to have all mismatches located at the 5’ region (8 nt for example), which might lead to a moderate level of unexpected gene repression. However, the occurrence of this issue is quite low ($(8/23)^5 \sim 0.005$) considering the random distribution of mismatches. Hence, we believe the problem caused by gene repression via negative control sgRNAs in our experiments should be minimal. We are going to revise our open source scripts to apply a more stringent threshold here, for example, by searching for

N12NGG without significant off-target sites to be used as negative control sgRNAs.

2. Revise the script to take into consideration a more rigorous definition of off-targets. Set the threshold to 20 ($2.5 \times 8 = 20$, or a 12 nt sgRNA), and run all of the sgRNAs (including the controls) through this filter. If you find that any of the sgRNAs fail this new threshold, report where these off-targets occur in a new supplementary figure or table.

Response: as we demonstrate above, this (21) is just the off-target threshold we apply in genome-wide sgRNA library design, the usage of which consists the major part of the current paper.

3. Make clearer in discussion exactly why you chose the various weights at various locations on the sgRNA – this is sound science and will help other researchers in designing sgRNAs. Also, clarify the importance of applying a higher threshold if the designed sgRNAs are to be used for guiding Cas9, where off-targets are significantly more of a concern.

Response: we have mentioned the literatures (Gilbert, et al. Cell (2014). Hsu. et al. Nat. Biotechnol. (2013)) (Line 603 in the manuscript) from which we adopted the threshold values in the manuscript (Line 602~604). We also emphasize the importance of applying more stringent thresholds in designing sgRNAs for genome editing in our paper posted in the preprint website (BioRxiv, doi: 10.1101/129668).

4. The author's claim that they present "the first guideline for genome-wide sgRNA library design in prokaryotes" is misleading. More accurately, the present guidelines for the construction of sgRNA libraries for subsequent use towards identifying genes important for particular phenotypes. There is a plethora of tools freely available for researchers to design a set of sgRNAs to target individual genes (for example, Liu et al, Bioinformatics 2015), from which the researchers could have identified multiple sgRNAs for each of the 67 genes they wanted to target, with minimal off-target affects. I fail to see how this would have resulted in anything different from what the author's code had provided. The real take-away that should be presented to the reader that a resolution of about 10 strains with CRISPRi knockdown are enough to resolve significant genotype-phenotype interactions, as this is the novel "guideline" they have developed.

Response: we thank the reviewer to point out this improper statement in our previous manuscript. We accordingly revised our manuscript to emphasize the importance of our guidelines for sgRNA library design towards functional genomics studies in microbes (Line 96~100). In addition, in spite the design guideline is only a minor part of the current manuscript, we want to emphasize

the importance of these guidelines in the scenario of prokaryotic functional genomics study by CRISPRi pooled screening method. Indeed, there are many open-source packages or web servers (such as Liu et al, Bioinformatics 2015) providing access for wet lab biologists to designing sgRNA library for the use of their own. However, as far as we know, all the available tools established so far are built upon knowledge basically learned from screening experiments carried out in mammalian cell lines. For example, for sgRNA design in CRISPRi experiment, these tools only provides utility to design sgRNAs targeting to the TSS of target gene. Due to the different gene organization of prokaryotic and eukaryotic genomes (operons), such design will result in the repression of all genes in one operon. Moreover, the sgRNA activity-positioning window suggested by the established tools are also far from the real world in prokaryotic genomes. For example, they suggest designing sgRNAs within the 500 nt downstream the TSS, which is beyond the entire coding region of a typical prokaryotic gene in many cases. That is the reason why we took a two-phase approach (learn and application) in this work. The result we obtained here indeed elucidated that directly applying the guidelines established in mammalian cell line screenings is not suitable to sgRNA library design for functional genomics study in prokaryotic organisms. We discuss this issue briefly in our manuscript (Line 117~128) to show the importance of the guidelines as well as our user-friendly software developed in this work.

5. The authors have not performed enough experiments to sufficiently convince the reader that they have confirmed the ability of their targets to perturb gene expression. In the main text they point to the fact that because they see differences (z -score >1), their library must be working as intended. However, this strikes me as circular logic – “Our set of CRISPRi works because we see these differences, and the differences we see are because the CRISPRi is working”. I would like to see further characterization of at least subset of these constructs to see how well they actually behave at knocking down endogenous gene expression through an approach such as RT-qPCR.

Response: we thank the reviewer for this critical point. Firstly, we believe that CRISPRi is a well-established method in *E. coli*, or more generally prokaryotic organisms. There are many previous works demonstrating the robust perturbation effect of CRISPRi complex on gene expression via methods such as RT-qPCR (Lv. et al., Metabolic Engineering, 2015). Secondly, we also tested the activity of CRISPRi system used in this work comprehensively by targeting three different genes and assayed directly the gene expression (sfGFP) or indirectly other relevant phenotypes (lycopene accumulation or toxin lethality) (Figure S18). Thirdly, our screening results

suggest that most of our sgRNAs in the library provide sustainable repression activity (Figure 1b and Figure S10). Lastly, we also used this system in our lab to repress individual genes for some metabolic engineering works and observed sustainable phenotypes (Wang. et al., unpublished data). Hence, we believe the results presented in this paper, as well as many previous pioneer works are enough to convince the readers that our system is reliable.

6. The primary way the authors have attempted to address this is through supplementary figure S1; a figure which I have major reservations about. In the system the authors have created, IPTG should relieve repression of the sgRNAs, theoretically increasing repression rates. However, this does not appear to be the case, as the authors demonstrate in S1b – here, IPTG seems to have no impact on GFP expression. The lack of error bars/replicates is also highly concerning; the authors cannot claim that lycopene production was “significantly reduced” without providing such statistics. Finally, there is no discussion on the differences between each of the pdCas9s 109-116; why have the authors chosen to test these different Anderson promoters, and what is the difference between each of these?

Response: we thank the reviewer for this point. We think the lack of inducibility of the CRISPRi system may not be surprising, and the data is consistent with our system being dCas9-limited (Figure S18b, the repression level is proportional to the strength of Anderson promoter to drive dCas9 expression, J23111 > J23116 > J23109 > J23113 > J23112) and having sgRNA (even leaky expression) in abundance. This could be due to the relatively high copy sgRNA plasmid (pMB1 origin, ~15-20 copies/cell). We provide a brief discussion about this issue in the Methods section (Line 546~556). For the lycopene issue (Figure S18c), we confessed it is a drawback not to present biological replicates. However, in our experience in metabolic engineering, the result presented in Figure S18c is very significant, evidenced by the fact that more than 10-fold titer change was observed. In this field, even some pioneer works just improves the titer by less than 2-fold after comprehensive engineering efforts. In addition, we also used the CRISPRi system developed in this work to screen for some other genetic targets related to lycopene production. Indeed, we observed many positive hits and confirmed our results by gene knockout (Wang et al. unpublished data). Here is an example below. We believe these data is sufficiently enough to prove the reliability of our system.

7. Provide proof that the dCas9 plasmid was not lost during the selections. Was this tested in some of the selected pools? This can be checked easily by sequencing a few individual colonies picked from a plate of the streaked freezer stocks.

Response: we actually tested the maintenance of either dCas9 or sgRNA plasmid before library construction by gel electrophoresis and colony PCR. We provide the relevant proofs in our preprint paper (BioRxiv, doi: 10.1101/129668). It is given here as below (plasmids extracted after tiling library screening experiments).

Figure S21 The extracted plasmids from cells after selection confirmed the maintenance of both dCas9 and sgRNA expression vectors. From left to right (for library name, see Figure S3), *LowpH_start*, *LowpH_LB4.5_R1*, *LowpH_LB4.5_R2*, *LowpH_LB_R1*, *LowpH_LB_R2*, DNA ladder, *pTarget_lac*, *pdCas9-J23111*, DNA ladder, *Min_start*, *Min_MOPS_R1*, *Min_MOPS_R2*, *Min_LB_R1*, *Min_LB_R2*.
DNA ladder: 10kb, 8kb, 6kb, 5kb, 4kb, 3kb, 2kb, 1kb.

8. The authors should discuss the relevance of 5 or 10 generations for the selection experiment. Assuming the doubling time of E. coli is 30 min, this means the experiment was done only for 3-5 hours? The authors must explain reason for such a short selection. Also it will be useful to explain the 'k' term in the equations for Rho calculation.

Response: we thank the reviewer for this important point. As we demonstrated in the manuscript, in this experiment setting our method provides sufficient resolution to identify the genes associated with studied phenotypes (Figure 3). Taking this factor into account, we minimize the incubation time to avoid the

potential *de novo* mutations on the chromosome as much as possible, because such mutations can cause false positive results in data analysis, which cannot be detected by sequencing only the variable regions in sgRNA expression plasmid. Actually, we compared the performance of different cultivation time and the result shows that 5~10 doublings is fairly enough, at least in our experiments. We eliminated the usage of ‘k’ or ‘Rho’ term in data analysis in this manuscript and unify the treatment across the manuscript. For detailed methods of data analysis, please see the Methods part.

9. The authors present no proof that knocking down expression of one gene (*nirB*) within an operon does not impact knocking down expression of other genes in the same operon. This is especially concerning because a major claim they make is the ability to resolve genotype-phenotype interactions of individual genes within operons. However, the mechanism through which CRISPRi works is to cause transcriptional interference – RNA polymerase hits dCas9 sitting on DNA and falls off, causing early termination. Therefore, within an operon, genes downstream of particular targets should also be repressed equally. This might not be the case if a gene within an operon has its own promoter (for instance, *cysG* in the *nirB-nirD-nirC-cysG* operon). However, not all of the operons the authors have investigated are as such. For instance, in the *aroF-tyrA* operon, there is only one known promoter that drives expression of both genes (RegulonDB). I’m therefore skeptical of the results presented in Figure 3, where they show that targeting *aroF* with dCas9 does not cause a selective disadvantage, while targeting *tyrA* does. The authors should provide a convincing argument as to why they were able to resolve this difference when the mechanism of CRISPRi dictates they should not. It could be possible that the authors have discovered a previously uncharacterized hidden promoter between these genes, or that the repression of *aroF* was too weak, or something else I have not considered.

Response: we thank the reviewer for this very important point. We admit that due to the mechanism of CRISPRi, the failure to assign phenotype to individual genes in polycistronic operon is a potential drawback of our method. We have adjusted our previous misleading assertion about this issue in this manuscript. This argument is no more the main conclusion in this manuscript, but rather a point needs further investigation. In fact, we believe such assignment (correctly identifying target genes in operons directly by screening) is possible in two cases (Figure S11). Firstly, when the phenotype-associating gene is located at the very upstream of the operon (*nadA-pnuC*, Figure S3); secondly, when multiple transcriptional units exist in one previously assigned polycistronic operon (*nirB-nirD-nirC-cysG* operon, Figure S3). We provide a utility in our software to reorganize the gene fitness score according to the given operon file to help the users to look into the responses of different genes in one operon. Besides, we also checked the potential false positive effect

caused by this factor in functional genomics screening (essential gene by using Keio collection set as gold standard, Figure 3a) and found such effect is moderate (~ 8%). We also talked about this issue in the Discussion section and suggested to bypass this effect by using RNA-targeting CRISPR system or simply improving the hit-gene calling algorithm (Line 488-491).

10. Across this library of 3000+ sgRNAs, it is likely that a number of these sgRNAs do not actually exhibit any functionality. In fact, dCas9 binding to DNA can occasionally lead to the unintended consequence of increasing mRNA production, especially if the target location offsets transcriptional regulatory machinery of the cell.

Response: we thank the reviewer for this interesting suggestion. In our result, we found most (>90%) sgRNAs in our library exhibit strong repression activity (Figure S10, checking the fitness of sgRNAs targeting to essential genes). About the issue to block the access of potential repressing transcription factors, because in this work we design sgRNAs within the coding region rather than the promoter part, we believe the possibility of such potential interaction should be minimal.

11. The general usefulness of this approach is not particularly demonstrated by the authors – they have only extracted previously identified genotype-phenotype interactions. All 67 genes have already demonstrated known knockouts, so in the end the authors have mainly demonstrated a new-approach at looking at old data, without showing how said approach is better. This greatly diminishes the impact of this paper has. A more impactful paper that would be the discovery of novel genotype-phenotype interactions based on this CRISPRi mechanism.

Response: As we describe above in the general part, in the genome-wide library (targeting all protein and ncRNA-coding genes) screening experiments, the advantage of this approach is comprehensively demonstrated for either methodology part or bringing fresh biological knowledge. For example, we provided evidences suggesting CRISPRi pooled screening outperforms Tn-seq, the benchmark gene-deletion method in the field of high-throughput microbial genetics (Figure 2 and 3). During this process, we suggested some potential false positive essential genes assigned by Keio collection (Figure 3a, Table S14). Moreover, we showed the power of CRISPRi screening to identify ncRNAs associated with particular phenotypes (Figure 4 and 6), a major drawback of Tn-seq, which generally misses many ncRNAs during library construction. Of particular interest is a tRNA-fitness map constructed by our new method, for the first time demonstrating the essentiality of tRNAs

encoded by *E. coli* genome. We also identified many novel (with some other known) genetic targets, which can improve the performance of strains in the stressed conditions of industrial interests (Figure 6, Figure S16). Moreover, we developed a user-friendly software package to design sgRNA library using our guidelines and performed data processing after NGS by directly feeding raw data. We believe this is especially helpful for web lab biologists with limited programming skills to have the one-stop solution to use our method.

<https://github.com/zhangchonglab/CRISPRi-functional-genomics-in-prokaryotes>

Reviewer #2 (Remarks to the Author):

SUMMARY, GENERAL REMARKS

- The authors report the first pooled CRISPRi screen in bacteria as a proof-of-principle experiment to determine “design principles” to optimize detection of significant growth phenotypes in co-culture; however, the design principles that the authors emphasize were previously reported, and their experimental conditions confirm known phenotypes rather than discover new ones.
- They develop a CRISPRi system for *Escherichia coli* based on plasmid-borne dCas9 and plasmid-borne sgRNA, and present a high-throughput cloning strategy to assemble complex sgRNA pools. Their experiments consist of pools of knockdown strains, targeting genes with previously known phenotypes for the given condition: auxotrophic genes in minimal media, and acidic-growth-sensitive genes in acidic media.
- The result of the authors’ comprehensive analysis of sgRNA efficacy largely confirms what was previously reported. The screens focus on a small number of genes with known phenotypes (a total of 67 genes across the two conditions), 50 sgRNAs were designed per gene and included in the library. This experimental design is more comprehensive than other bacterial CRISPRi studies to date, however it is unclear why—given the scalability of the authors’ strategy—sgRNAs were only designed to target the non-template strand of the ORF (line 135). A more comprehensive analysis of sgRNA efficacy on the non-template strand is valuable on its own, although the conclusion that the protospacer location within the targeted open reading frame is the most informative predictor of efficacy (Fig. S13) was mostly previously known. The fact that this effect is limited to the first 5% of the coding region (although this metric is slightly unclear) is interesting, and their comprehensive approach lends support to this being true generally.
- The authors report an expected pitfall of pooled screening, which is suppression of phenotypes when they are complemented in a mixed population.

MAJOR POINTS

- A major limitation of the manuscript is the lack of new biology. Given that their analysis of characteristics determining sgRNA efficacy also did not result in a meaningful contribution, this is a major issue.

Response: we thank the reviewer for this comment. As we described above, different from the last paper (NCOMMS-17-00870), this new manuscript is a novel advancement by applying the CRISPRi method and a genome-wide sgRNA library (~60,000 sgRNAs) to study the functional genomics of *E. coli*. Much novel biological insight is presented. For example, we suggested some potential false positive essential genes assigned by Keio collection (Figure 3a, Table S14). Moreover, we showed the power of CRISPRi screening to identify ncRNAs associated with particular phenotypes (Figure 4 and 6). Of particular interest is a tRNA-fitness map constructed by our new method, for the first time demonstrating the essentiality of tRNAs encoded by *E. coli* genome. We indicated some potential wobble interaction pair from this result. We also identified many novel (with some other known) genetic targets, which can improve the performance of strains in the stressed conditions of industrial interests (Figure 6, Figure S16). Besides, for the methodology part of this paper, we provided evidences suggesting CRISPRi pooled screening outperforms Tn-seq, the benchmark gene-deletion method in the field of high-throughput microbial genetics (Figure 2 and 3). One particular difference is that CRISPRi is more superior in terms of assigning phenotypes to ncRNAs, which usually have a very short coding region (missed by random transposon insertion) but important biological functions.

- The second major limitation of the manuscript is misunderstanding of the mechanism of CRISPRi in bacteria; the claim that bacterial CRISPRi can give single-gene level information is a seriously misleading one, and is not reflected in the data they present. In part this is an issue with over-stating their contribution: (lines 88-91) “Previously, the resolution of genotype-phenotype association [...] has been limited to the operon level. In our analysis, functional resolution could be achieved at the single-gene level [...]” In bacteria CRISPRi functions at the level of the transcriptional unit—there is no mechanism by which it could give single-gene level information as the authors claim. Rather, the cases the authors mention most likely reflect the existence multiple transcriptional units within an operon (see below). The authors themselves suggest that “[f]urther investigation is still needed to more comprehensively understand the mechanisms of polar and reverse polar effects when applying CRISPRi [...]” (lines 280-281) and we agree; given this, their strong assertions of single-gene information need to be adjusted.

Response: we thank the reviewer for this very important point. We admit that due to the mechanism of CRISPRi, the failure to assign phenotype to individual genes in polycistronic operon is a potential drawback of our method. We have adjusted our previous misleading assertion about this issue in this

manuscript. This argument is no more the main conclusion in this manuscript. In fact, we believe such assignment (correctly identifying target genes in operons directly by screening) is possible in two cases (Figure S11). Firstly, when the phenotype-associating gene is located at the very upstream of the operon (*nadA-pnuC*, Figure S3); secondly, when multiple transcriptional units exist in one previously assigned polycistronic operon (*nirB-nirD-nirC-cysG* operon, Figure S3). We provide a utility in our software to reorganize the gene fitness score according to the given operon file to help the users to look into the responses of different genes in one operon. Besides, we also checked the potential false positive effect caused by this factor in functional genomics screening (essential gene by using Keio collection set as gold standard, Figure 3a) and found such effect is moderate (~ 8%). We also talked about this issue in the Discussion section and suggested to bypass this effect by using RNA-targeting CRISPR system or simply improving the hit-gene calling algorithm (Line 488-491).

- We suggest that the authors consult, in addition to RegulonDB, Conway et al. (2014 MBio) for additional transcriptional units within the operons studied here. As the authors suggest (lines 268-270), promoters within the operon can “mask” polar effects (although again, this language is misleading). Specifically, *cysG* and *glnA* are predicted to be transcribed as their own transcriptional unit, in addition to being transcribed in a polycistronic mRNA, potentially explaining the lack of forward polarity when targeting upstream of *cysG*, and the lack of reverse-polarity when targeting downstream of *glnA* (Fig. 3).

Response: we thank the reviewer for this useful suggestion. We read this paper (Conway et al. (2014 MBio)) and included relevant information extracted from this paper in our post in the preprint server (BioRxiv, doi: 10.1101/129668). However, because the main conclusion of the current manuscript is shifted to a more general perspective, due to the limitation of paper length, we did not include it in the current paper.

- The authors refer inconsistently to operons (eg. line 687: “monocistronic operons” and many comparisons to “polycistronic operons”). Mono- and polycistronic refer to the mRNA, whereas operons are groupings of genes that may be expressed as a polycistronic mRNA. Operons may have multiple, overlapping “transcriptional units”: this is obscured by the authors’ use of mono- and polycistronic to describe them.

Response: we thank the reviewer for this useful suggestion. We revised our manuscript accordingly, by terming such operons as polycistronic-mRNA transcribing operons.

清华大学
Tsinghua University

Please contact us if you have any further question or concern about this manuscript.

With best regards and many thanks,

Chong Zhang

601 Yingshi Building, Department of Chemical Engineering,
Tsinghua University, Haidian District, Beijing, China 100084

Phone: 86-10-62794771

Fax: 86-10-62770304

E-mail: chongzhang@tsinghua.edu.cn

Reviewers' comments:

Reviewer #1 (Remarks to the Author):

Wang et al. provide a thorough, well-written case for pooled CRISPRi as a gene-phenotype mapping tool in prokaryotes. CRISPRi pooled screening is demonstrated to identify essential genes in *E. coli* (Figure 3), probe metabolic networks (Figure 5), identifying the phenotypes of non-coding RNAs (Figure 4), and identifying genes that confer tolerance to toxic chemicals (Figure 6). Compared with other methods used in prokaryotes, namely transposon sequencing, CRISPRi identifies fewer false positives and achieves a more uniform distribution of sgRNA across the chromosomal DNA without bias for longer genes. Moreover, pooled CRISPRi is a more transferable screening method compared with single gene knockout strategies. The authors demonstrate a wide range of uses for this tool and validate its effectiveness by identifying genes already known to be essential (while also identifying more that have not been studied.) The manuscript has been substantially improved over the previous submission and that the experimental logic is clear and the figures tell a clean story. Addition of the identification of ncRNAs is great, as it really demonstrates the advancement of the paper over previous work. A few points of confusion or critiques are detailed below. Overall, I found this paper to be of high quality—clear, thorough and significant.

Major comments:

- The insight that the best guide RNAs fall within the 5% of ORF is unsurprising based on the wealth of work reported. The relevance with respect to this screening method needs to be explained. Also, explain why multiple guide RNAs (10-15) are needed for targeting a single gene. Also, what are the actual rules the authors find from Fig 1b for designing their sgRNAs? Is it targeting the first 5% of the ORF, which doesn't seem particularly connected to the actual figure? Is it the number of sgRNAs required to call out successful hits (10)? Should be made more clear.
- The distribution of the number of guide RNAs across all the genes in the genome is not clear? Do the different numbers of guide RNAs per gene impact the experiments or introduce any bias?
- What are their criteria for a negative control? There are 400 of them – is there some threshold for what counts as a non-target. 2,3 mismatches? Did the authors check PAMs? This needs more explaining in the text around line 143.
- Is there some correlation between the 22 essential genes the authors failed to identify in fig 3a and the "strength" of the sgRNAs used to perturb them? For instance, did they have less than the 15 desired sgRNAs inhibiting them? Confirming that this is not the case would actually strengthen the paper, as it would suggest there is no biasing effect arising from their sgRNA design. Also in figure 3 it must be clarified why CRISPRi growth phenotypes are generally weaker than the Keio collection counterparts.
- Would the cluster strategy the authors introduce in line 202 cause a lack of resolution in identifying gene functionality? If highly similar sequences are treated as the same, wouldn't subtle differences in those genes be lost? I understand why this might be a necessary component of the method, given that the same sgRNAs might bind to sequences with similar composition, but I did not notice any discussion about potential downsides of this strategy. It is unclear how the cluster strategy enables the authors to study different tRNAs (line 344).
- (starting at line 267). It seems as though the authors label all genes in polycistronic operons that were deemed growth-impairing false positives if the operon contains a 'downstream Keio essential gene'. The authors acknowledge that this false positive is an overestimation, but I am not sure how they can state that 'false positive genes are expected to truly impair cell growth to a less extent compared with lethality', based on the fact that there is one Keio essential gene within the operon. There could still be other genes within the operon that are just as essential. I would think

this information could be used to probe the essentiality of other genes with a polycistronic operon.

- (lines 313-321) One part of the results analysis that could be further developed concerns 22 genes that were found to be essential using single-gene knockout mutants (Keio analysis) but were not found to have any significant phenotype using CRISPRi. Are these genes considered false positives from the Keio collection (the paper does not label them as such), and if so, did the paper make any effort to identify false positives? How did they miss these? Moreover, Table S14 reports that most of the genes are 'not related to paramount biological processes', and one example is given in the manuscript, but are there any of these 22 genes that have not been studied that may actually be essential? The end of the paragraph mentions alternative approaches to test essentiality of these genes, but more detail here would be welcome.

Reviewer #3 (Remarks to the Author):

NCOMMS-17-00870A-Z
Previously NCOMMS-17-32460

Summary

The authors present a significantly more developed manuscript than we previously reviewed, and have addressed the majority of our previous concerns regarding the former manuscript. The new manuscript presents pooled, tiling and genome-wide CRISPRi libraries in *Escherichia coli* and uses them to: (1) determine rules for targeting ORFs within operons, (2) assess gene essentiality on a genome-wide scale by identifying extreme fitness defects, and (3) explore tolerance to industrial chemicals by identifying increases in fitness.

The major contributions of this paper are as follows: systematic demonstration that targeting the first 5% of an ORF results in the strongest phenotype, demonstration that genome-wide CRISPRi outperforms Tn-seq in essential gene identification on some important parameters (i.e. identifying small essential regions), present a new conceptual approach to targeting "clusters" of genes and apply this to investigate tRNA redundancies.

Improvements since last revision

As we and the other reviewer requested, the manuscript is clearer with regard to the mechanism of CRISPRi and how targeting genes within operons may be interpreted. With a few exceptions (see below) this has been addressed.

Should be improved

The vast amount of new data added to the new manuscript does suggest that generally the authors' CRISPRi system is working. However, this does not fully address our concerns about presenting a thorough characterization of the CRISPRi system, which is important for interpreting all the presented screening data. Here are the points that need to be addressed:

1. sgRNAs cause depletion of the targeted mRNA and/or protein in the cell, and targeting a gene for knockdown has a similar effect as a gene deletion. For quantifying and demonstrating knockdown, a single-cell assay would be the most convincing. Bulk fluorescence (as shown Fig.

S18) is less informative without a growth curve or some demonstration that an equal number of cells, in an equivalent physiological state, are being compared. Presenting clearly how much variability in knockdown is important for readers to understand sources of noise in this screen, and to be able to rationally design improvements or variations for other purposes. In particular, more not-pooled experiments would clarify some of the effects from targeting within operons. For example the “false positive” result in Figure S3: does targeting *glnL* actually increase the amount of *glnA* mRNA and/or protein?

2. A comparison of CRISPRi mutants to deletion mutants (outside of the screening context) is also necessary. Something like what is presented in response to Reviewer #1 question #6 would be helpful for the reader.

3. A clearer explanation of their pooled growth experiments is needed: the explanation in the Methods section (Lines 705-717) is unclear... is the number of doublings or the sequencing depth limiting resolution? If the number of doublings is the major limitation, is there a rationale for using only 5 doublings? Further, in these experiments fitness is calculated as a ratio between a selective and control condition. Eg. for the essentials, +/-dCas9. Could the authors comment on this approach compared to a more standard fitness calculation used in the field, looking at abundance over time (gamma, etc)?

4. The auxotrophy experiments are very nice, but a few questions remain, in particular around the “false negatives”: genes whose deletions render them unable to grow in minimal media but whose knockdown by CRISPRi does not affect fitness. For those genes, what does the distribution of sgRNA phenotypes look like (i.e. is it unimodal)? Is it the case that some sgRNAs do actually “work” and some do not (i.e. does Fig1b look different for genes that work and genes that don’t work)? Or is the assertion that targeting this gene with any sgRNA does not cause a phenotype? If the latter, could the authors demonstrate that a deletion strain does/does not have a fitness defect in these conditions? Is knockdown happening for those targets in the screen conditions (qPCR, WB)?

5. We look forward to more data (Line 320-21) on the 22 genes determined to be essential by the CRISPRi method but not by deletion (although please see update to the Keio above). In particular does complementation with the gene product allow deletion? It may be a good time to note that the genetic backgrounds of the Keio collection and the strain used in this manuscript are not identical.

6. A rationale for forward and reverse polarity is presented in FigS11, but does the data support this? Do the authors observe any instances of “reverse” polarity, where targeting a downstream non-essential gene affects expression of an upstream essential gene? I think this will be an interesting question for readers and the authors have a lot of data to bring to bear on it.

7. The potential for single-gene information: the authors assert that the risk of conflating the phenotypes of all genes in an operon is “moderate” (in the letter response) based on the low percentage of false positives (fitness defect for non-essential knockdown) attributable to an essential gene in the operon. I think this argument is not a good one, because it may be rare for essential genes to be in operons with non-essential genes, but for other phenotypes it is more likely that genes will be grouped by function. Therefore the knockdowns at the operon level will almost certainly conflate the phenotypes of all member genes in a way that will require orthogonal approaches to disentangle. Caution around falsely promising single-gene phenotype information from CRISPRi is important.

8. While this manuscript includes many thoughtful comparisons to gold standards in order to validate their methodology, there is little discussion of any new biological findings even in the newer screening conditions. In the cases of the isobutanol and furfural screens, could the authors provide some explanation for why many hits overlap (i.e. any known mechanisms of toxicities for

these chemicals?), and if this was known. Were any interesting sensitivities found? It seems that ncRNAs with significant phenotypes are highlighted, but do the gene phenotypes fit with what is known or provide any new insight?

Small concerns

The set of essential genes from the construction of the Keio collection (Baba et al. 2006) should also include reference to the update to the Keio collection (Yamamoto et al. 2009) in which some designations of essentiality are changed.

- Line 45 "multiples conditions simultaneously" is not what the authors present. Most likely they mean to say in parallel or rapidly.
- Lines 80-82: the advantage is not that plasmids can be delivered without bias (because they are surely unequal in proportion) but that the targets can be designed without bias for long genes.
- Line 97: the authors do not demonstrate here that their positioning actually matters for expression level (would need qPCR, WB etc.)
- Line 180: A clearer way to describe this approach might be "sampling" instead of "subsampling", as the samples are not intended to reflect the larger distribution as they might when actually subsampling.
- Line 184: "sufficient for robust functional genomics screening" might be better replaced with something more descriptive that emphasizes the caveates. 10 sgRNAs/gene is sufficient to pick out auxotrophic genes in competitive growth over 10 doublings.
- Line 220: issues of biased coverage in Tn-mutagenesis do not occur in CRISPRi screens. I think it would be more accurate and clear to say that CRISPRi libraries can be designed to cover regions that Tn-mutagenesis hits at low frequency. The positive point is that sgRNA design with the parameters the authors have developed still allows for high coverage across the genome. The negative point is that design of sgRNAs and sgRNA efficacy are not equal, and the data in this manuscript cannot address that.
- Line 244: according to the methods the strains were cultivated to OD=1 which is not exponential phase.
- Line 262-3: should say "Among 313 genes determined to be essential during the construction of the Keio collection" or something to indicate that they are not in the Keio collection
- Line 299: The authors have wrote themselves into a difficult position in using the Keio essentials list both as a gold standard and then using their data to say that the Keio list has false negatives.

Response to the Reviewers:

Reviewer#1

Wang et al. provide a thorough, well-written case for pooled CRISPRi as a gene-phenotype mapping tool in prokaryotes. CRISPRi pooled screening is demonstrated to identify essential genes in *E. coli* (Figure 3), probe metabolic networks (Figure 5), identifying the phenotypes of non-coding RNAs (Figure 4), and identifying genes that confer tolerance to toxic chemicals (Figure 6). Compared with other methods used in prokaryotes, namely transposon sequencing, CRISPRi identifies fewer false positives and achieves a more uniform distribution of sgRNA across the chromosomal DNA without bias for longer genes. Moreover, pooled CRISPRi is a more transferable screening method compared with single gene knockout strategies. The authors demonstrate a wide range of uses for this tool and validate its effectiveness by identifying genes already known to be essential (while also identifying more that have not been studied.) The manuscript has been substantially improved over the previous submission and that the experimental logic is clear and the figures tell a clean story. Addition of the identification of ncRNAs is great, as it really demonstrates the advancement of the paper over previous work. A few points of confusion or critiques are detailed below. Overall, I found this paper to be of high-quality, clear, thorough and significant.

Major comments:

1. The insight that the best guide RNAs fall within the 5% of ORF is unsurprising based on the wealth of work reported. The relevance with respect to this screening method needs to be explained. Also, explain why multiple guide RNAs (10-15) are needed for targeting a single gene. Also, what are the actual rules the authors find from Fig 1b for designing their sgRNAs? Is it targeting the first 5% of the ORF, which doesn't seem particularly connected to the actual figure? Is it the number of sgRNAs required to call out successful hits (10)? Should be made more clear.

Response: we thank the reviewer for these suggestions. Two main conclusions extracted from the tiling library experiment are used in genome-wide library design. Firstly, as seen in Figure 1b, more sgRNAs provide stronger statistical power (smaller P value) and better statistical reliability (robustness against noise), although as less as 3 sgRNAs per gene is enough for hit calling. This conclusion makes us design 15 sgRNAs per gene to provide better hit-calling ability when facing phenotype of moderate readout. Secondly, our results in test phase also suggests that better sgRNA activities can significantly improve hit gene calling, elucidated by the comparison between 'position' based and random sgRNA sampling (Figure 1b). We hence identified that the first 5% of ORF is a good indicator of sgRNA activity and used this principle to maximize the activity of sgRNAs in genome-wide library design. To

make it clearer, we revised Figure 1c to highlight these two principles we used in genome-wide sgRNA library design. The related text is presented in the revised manuscript (Line 189-194).

“Based on the result that 10 sgRNAs/gene was sufficient for reliable hit-gene calling and that there was an increase in statistical significance if more sgRNAs were available, we chose a library size of 15 sgRNAs/gene to ensure robust performance for genes with moderate phenotypes. Moreover, the result of active sgRNA positioning resulted in our selection of as many sgRNAs as possible from within the first 5% of the ORF.”

(c) Genome-wide library:

Phenotype	Selective condition	Control condition	Related figure and/or dataset
Essentiality	dCas9, LB	empty plasmid, LB	Fig. 3, Fig. 4, Fig. S7-8, Fig. S10, Fig. S12, Table S6-8, Table S13-15
Auxotrophy	MOPS	LB	Fig. 5, Fig. 6, Fig. S7-9, Fig. S13-16, Table S6-7, Table S9
Amino acid addition	0.5 g/L casamino acid, MOPS	LB	Fig. 5, Fig. S7-8, Table S6-7, Table S10
Furfural tolerance	0.4 g/L furfural, MOPS	loginitial, see Fig. S7	Fig. 6, Fig. S7-8, Fig. S16, Table S6-7, Table S11, Table S16
Isobutanol tolerance	4 g/L isobutanol, MOPS	loginitial, see Fig. S7	Fig. 6, Fig. S7-8, Fig. S16, Table S6-7, Table S12, Table S16

2. The distribution of the number of guide RNAs across all the genes in the genome is not clear? Do the different numbers of guide RNAs per gene impact the experiments or introduce any bias?

Response: we thank the reviewer to raise these concerns. We presented the distribution of the number of sgRNAs across all genes in Supplementary Fig. 5. Additionally, according to our result of computational sampling of sgRNAs belonging to each gene for hit gene calling (Figure 1b), we found that for phenotypes with strong readout (such as auxotrophy in this case), even 3 sgRNAs per gene is enough for robust hit identification. This is also true when we focus on essential genes, where our CRISPRi method maintained similar performance when coping with genes with shorter coding regions (< 300 bp), which typically have less sgRNAs (Figure 3b, c). In principle, more sgRNAs provide better statistical reliability as stated above, which is especially useful when the phenotype under investigation exhibits only moderate readout (e.g. partial growth impairment). Taken together, when

phenotype is strong, the issue is not a problem. While the phenotype is moderate, loss of information to a certain extent is possible. Hence, we think more is always better if the cost of library preparation and handling is not a problem. Using multiple (d)Cas proteins (Cpf1 for example) with different PAM preferences and combining the data may be a potential solution to design more sgRNAs for genes with small coding region, as we stated in the Discussion section (Line 529-533).

3. What are their criteria for a negative control? There are 400 of them – is there some threshold for what counts as a non-target. 2,3 mismatches? Did the authors check PAMs? This needs more explaining in the text around line 143.

Response: we thank the reviewer to point this out. As stated in Methods, we designed negative control sgRNAs by searching random 23-mers (with the format N20NGG) with the proper GC content to select those without any potential off-target candidates as identified by SeqMap. Here a more stringent off-target threshold is applied that hits with ≤ 5 mismatches are considered as off-targets. To make it clearer, we describe this in Results section in the revised manuscript (Line 136-138).

“In this case, we designed negative control sgRNAs by searching random 23-mers (with the format N20NGG) with the proper GC content to select those without any potential off-target candidates (≤ 5 mismatches).”

4. Is there some correlation between the 22 essential genes the authors failed to identify in fig 3a and the “strength” of the sgRNAs used to perturb them? For instance, did they have less than the 15 desired sgRNAs inhibiting them? Confirming that this is not the case would actually strengthen the paper, as it would suggest there is no biasing effect arising from their sgRNA design. Also in figure 3 it must be clarified why CRISPRi growth phenotypes are generally weaker than the Keio collection counterparts.

Response: we thank the reviewer for these suggestions, which are really important. We checked the number of sgRNAs belonging to these 22 genes and found that most of them have more than 10 sgRNAs, while one gene (*ydfB*) has only one sgRNA. The distribution histogram is shown below and exhibits no significant difference compared with that of all protein-coding genes (Supplementary Fig. 5a) ($P = 0.649$ by two-tailed MWU test), suggesting no bias derived from sgRNA number is introduced here.

To highlight this, we revised Supplementary Table 14 (currently Table S3) to add one additional column presenting the number of sgRNAs for each of these 22 genes. Moreover, the manuscript is revised accordingly as below (Line 324-329).

“We firstly checked the number of sgRNAs belonging to these 22 genes and found that most of them have more than 10 sgRNAs (Supplementary Table 3). The distribution exhibits no significant difference compared with that of all protein-coding genes ($P = 0.649$ by two-tailed MW U test), suggesting no bias derived from sgRNA design is introduced when performing hit gene calling.”

In addition, we reason that the weaker phenotype found in CRISPRi experiments may be derived from the non-uniform distribution of sgRNA activities (Figure 1b, Supplementary Fig. 10). The phenotype score of a gene is calculated as the median of fitness scores of all its sgRNAs. The sgRNAs with poor activities give rise to residual expression of target gene, thereafter resulting in overall weaker phenotype in contrast to gene knockout, which is used by Keio collection to absolutely delete gene expression. According to the suggestions of reviewer III, the diversity of sgRNA activities is demonstrated experimentally at the single-cell level (Supplementary Fig. 22 as below, sgRNA_414 and 83 exhibit poorer activities compared with others). For details, please see response to reviewer III, point I.

We also talked about this issue in the revised manuscript (Line 257-263).

“The observed weaker phenotype found in our CRISPRi experiments in contrast to the lethal phenotype after essential gene knockout may be derived from the non-uniform distribution of sgRNA activities (Figure 1b, Supplementary Fig. 10). The phenotype score of a gene is calculated as the median of fitness scores of all its sgRNAs. The sgRNAs with poor activities give rise to residual expression of target gene, thereafter resulting in overall weaker phenotype in contrast to gene knockout method.”

5. Would the cluster strategy the authors introduce in line 202 causes a lack of resolution in identifying gene functionality? If highly similar sequences are treated as the same, wouldn't subtle differences in those genes be lost? I understand why this might be a necessary component of the method, given that the same sgRNAs might bind to sequences with similar composition, but I did not notice any discussion about potential downsides of this strategy. It is unclear how the cluster strategy enables the authors to study different tRNAs (line 344).

Response: we thank the reviewer for these important suggestions. We confess that the clustering strategy is somehow a compromise given the inherent specificity-programing logic of CRISPR/Cas9 system. For example, resolution is limited when applying this strategy to study genes with highly similar sequences (for example, genes with > 95% nucleotide identity, the threshold for clustering used in this work). However, we argue that this loss of information is minimal. In our recent experiment using a saturated sgRNA mutant library (covering all sgRNA mutants with one or two mutations, Wang et al. unpublished data), we found that in bacteria, two mismatches between DNA target and sgRNA are enough to completely abolish CRISPRi activity in most of the cases (see heatmap below for two representative sgRNAs), consistent with the previous report (Luke Gilbert et al. *Cell* 2014).

Figure We synthesized two saturated sgRNA libraries for two sgRNAs, covering all the possible single or double mutations of the N20 relevant sequence. Based on a cell-lethality-associated method, we profiled the CRISPRi activity of every member in these two libraries. Shown above is the mean activity of all mutants containing the two mutations at two defined positions. One cell is

the average of 9 (3×3) mutants, whose nucleotides at the two positions are both mutated. The activity of wild type is defined as 0.

This result suggests that even for genes with highly similar sequences, it is possible to study them individually by our method (2/20 base difference is equal to 90% sequence similarity on average). Meanwhile, clustering strategy enables us to knockdown genes with similar sequences (thus logically similar functions), which are hard to achieve by conventional methods in a large scale, as exemplified by tRNA and *tufA/B* in this work. In this line, we think clustering strategy can be very customized depending on the goal of research. If genes need to be assayed individually, just try to find the sgRNA distinguishing similar genes. On the opposite, clustering strategy makes it possible to investigate several genes as a functional group. The programmable CRISPRi system is suitable for both of these purposes. To make it more clearer, we talked about this issue a little bit in the revised manuscript (Line 312-320).

“We assume that by applying a more relaxed cutoff (>95% nucleotide identity in this work), it is possible to use cluster-level gene repression via CRISPRi screening to explore prokaryotic genetic interactions. Moreover, although loss of information by using clustering strategy is inevitable facing gene duplicates with identical sequences, considering the fact that two mismatches between DNA target and sgRNA are enough to abolish CRISPRi activity³⁰, it is still possible to study genes with highly similar (not completely identical) sequences individually. Hence, in practice, the threshold of clustering can be regarded as a customized parameter in CRISPRi screening to fulfill the requirements of research to use this method.”

The tRNA collection encoded by *E. coli* genome is an ideal case to test this idea. As shown below (extracted from GtRNAdb, Chan, P.P. & Lowe, T.M. (2016) Nucl. Acids Res.), the tRNA pool within a microorganism consists of various tRNA isoacceptor families (for example, tRNA-Arg-ACG). Each family has a different anticodon that decodes the corresponding codon. Each tRNA family has a single (e.g. tRNA-Arg-CCG) or multiple (e.g. tRNA-Arg-ACG) gene copies. Multiple copies are suggested as a solution to maintain tRNA availability in living cells. In addition, due to the conserved structure of tRNA to fit the ribosome, all tRNAs encoded by a particular organism share overall sequence similarity (e.g. highlighted regions below in alignment). All these issues lead to difficulty in studying tRNA fitness by conventional gene knockout or studying each gene individually by CRISPRi.

enables us to study the phenotype of tRNA at the isoacceptor family level, covering in total 32 isoacceptor families and 61 tRNA-coding genes (Figure 4b).”

6. (starting at line 267). It seems as though the authors label all genes in polycistronic operons that were deemed growth-impairing false positives if the operon contains a ‘downstream Keio essential gene’. The authors acknowledge that this false positive is an overestimation, but I am not sure how they can state that ‘ false positive genes are expected to truly impair cell growth to a less extent compared with lethality’, based on the fact that there is one Keio essential gene within the operon. There could still be other genes within the operon that are just as essential. I would think this information could be used to probe the essentiality of other genes with a polycistronic operon.

Response: we thank the reviewer for this comment. We actually intend to state the same opinion as the reviewer pointed out. We improved the manuscript as below to clarify this misleading statement (Line 268-271).

“It should be noted that this value is an overestimation, as there could still be other genes within the operon that are just essential. The essentiality of other hit genes within such polycistronic operons needs to be further investigated.”

7. (lines 313-321) One part of the results analysis that could be further developed concerns 22 genes that were found to be essential using single-gene knockout mutants (Keio analysis) but were not found to have any significant phenotype using CRISPRi. Are these genes considered false positives from the Keio collection (the paper does not label them as such), and if so, did the paper make any effort to identify false positives? How did they miss these? Moreover, Supplementary Table 14 reports that most of the genes are ‘not related to paramount biological processes’, and one example is given in the manuscript, but are there any of these 22 genes that have not been studied that may actually be essential? The end of the paragraph mentions alternative approaches to test essentiality of these genes, but more detail here would be welcome.

Response: we thank the reviewer for these constructive suggestions, which are important to further strengthen this work. These 22 genes are indeed labeled as essential gene candidates in Keio collection and its update (Baba et al. *Molecular Systems Biology*, 2006, Supplementary Table 6 and Yamamoto et al. *Molecular Systems Biology*, 2009, Table 1). Researchers constructing Keio collection used lambda Red system to delete every gene encoded by *E. coli* K12 genome. Failure to obtain knockout mutant (confirmed by PCR) was used as evidences to suggest essential gene candidate. Apparently, using such indicator faces risks of both false positive (e.g. locus-specific poor recombination activity, unexpected polar effect, etc) and false negative (e.g. gene duplication,

etc). However, as a genome-wide study, it is impossible to rule out all such possibilities for each gene due to the big amount of work to do.

We think CRISPRi pooled screening provides advantages to identify essential genes over arrayed gene knockout not only in terms of cost and speed as mentioned in the manuscript, but also because it avoid potential problems faced by gene knockout such as poor recombination activity and gene duplication. To confirm this and update Keio essential gene list, we recently tried a novel strategy to test the essentiality of 5 of these 22 genes. We reasoned that the failure to obtain knockout colonies for these genes might be due to the polar effect of the introduced kanamycin marker or large deletion used in Keio collection construction, or possibly the recombination efficiency issue. To minimize polar effect, we intended to delete only 2 bp within the ORF to introduce frame shift mutation by lambda Red system. To boost the recombination efficiency, we used CRISPR/Cas9 system (Jiang et al. *Applied Environmental Microbiology*, 2015) to introduce double strand break within ORF. We designed oligonucleotides carrying the 2 bp indel as recombination donor DNA. The 2 bp indel is designed to delete the guanine dimer in PAM region, which is absolutely required by CRISPR/Cas9 system. In such a way, CRISPR/Cas9 can be used as selection pressure to enrich for the knockout mutant. Among the five genes (currently Supplementary Table 3) we tested, we succeeded to obtain indel mutants for 3 of them (*yhhQ*, *gpsA* and *chpS*) (Supplementary Fig. 14). In fact, via searching the literature, we found a recent report stating *yhhQ* can be deleted (Zallot, et al. *Biomolecules* 7, 12 (2017).) while failed to find such evidences for *gpsA* and *chpS*. We think comprehensive description of these 22 genes may be beyond the scope of this work, which is to present the usage of CRISPRi as screening method. Taken together, these results suggest that firstly, our CRISPRi method is reliable that it can capture the false positive information in other arrayed gene knockout library, update the essential gene list for model microorganism *E. coli*; and secondly, complementary approaches (such as CRISPR/Cas9 facilitated recombineering here) are needed for genome-wide microbial genetics studies. We revised the manuscript in relevant paragraph as below (Results: Line 335-343; Method: Line 668-678)

“To test this, we used CRISPR/Cas9 facilitated recombination (see Methods) to introduce 2 bp indel frameshift mutations to five (*chpS*, *folk*, *gpsA*, *grpE* and *yhhQ*) of these 22 false negative genes. Colony PCR confirmed that we obtained mutant strains successfully for three of them (*chpS*, *gpsA* and *yhhQ*) (Supplementary Fig. 14). These results suggest that firstly, our CRISPRi method is reliable to capture the false positive information in other arrayed gene knockout library, enabling updating the essential gene list for model microorganism such as *E. coli*; and secondly, complementary approaches (such as CRISPR/Cas9 facilitated recombineering here) are needed for genome-wide microbial genetics studies.”

“Indel mutation introduction by CRISPR/Cas9 recombineering We used CRISPR/Cas9 recombineering reported by Jiang et al⁶³ to introduce indel mutations. The relevant sgRNAs (Supplementary Table 6) were designed to target the first 200 bp of the coding region belong to target genes. Oligonucleotides as recombination DNA donor were designed by MODEST server⁶⁸ (Supplementary Table 7) to delete the guanine dimer (‘GG’) in DNA PAM region targeted by the corresponding sgRNA. Thus the introduced 2 bp indel is enriched by eliminating the lethal DNA cleavage via Cas9 nuclease and leads to the frameshift mutation of relevant gene. Primers within the ORF flanking the indel were designed (Supplementary Table 7) to amplify the target region from survival colonies and the PCR products were subjected for Sanger sequencing.”

Reviewer #3 (Remarks to the Author):

NCOMMS-17-00870A-Z

Summary

The authors present a significantly more developed manuscript than we previously reviewed, and have addressed the majority of our previous concerns regarding the former manuscript. The new manuscript presents pooled, tiling and genome-wide CRISPRi libraries in *Escherichia coli* and uses them to: (1) determine rules for targeting ORFs within operons, (2) assess gene essentiality

on a genome-wide scale by identifying extreme fitness defects, and (3) explore tolerance to industrial chemicals by identifying increases in fitness.

The major contributions of this paper are as follows: systematic demonstration that targeting the first 5% of an ORF results in the strongest phenotype, demonstration that genome-wide CRISPRi outperforms Tn-seq in essential gene identification on some important parameters (i.e. identifying small essential regions), present a new conceptual approach to targeting “clusters” of genes and apply this to investigate tRNA redundancies.

Improvements since last revision

As we and the other reviewer requested, the manuscript is clearer with regard to the mechanism of CRISPRi and how targeting genes within operons may be interpreted. With a few exceptions (see below) this has been addressed.

Should be improved

The vast amount of new data added to the new manuscript does suggest that generally the authors' CRISPRi system is working. However, this does not fully address our concerns about presenting a thorough characterization of the CRISPRi system, which is important for interpreting all the presented screening data. Here are the points that need to be addressed:

1. sgRNAs cause depletion of the targeted mRNA and/or protein in the cell, and targeting a gene for knockdown has a similar effect as a gene deletion. For quantifying and demonstrating knockdown, a single-cell assay would be the most convincing. Bulk fluorescence (as shown Fig. S18) is less informative without a growth curve or some demonstration that an equal number of cells, in an equivalent physiological state, are being compared. Presenting clearly how much variability in knockdown is important for readers to understand sources of noise in this screen, and to be able to rationally design improvements or variations for other purposes. In particular, more not-pooled experiments would clarify some of the effects from targeting within operons. For example the “false positive” result in Supplementary Fig. 3: does targeting *glnL* actually increase the amount of *glnA* mRNA and/or protein?

Response: we thank the reviewer for these insightful suggestions. We performed CRISPRi experiment at the single-cell level by repressing sfGFP expression with 6 different sgRNAs and subjected to flow cytometry analysis (Supplementary Fig. 22). Importantly, we observed sustainable sfGFP depletion across the population at the single-cell level and no skewness or increase of expression variability compared with the control group (see below). Interestingly, diversity of sgRNA activity (average of repression fold) is also found here. We supplemented two

new paragraphs in the Methods section of the revised manuscript to describe this experiment (Line 656-666). One of them is attached below.

“The result (Supplementary Fig. 22) shows that the cell population with repressed sfGFP expression exhibits log normal distribution of sfGFP fluorescence. Weak skewness is observed for all sgRNAs tested here, suggesting a sustainable gene knockdown efficacy across the cell population at the single-cell level. Moreover, the noise of sfGFP expression after CRISPRi treatment at single-cell level is comparable or even smaller than that of positive control group with a non-targeting sgRNA. Hence, we deduce that CRISPRi machinery does not lead to increased expression noise at the single-cell level, which is important for reliable CRISPRi based screening in a pooled format. It is also worthy noting that the diversity of sgRNA repression efficiency is observed here (as much as 10-fold, sgRNA_166 vs. sgRNA_414), which is consistent with our conclusions at the functional level (Figure 1b, Supplementary Fig. 10).”

For the concerns about the operon issue, please see response of point 6.

2. A comparison of CRISPRi mutants to deletion mutants (outside of the screening context) is also necessary. Something like what is presented in response to Reviewer #1 question #6 would be helpful for the reader.

Response: we thank the reviewer for this suggestion. We revised the manuscript accordingly as below (Line 636-639).

“In addition, we also used the CRISPRi system developed in this work to screen for some other genetic targets related to terpenoid production. Indeed, we observed many positive hits and confirmed our results by gene knockout (Wang et al. unpublished data). We believe these data is sufficiently enough to prove the reliability of our system.”

3. A clearer explanation of their pooled growth experiments is needed: the explanation in the Methods section (Lines 705-717) is unclear... is the number of

doublings or the sequencing depth limiting resolution? If the number of doublings is the major limitation, is there a rationale for using only 5 doublings? Further, in these experiments fitness is calculated as a ratio between a selective and control condition. Eg. for the essentials, +/-dCas9. Could the authors comment on this approach compared to a more standard fitness calculation used in the field, looking at abundance over time (gamma, etc)?

Response: we thank the reviewer for this useful suggestion. The resolution limit is dependent on the ability to detect a cell exhibiting no division during the process after screening experiment. Here, we assume that the bulk population has the same doubling rate with that of wild type cell. Hence, the growth of the bulk population leads to the dilution of cell with impaired growth or lethality. The factor limiting experiment resolution can be expressed as following:

A = $\log_2(\text{sequencing depth})$, typically 6 assuming 10 M reads (limited by current NGS capacity) vs. 100 K library (20 sgRNA per gene plus 5 K genes for a bacterial genome);

B = cell doublings during screening;

Resolution limit = $\min(A, B)$

In both cases of essential gene identification and other experiments, $A \sim \log_2(100) = 6.6$

In the case of essential gene identification, the number of initial living cells of incubation is determined by electroporation efficiency, around 10^5 colonies per well \times 30 wells = 3×10^6 cells. Cultivation to OD600 \sim 1 results in final cell number around 10^9 cells/mL \times 100 mL = 10^{11} cells. Here, the number of doublings (B) equal to $\log_2(10^{11}/(3 \times 10^6)) = 15$

In the case of other experiments, simply 5 doublings (B) are maintained (OD₆₀₀ from 0.03 to 1).

Shown above is the calculation of experiment resolution in different experimental settings. The rationale to allow only 5 doublings is to reduce the risk of *de novo* mutation in bacterial genome, which may influence cell growth but cannot be detected by sequencing only the variable region of sgRNA in this work, thus leading to noise in the final data. It is worthy noted that 5 doublings is pretty good for robust hit calling for auxotrophic genes (Supplementary Fig. 16, AUC-ROC = 0.9). In principle, 5 doublings can amplify the fitness advantage of 15% by 2-fold. ($1.15^5 \sim 2$), which is sufficient for NGS to detect. Actually, we

allowed for 10 doublings in another experiment for comparison and observed no significant differences. To make this issue clearer, we revised the manuscript as following (Line 802-826).

“5 doublings during screening experiments were designed to reduce the risk of undetected *de novo* mutation in the bacterial genome, which might influence growth and lead to noise in final data. Meanwhile, 5 doublings is reliable for hit calling of auxotrophic genes (Supplementary Fig. 16, AUC-ROC = 0.9). In principle, 5 doublings can amplify the fitness advantage of 15% by 2-fold ($1.15^5 \sim 2$), which is sufficient for NGS to detect. Actually, pre-experiments suggested no significant differences between 5 and 10 doublings screening in our case.

The resolution limit of our method is dependent on the ability to detect a cell exhibiting no division during the process after screening experiment. Here, we assume that the bulk population has the same doubling rate with that of the wild type cell. Hence, the growth of the bulk population leads to the dilution of cell with impaired growth or lethality. The factor limiting experiment resolution can be expressed as following equations:

Resolution limit = $\min(\log_2(\text{sequencing depth}), \text{cell doublings during screening})$

Generally, the median read count of one sgRNA in the genome-scale sgRNA library was ~ 100 (Supplementary Fig. 6c). Considering the number of doublings was ~ 15 for essential gene identification, we reasoned that the resolution of our method for gene dropout screenings is approximately 6 to 7 in essential gene search ($2^6 < 100 < 2^7$). For example, in the essential gene screening experiment, we used a fitness threshold (≤ -6) to infer gene essentiality, whose knockdown led to no cell division. We can improve this resolution by increasing the sequencing capacity applied to each NGS library (currently 10 million reads per library). In the following experiments (auxotrophy, casamino acid addition and chemical tolerance), because we only allowed the cells to grow for about five doubling times, the limiting factor controlling the resolution—in these cases, a fitness level of about -5 is expected for cells carrying a gene knockdown without any growth—is the doubling number rather than the sequencing depth of the NGS libraries.”

Looking at mutant abundance over time is indeed a more standard treatment. However, it is no suitable in all cases, such as the essential gene identification experimental setting in this work. Due to the constitutive expression of dCas9 (aiming to reduce noise), CRISPRi starts to repress gene expression once the sgRNA plasmid is transformed. Hence, t_0 must be set to the very time immediately after transformation rather than the plasmid library before transformation (actually many previous papers made this mistake in our opinion by using the plasmid format as t_0), because the transformation is one of the bottleneck steps, which can reshape the sgRNA relative abundance profile, especially for those with low initial abundance. However, it is very hard to do so, as only around 3×10^6 successful transformed cells are available for each sample by then and the majority of them needs to be subjected to the following experiments to maintain library coverage. To address this, we used a negative control condition with no dCas9 expression in essential

gene identification experiment, where in principle all mutants exhibit the same growth profile. It is noted that if the same doubling numbers are maintained for each sample, we think that our data processing method is proper. In contrast, if the same cultivation time is maintained for each sample, methods described by Kampmann et al. (PNAS 2013, developed for RNAi screening) is well established.

Once the cell population containing the library is established (such as the other experiments in this work except for essential gene identification), we think there should be no big difference between these two data processing methods, if and only if the negative control condition satisfies two terms that firstly, all mutants present the same growth profile and secondly, the control condition should be similar to that of the selective condition. In this work (Figure 1c), we used LB dCas9(+) as negative control condition for auxotrophic gene experiments to make it consistent with essential gene experiment. We also measured directly the mutant abundance change over time for toxin tolerance experiments, because if similar MOPS medium is used here as control, many (auxotrophic) mutants present perturbed growth. It is noted that using different reference (control) condition may results in different results. However, the biological insight can be extracted combining results using different reference states (see Response to Point 8, indole and toxin tolerance).

4. The auxotrophy experiments are very nice, but a few questions remain, in particular around the “false negatives”: genes whose deletions render them unable to grow in minimal media but whose knockdown by CRISPRi does not affect fitness. For those genes, what does the distribution of sgRNA phenotypes look like (i.e. is it unimodal)? Is it the case that some sgRNAs do actually “work” and some do not (i.e. does Fig1b look different for genes that work and genes that don’t work)? Or is the assertion that targeting this gene with any sgRNA does not cause a phenotype? If the latter, could the authors demonstrate that a deletion strain does/does not have a fitness defect in these conditions? Is knockdown happening for those targets in the screen conditions (qPCR, WB)?

Response: we thank the reviewer for these insightful suggestions, which are very important to further strengthen this work. Because the paper describing Keio collection does not give an exact definition for auxotrophic genes in MOPS medium, we hence checked the OD₆₀₀ distribution of Keio collection in MOPS medium (as below) and chose OD₆₀₀<0.09 as threshold to define auxotroph.

Meanwhile, $FDR > 0.05$ or median Z score > 0 is used to define gene knockdown without significant growth impairment in MOPS medium. Combining these two set together, we identified 16 false negative genes in genome-wide CRISPRi screening for auxotroph. The distribution of sgRNA fitness exhibits unimodal distribution (with average around 0), the same as genes that work (see below for representative examples, *trpABC* is used as true positive example).

Figure Each panel describes the distribution of sgRNA fitness scores of a particular gene. The panels colored green are representative genes found to be false negative in auxotroph screening, while the yellow ones are true positive ones identified in either Keio collection and our dataset.

We think this phenomenon is not related to poor sgRNA activities. As observed in Supplementary Fig. 10, more than 90% sgRNAs in the

library should be highly active. Moreover, according to our recent work using Cas9 (rather than dCas9 in this work) to probe sgRNA activities in the same *E. coli* host (Guo and Wang et al., bioRxiv: <https://doi.org/10.1101/272377>), the majority of sgRNAs belonging to these 16 genes presents strong activities (see below). Although we cannot simply regard sgRNA activity of CRISPRi and CRISPR/Cas9 DNA cleavage as the same, this result still suggests indirectly that these sites are accessible for CRISPR/(d)Cas9 complex, a necessary condition for CRISPRi to work.

Hence, it is reasonable to suggest that targeting these 16 genes with any sgRNA does not cause a phenotype in our screening experiments. It can be deduced to the pooled format of CRISPRi screening, where the cross-feeding of essential nutrient is possible, which is demonstrated in our previous paper (bioRxiv: doi: <https://doi.org/10.1101/129668>). Another possibility is that the stock of nutrient (such as cofactor biotin) in the cell from seed culture is enough for cells to growth for 5 doublings without any supplementation. For example, biotin biosynthesis is significantly enriched in these 16 genes (5/16), where *bioH* is found to be a false negative in either tiling library screening (Figure 1b) or genome-wide library experiment. Active pumps in *E. coli* for biotin uptake (*bioP*) are known. Moreover, prolonged cultivation time (10 doublings) results in the identification of these genes as auxotrophic (see Table below). We think more careful comparison between different conditions (e.g. doublings) may bring more biological insight about the bacterial physiology. However, it is beyond the scope of this work. We hope to perform more comprehensive and focused analysis in the following work.

gene	Fitness-5d	FDR-5d	Fitness-10d	FDR-10d
bioB	-0.78	0.056	-1.60	0.004
bioC	-0.57	0.156	-1.62	0.011
bioD	-0.77	0.124	-1.76	0.008
bioF	-0.72	0.092	-1.71	0.004
bioH	-0.89	0.272	-1.38	0.219

To meet the paper length requirement and avoid overwhelming details, we added a supplementary note in SI to talk about this issue. It is mentioned at Line 421-424 of main text.

“To better understand this method, we studied the reason for false negative hits in this experiment (Supplementary Note I). Briefly, we suggest that some experimental parameter settings and pooled format of screening, rather than sgRNA activity issue, should be responsible to these false negatives.”

5. We look forward to more data (Line 320-21) on the 22 genes determined to be essential by the CRISPRi method but not by deletion (although please see update to the Keio above). In particular does complementation with the gene product allow deletion? It may be a good time to note that the genetic backgrounds of the Keio collection and the strain used in this manuscript are not identical.

Response: we thank the reviewer for these useful suggestions. Actually the essential gene list used in this work is adopted from original Keio collection report (Baba et al. *Molecular Systems Biology*, 2006, Supplementary Table 6) and its update (Yamamoto et al. *Molecular Systems Biology*, 2009, Table 1). We mentioned in the revised manuscript the difference between the two strains used in Keio collection (K12 BW25113) and this work (K12 MG1655) (Line 300-303). We think the minor genetic differences between these two strains results in little influence on the comparison result (figure below is adopted from Baba et al. *Molecular Systems Biology*, (2006)).

“Note that the strain used by Keio collection¹ and the Tn-seq study referred here²⁵ is *E. coli* K12 BW25113, while K12 MG1655 strain is used in this work. The genetic differences between these two strains are minimal¹, which does not influence the comparison performed here.”

Rather than gene deletion with gene product complementation, we recently tried a different but more direct strategy to test the essentiality of 5 of these genes and succeeded in 3 of them. For details, please see response to reviewer I, point 7.

6. A rationale for forward and reverse polarity is presented in FigS11, but does the data support this? Do the authors observe any instances of “reverse” polarity, where targeting a downstream non-essential gene affects expression of an upstream essential gene? I think this will be an interesting question for readers and the authors have a lot of data to bring to bear on it.

Response: we thank the reviewer for this important suggestion. Note that Supplementary Fig. 11 only shows the rationale for forward polarity when using CRISPRi, which is intuitive considering the working mechanism of CRISPRi. We think this issue (forward polarity) contributes mostly to the failure of this method to provide single-gene resolution in polycistronic-mRNA-transcribing operons, which is discussed in our response to point 7. Additionally, Peters et al (*Cell* 2016, Supplementary Fig. 1E) for the first time observed the “reverse polarity” using an artificial construct (*rfp-gfp*), that repressing downstream gene by CRISPRi perturbs (knockdown) the expression of upstream gene in operon. This cannot be simply deduced to the current molecular model of CRISPRi, hence worthy further study. We investigated this issue using the big data in this work. Briefly, we extracted all genes downstream of a known Keio essential gene organized in one operon and excluded those with significant phenotypes (FDR < 0.05 and median phenotype < 0). Our hypothesis is that these genes exhibit no growth phenotype (average of sgRNAs), hence the phenotypes of some sgRNAs are derived from the “reverse” polarity (repressing upstream essential genes). Interestingly, the 54 genes we collected present a position-dependent “reverse” polarity (see below).

We observed strong phenotypes for sgRNAs within the first 50 bp in the coding regions of these genes. As the distance from the upstream essential genes gets longer, such effect becomes weaker until no difference can be found between them and negative control sgRNAs. Moreover, in spite of the “reverse” polarity within a 50 bp window, the phenotypes (perhaps the expression perturbation) caused by such effect are significantly weaker than directly targeting dCas9-sgRNA complex to the essential genes. This is also consistent with previous observations (Peters et al *Cell* 2016) that “forward” beats “reverse” in terms of strength. We also checked the overlaps between these 54 genes with their neighbor essential genes. We found only 3 of them have 3, 6 and 7 bp overlap between ORFs with the upstream counterparts, thus ruling out the impact of overlaps on this analysis. We think the “reverse” polarity of CRISPRi may suggest some unknown aspects of its working mechanisms. For example, some recent papers (Connell et al (2014) *Nature*, 516, 263–266; Nelles et al (2016) *Cell*, 165, 488–496; Liu et al (2016) *Sci. Rep.*, 6, 29652) suggest CRISPRi complex also binds RNA besides conventional DNA substrate. “Reverse” polarity may hence be due to the perturbation to the translation process (e.g. CRISPRi blocks the access of ribosome to the stop codon of upstream gene). Further investigations are needed to shed light on the molecular mechanism. We revised the manuscript accordingly (Line 273-280) as below.

“We also checked the reverse polarity of CRISPRi using our data, which was firstly described by Peters et al¹⁷ that repressing the downstream gene by CRISPRi perturbs the expression of its upstream counterpart in one operon. Consistent with previous reports, our result (Supplementary Fig. 12) suggests a position-dependent reverse polarity effect, which is within the window of the first 50 bp of downstream genes. This observation cannot be simply explained by the current model of CRISPRi to shutdown RNAP transcription, indicating more works are needed to understand the mechanism of CRISPRi better at the molecular level.”

7. The potential for single-gene information: the authors assert that the risk of conflating the phenotypes of all genes in an operon is “moderate” (in the letter response) based on the low percentage of false positives (fitness defect for non-essential knockdown) attributable to an essential gene in the operon. I think this argument is not a good one, because it may be rare for essential genes to be in operons with non-essential genes, but for other phenotypes it is more likely that genes will be grouped by function. Therefore the knockdowns at the operon level will almost certainly conflate the phenotypes of all member genes in a way that will require orthogonal approaches to disentangle. Caution around falsely promising single-gene phenotype information from CRISPRi is important.

Response: we thank the reviewer for this critical comment. Indeed, the inherent working mechanism of CRISPRi limits its resolution when multiple genes are organized in one operon. We accordingly revised the relevant text in the manuscript (Line 265-273). In Discussion section, this issue is also carefully addressed and recently reported RNA-targeting-CRISPRi tools are suggested there to be potent to solve this problem (Line 525-532).

“we checked the operon structure of all growth-impairing genes identified via CRISPRi and tagged those with a downstream Keio essential gene(s) in one polycistronic operon as a false positive (Figure 3a), which was the case for 8.0% of all hit genes exhibiting impaired growth. It should be noted that this value is an overestimation, as there could still be other genes within the operon that are just essential. Even though, the essentiality of other hit genes within such polycistronic operons needs to be further investigated by other orthogonal methods. This result suggests that polycistronic operons result in only moderate interference with our method at the genome level, notwithstanding that cautions are still needed to cope with polycistronic operons via CRISPRi method.”

“The potential limitations of this method (which are indeed shared with Tn-seq to a certain extent) include, first, possible false positives occurring for genes transcribed as polycistronic mRNAs because of the polar effect of CRISPRi and, second, ... We suggest here to solve the first problem by either designing a better algorithm for hit-gene calling or adopting the recently described RNA-targeting CRISPR/Cas system⁵³ to modulate gene expression at the RNA level.”

8. While this manuscript includes many thoughtful comparisons to gold standards in order to validate their methodology, there is little discussion of any new biological findings even in the newer screening conditions. In the cases of the isobutanol and furfural screens, could the authors provide some explanation for why many hits overlap (i.e. any known mechanisms of toxicities for these chemicals?), and if this was known. Were any interesting sensitivities found? It seems that ncRNAs with significant phenotypes are highlighted, but do the gene phenotypes fit with what is known or provide any new insight?

Response: we thank the reviewer for this comment. In fact, both furfural and isobutanol are regarded as organic-solvent-like chemicals, which have a similar stress response at the transcriptome level reported in *E. coli* (Rau et al. *Microb. Cell Fact.* (2016)). This is probably the reason for the hit overlap of these two conditions in our experiments. Currently, the detoxification or tolerance mechanism towards these two chemicals is known to be a systems-level response, where many different pathways work together to maintain the cellular viability. Some molecular contributors are already known, such as the NADH-dependent reduction of furfural in *E. coli* (Wang et al. *Proc. Natl. Acad. Sci.* (2013)), the reprogramming of cell envelop for (iso)butanol in bacteria (Kanno et al. *Appl. Environ. Microbiol.* (2013)) and the benefit of strengthening the potassium and proton electrochemical membrane gradients for isobutanol in yeast (Lam et al. *Science* (2014)), etc. We believe these reports only reflect the tip of iceberg of potential mechanism considering the large-scale gene expression reprogramming of bacteria facing these chemicals. Indeed, our data identified some known targets with tolerance for these stresses related to these processes (Supplementary Table 5).

Additionally, we identified two novel interesting targets, which are now under further investigation in our lab. The first one is identified by GO analysis: the role of indole molecule for these two stresses. Indole is known to modulate persistence towards aminoglycoside antibiotics in spite of contradictory results of its effect (induction or reduction) (Vega et al. *Nat. Chem. Biol.*, (2012) and Hu et al. *Environ. Microbiol.* (2015)). Indole is converted from tryptophan (reversible reaction) via lysase encoded by *tnaA*, whose knockdown here results in change of tolerance facing either furfural or isobutanol. Such effect is also observed for indole transporter, TnaB and tryptophan biosynthesis genes (see below). Note that using difference control conditions result in different results (because these genes exhibit phenotypes cultivated in MOPS minimal media, which is a control state) in spite of a big fraction of them showing significantly perturbed responses in both control conditions. If the performance of tolerance at industrial conditions is of concern, the absolute abundance change over time (initial state as control) should be the focus (this result is presented in the manuscript). In contrast, the results using MOPS media as control suggests the modulation of persistence in these stressed conditions if repressing these genes. Due to the complexity of indole signaling, this is still under investigation in our lab. We hope to provide more insightful results in the following works. Based on previous reports and the results here, we anticipate that indole is

a general signaling molecule for bacteria to cope with diverse environmental stresses.

Gene	furfural vs. initial		furfural vs. MOPS		isobutanol vs. initial		isobutanol vs. MOPS	
	Phenotype score	FDR	Phenotype score	FDR	Phenotype score	FDR	Phenotype score	FDR
tnaA	2.845	0.002	1.114	0.004	0.678	0.019	-0.752	0.052
tnaB	2.013	0.002	0.580	0.037	0.871	0.095	-1.230	0.148
trpA	-3.000	0.002	-1.011	0.123	-1.014	0.004	1.672	0.084
trpB	-3.057	0.002	UD	UD	-0.839	0.004	UD	UD
trpC	-2.886	0.002	0.393	0.311	-1.083	0.004	2.334	0.020
trpD	-2.948	0.002	0.425	0.368	-0.804	0.004	1.814	0.067
trpE	-2.776	0.002	0.349	0.259	-0.791	0.004	2.082	0.008
trpR	1.725	0.191	0.305	0.272	-0.245	0.306	-0.187	0.482

tnaA: convert Trp to indole (reversible); *tnaB*: indole transport; *trpABCDE*: Trp biosynthesis; *trpR*: Trp-induced *trpABCDE* repressor
UD: undetected due to the dilution of sgRNAs in control condition

Another interesting target is *pcnB*, which encodes a Poly(A) polymerase I responsible for the polyadenylation of 3' ends of RNA molecules, thus reshaping the stability (lifetime) of bacterial transcriptome. We observed strong tolerance towards these two chemicals if the knockdown of *pcnB* was executed by CRISPRi. This is consistent with our recent work (Liu et al, *ACS Synthetic Biology*, (2017)) that the mutations in this gene contribute to the Trp production of engineered strain at the stressed conditions. We suggest that this gene is also a global modulator for bacterial stress response and may serve as a novel target for engineering.

The main concern to highlight the ncRNAs with significant phenotypes here is that we think this is a methodology paper and ncRNAs is more rarely explored for stress response study by other methods. We confess that we are not the experts in bacterial RNA biology. After literature search, we found that due to the lack of prior knowledge in this field, it is hard to associate our discoveries with the known facts of these ncRNAs except for *rdlD*, encoding an antitoxin, which might be explained by the dormancy-stress-tolerance theory. We hope to deliver the dataset found here to the microbiology community for further mechanical discoveries.

To highlight the role of indole and *pcnB* in solvent tolerance, we revised the manuscript as below (Line 471-485).

“Additionally, we identified by GO analysis the important role of indole molecule for these two stresses. Indole is known to modulate persistence towards aminoglycoside antibiotics in spite of contradictory results of its effect (induction⁵⁰ or reduction⁵¹). Indole is converted from tryptophan (reversible reaction) via lysase encoded by *tnaA*, whose knockdown here results in change of tolerance facing either furfural or isobutanol. Such effect is also observed for indole transporter, TnaB and tryptophan biosynthesis genes. Based on previous reports and the results here, we anticipate that indole is a general signaling molecule for bacteria to cope with diverse environmental stresses, whose molecular mechanism needs further study. Another interesting target is *pcnB*, which encodes a Poly(A) polymerase I responsible for the polyadenylation of 3' ends of RNA molecules, thus reshaping the stability (lifetime) of bacterial

transcriptome. We observed strong tolerance towards these two chemicals upon the knockdown of *pcnB*. This is consistent with our recent work⁵² that the mutations in this gene contribute to the Trp production of engineered strain at the stressed conditions. We suggest that this gene is also a global regulator for bacterial stress response and may serve as a novel target for engineering effort.”

Small concerns

The set of essential genes from the construction of the Keio collection (Baba et al. 2006) should also include reference to the update to the Keio collection (Yamamoto et al. 2009) in which some designations of essentiality are changed.

Response: we thank the reviewer for these useful suggestions. Actually the essential gene list used in this work is adopted from original Keio collection report (Baba et al. *Molecular Systems Biology*, 2006, Supplementary Table 6) and its update (Yamamoto et al. *Molecular Systems Biology*, 2009, Table 1), although we only cited the original paper in our manuscript. To clarify it, we revised the manuscript to include the reference of updated Keio collection (Line 250-252).

“We first investigated the protein-coding genes that, when repressed, caused a reduction in the number of *E. coli* cells over the course of CRISPRi screening in LB broth and compared the result with the essential gene set of the Keio collection^{1,27} (Figure 3a).”

- Line 45 “multiples conditions simultaneously” is not what the authors present. Most likely they mean to say in parallel or rapidly.

Response: we thank the reviewer for the comment. The manuscript is revised accordingly (Line 40).

- Lines 80-82: the advantage is not that plasmids can be delivered without bias (because they are surely unequal in proportion) but that the targets can be designed without bias for long genes.

Response: we thank the reviewer for this useful suggestion. The manuscript is revised accordingly (Line 74-76).

“Finally, compared with Tn-seq, the sgRNA library can be designed uniformly across the bacterial chromosome without bias towards longer genes, thus maintaining identical statistical robustness for genes with short coding regions.”

- Line 97: the authors do not demonstrate here that their positioning actually matters for expression level (would need qPCR, WB etc.)

Response: we thank the reviewer for these useful suggestions. The manuscript is revised accordingly (Line 90-92).

“we indirectly (at the functional level) learned the activity positioning of sgRNAs in coding regions where CRISPRi maximally changes the expression of endogenous genes”

- Line 180: A clearer way to describe this approach might be “sampling” instead of “subsampling”, as the samples are not intended to reflect the larger distribution as they might when actually subsampling.

Response: we thank the reviewer for the useful comment. The manuscript is revised accordingly across the manuscript and supplementary materials.

- Line 184: “sufficient for robust functional genomics screening” might be better replaced with something more descriptive that emphasizes the caveates. 10 sgRNAs/gene is sufficient to pick out auxotrophic genes in competitive growth over 10 doublings.

Response: we thank the reviewer for this important suggestion. The manuscript is revised accordingly (Line 178-179).

“we determined that 10 sgRNAs/gene is sufficient to pick out auxotrophic genes in competitive growth over 10 doublings. (Figure 1b, right panel).”

- Line 220: issues of biased coverage in Tn-mutagenesis do not occur in CRISPRi screens. I think it would be more accurate and clear to say that CRISPRi libraries can be designed to cover regions that Tn-mutagenesis hits at low frequency. The positive point is that sgRNA design with the parameters the authors have developed still allows for high coverage across the genome. The negative point is that design of sgRNAs and sgRNA efficacy are not equal, and the data in this manuscript cannot address that.

Response: we thank the reviewer for these useful suggestions. The manuscript is revised accordingly (Line 213-215). Besides, we think that two points are worthy noted. Firstly, although the sgRNA efficacy cannot be regarded as equal, our data does confirm that most of them are quite active at functional level (Figure 1b, Supplementary Fig. 10) and at expression level (Supplementary Fig. 21, 22). Hence, we think the design of sgRNA libray in this work should present a more unbiased pattern in contrast to that of Tn-seq, considering multiple sgRNAs are designed for each gene. Secondly, we are currently developing machine-learning models to predict highly active sgRNA working in CRISPRi using the data in this work. It will be integrated into our sgRNA design pipeline, which can address the concern of sgRNA efficacy.

“Moreover, we noted the absence of transposon insertions within several chromosomal regions of up to 10 kb, whereas CRISPRi libraries can be designed to cover such regions that Tn-mutagenesis hits at low frequency (Figure 2b).”

- Line 244: according to the methods the strains were cultivated to OD=1 which is not exponential phase.

Response: we thank the reviewer for this important suggestion. The manuscript is revised accordingly across the manuscript.

- Line 262-3: should say “Among 313 genes determined to be essential during the construction of the Keio collection” or something to indicate that they are not in the Keio collection

Response: we thank the reviewer for this important suggestion. The manuscript is revised accordingly (Line 252-254).

“Among 313 genes determined to be essential gene candidates during the construction of the Keio collection (Supplementary Data 11, derived from the EcoCyc database)”

- Line 299: The authors have wrote themselves into a difficult position in using the Keio essentials list both as a gold standard and then using their data to say that the Keio list has false negatives.

Response: we thank the reviewer for this important suggestion. As a widely recognized high-quality dataset, Keio collection is regarded as the gold standard of *E. coli* genetics study since its establishment. The aim for us to use this set as a gold standard here is to benchmark the performances between different methods (namely CRISPRi, Tn-seq and genetic footprinting). This does not rule out the possibility that incorrect data points exist in Keio collection derived essential gene list. Actually, a high-quality set with the majority of its data as correct can just fulfill the requirement to be used as gold standard. Essential gene list from Keio collection definitely is such a set, in spite of a small fraction of incorrect entries in it, as we suggested as false negatives. In our opinion, there are no conflicts between these two points.

REVIEWERS' COMMENTS:

Reviewer #1 (Remarks to the Author):

The authors have satisfied this reviewers comments.

[Editorial note: The reviewer provided additional comments to pass along to the authors]

Reviewer 1-1

- The "Position" versus "Random" sgRNAs could be explained better. I'm assuming "Position" means in the first 5% of the gene, while "Random" means anywhere throughout the gene.
- "Amino acid addition" is not a phenotype, but the media they tested in.
- change "loginitial" to "log initial". What they mean by that is also confusing.
- The explanation for choosing 15 sgRNAs is fine and makes sense.

Reviewer 1-2

- I agree with their explanation largely. The only problem I have is that this suggests an inherent bias against identifying shorter genes related to a particular phenotype as the authors indicate. While I agree for certain phenotypes such as essentiality this bias will be miniscule, it may pose a greater problem for more difficult to parse phenotypes such as biofuel tolerance. I don't think this detracts from the novelty of this work, but I think the claim in lines 74-76 regarding no bias towards longer genes is misleading. The authors should change "without bias" to "minimal bias" or something along those lines. The authors should also briefly touch upon this limitation in the paragraph mentioned (lines 525-536), as it is currently absent. The CRISPRi approach presented by the authors certainly exhibits less bias towards short genes, but this reviewer believes they are stretching their claims a little too far in this context.

Reviewer 1-3

- Their criteria for selecting negative control sgRNAs would be made better if they biased their selection for the seed sequence. 18nt sgRNAs are enough to allow for dCas9 binding, and the criteria of only selecting <5 nt mismatches out of 23 allows for such an sgRNA if they all occur on the 3' end. The authors should consider biasing their screen of control sgRNAs by including this information going forward (i.e. mismatches in the 1st nt next to the PAM sequence is more important than the 2nd, and so forth).

Reviewer 1-7

- This is an intriguing find that these genes are actually not essential. The authors should clarify their methods here a bit more; which specific strain these mutants were made in (I'm assuming WT MG1655). And were these cells able to grow in both solid agar and liquid culture? Or did all growth occur only on plates including the colonies that were sequenced. Regardless, this additional information strengthens the paper significantly.

Reviewer 3-2

- This addition of lines 636-639 should be removed. Claims of unpublished data in text present more questions than answers, while adding little to the reader's understanding.

Reviewer 3-3

- Change "5" to "five"

Reviewer 3-4

- Change "5" to "five"

Reviewer 3-5

- I agree that the minor differences between MG1655 and BW25113 are probably negligible for creating the differences observed here. But the authors could strengthen their manuscript by further discussing the 22 genes identified as essential. The authors don't really answer this

question raised by this reviewer. Given that they claim their approach can be better than knockout approaches for determining essential genes, especially for shorter genes, this could be addressed by asking questions about these 22 genes. Are they shorter than the other essential genes, and therefore less likely to be identified by gene knockouts? Are they duplicated genes in the genome?

Reviewer #3 (Remarks to the Author):

Revisions of NCOMMS-17-00870A-Z
Previously NCOMMS-17-32460

Summary

The revised manuscript has addressed many of our concerns, and only a few questions remain.

Major concerns not addressed in this revision

- (#3) Although we appreciate the more thorough explanation of the fitness analysis and depth/growth limitations now in the manuscript, there are some remaining concerns with this analysis.
 - Agree that the analysis presented here (not growth over time (γ) but comparison of selection vs. non-selected growth) has some precedent in the RNAi field (Kampmann et al. 2013, PNAS). The “differential growth metric” δ in Kampmann (see Methods) is the same as the one presented here, except that it includes a scaling factor for the number of doublings. Is there a specific reason why this scaling factor is not included in the presented analysis? It seems appropriate that the measured changes in abundance be scaled to the number of generations. If the essentiality experiment described here had been carried out for different number of generations, how would the calculated fitness of each gene (and therefore designation of essentiality) have changed over time?
 - (Minor) Do the authors note or anticipate any cases where knockdown of an essential gene may lead to cell lysis not just cessation of growth? Might there be biological reasons why more sequencing depth won't help recovery of those strains?
- (#5 & Reviewer 1 #7) Claims about essentiality based on a lack of CRISPRi phenotype.
 - The manuscript now contains a Cas9-facilitated recombineering approach to test gene essentiality by introducing frameshift mutations early in the ORF. Orthogonal approaches to CRISPRi are important (as the manuscript now notes) for definitively assigning essentiality, but this particular assay should include a report of efficiency in order to be useful for the reader. The experiment shows that 3 out of 5 genes could be disrupted using this approach, while 2 could not. Was the efficiency of disruption comparable to that of a known non-essential gene? If it is significantly lower, the resulting strain is more likely to be an extragenic suppressor than a bona fide “knockout”.
 - If the authors wish to settle the record on the essentiality of these 22 genes, a complementation and deletion/disruption experiment would be necessary. If it is not in their interests to do this (as it would surely be a lot of work!) they should refrain from strong statements on the reliability of their approach. An appropriate framework (and more useful to the reader) for determining gene essentiality would be that the Keio collection has one set of errors, and this CRISPRi approach has a non-overlapping set of errors. The authors' argument that CRISPRi is faster and less time/resources intensive is a good one, so there is no need to overstate their case.
- Recently a new Tn-seq dataset for E. coli (in the same background as the Keio collection, BW25113) has been published including a thorough analysis of gene essentiality (Goodall doi: 10.1128/mBio.02096-17). This is a more appropriate reference set for comparing the efficacy of CRISPRi to that of transposon mutagenesis, as it has higher density of insertions, etc. This manuscript would benefit from including this newer dataset in the analysis, at least in Figures 2 and 3.

Minor concerns not addressed

- Additional references for gene essentiality in *E. coli*
 - Kato & Hashimoto 2007 (doi:10.1038/msb4100174) for essentiality in MG1655 (supplementary table II)

Major concerns addressed in this revision

- (#1) The addition of single-cell data on the efficacy of the CRISPRi system is very helpful.
 - How were these sgRNAs chosen? Could not find the associated figure...
- (#2) Comparison of knockdowns to deletions
 - Data not shown, but related to hits from industrial chemical resistance screen. This is acceptable.
- (#4) Appreciate the response to our concern about “false negatives” in the screen for auxotrophic genes. The supplementary note and reference to BioRxiv doi: 10.1101/129668 is likely sufficient.
- (#6) We applaud this new analysis on reverse polarity, and how it is discussed in the manuscript.
- (#7) Okay.
- (#8) New biology and discussion about screen for resistance to industrial chemicals is sufficient.

Response to the Reviewers:

Reviewer #1 (Remarks to the Author):

The authors have satisfied this reviewers comments.

[Editorial note: The reviewer provided additional comments to pass along to the authors]

Reviewer 1-1

The insight that the best guide RNAs fall within the 5% of ORF is unsurprising based on the wealth of work reported. The relevance with respect to this screening method needs to be explained. Also, explain why multiple guide RNAs (10-15) are needed for targeting a single gene. Also, what are the actual rules the authors find from Fig 1b for designing their sgRNAs? Is it targeting the first 5% of the ORF, which doesn't seem particularly connected to the actual figure? Is it the number of sgRNAs required to call out successful hits (10)? Should be made more clear.

• The “Position” versus “Random” sgRNAs could be explained better. I'm assuming “Position” means in the first 5% of the gene, while “Random” means anywhere throughout the gene.

Response: we thank the reviewer for this suggestion. “Position” here means that in each cycle we include the X sgRNAs most proximal to the start codon, giving rise to one sgRNA subset ($X = 3, 5, 10, 15, 20, 30$, see Figure 1b). “Random” definitely means that randomly selecting X sgRNAs throughout the gene in each cycle. We modified the manuscript accordingly (Line 172-176).

“Considering the position-dependent sgRNA activity observed above, for each gene, we included X sgRNAs most proximal to the start codon, giving rise to 6 sgRNA subsets ($X = 3, 5, 10, 15, 20, 30$). We then recalculated the fitness for each gene based on the sampled sgRNA set. This sampling strategy is termed as ‘position’, in contrast to ‘random’ method that selecting sgRNAs randomly throughout the gene.”

• “Amino acid addition” is not a phenotype, but the media they tested in.

Response: we thank the reviewer for this suggestion. “Amino acid addition” is revised to “L-Trp biosynthesis” phenotype in the table of Figure 1c.

• change “loginitial” to “log initial”. What they mean by that is also confusing.

Response: we thank the reviewer for this suggestion. “loginitial” is revised to “initial” phenotype in the table of Figure 1c and Supplementary Fig. 7. In these experiments (furfural and isobutanol tolerance), “initial” simply means that we use the seed culture rather than a culture at control condition (like all other experiments described here) as control condition

for data processing (equation II). This is explained in Supplementary Fig. 7.

- **The explanation for choosing 15 sgRNAs is fine and makes sense.**

Reviewer 1-2

The distribution of the number of guide RNAs across all the genes in the genome is not clear? Do the different numbers of guide RNAs per gene impact the experiments or introduce any bias?

- **I agree with their explanation largely. The only problem I have is that this suggests an inherent bias against identifying shorter genes related to a particular phenotype as the authors indicate. While I agree for certain phenotypes such as essentiality this bias will be miniscule, it may pose a greater problem for more difficult to parse phenotypes such as biofuel tolerance. I don't think this detracts from the novelty of this work, but I think the claim in lines 74-76 regarding no bias towards longer genes is misleading. The authors should change "without bias" to "minimal bias" or something along those lines. The authors should also briefly touch upon this limitation in the paragraph mentioned (lines 525-536), as it is currently absent. The CRISPRi approach presented by the authors certainly exhibits less bias towards short genes, but this reviewer believes they are stretching their claims a little too far in this context.**

Response: we thank the reviewer for this suggestion. Indeed, sgRNA number is always an important issue to cause bias. To clarify this, we revised the manuscript as below (Line 72-75, Line 559-565).

"Finally, compared with Tn-seq, the sgRNA library can be designed uniformly across the bacterial chromosome with minimal bias towards longer genes, thus maintaining identical statistical robustness for genes with short coding regions."

"The potential limitations of this method (which are indeed shared with Tn-seq to a certain extent) include, first, possible false positives occurring for genes transcribed as polycistronic mRNAs because of the polar effect of CRISPRi and, second, the failure to design enough sgRNAs for some genes with short coding length or microbial genomes with an extreme GC content because of the requirement of the protospacer-adjacent motif (PAM) of dCas9. The second problem is especially of concern to cause potential bias when assigning moderate phenotypes such as tolerance to shorter genes, where only several sgRNAs are available."

Reviewer 1-3

- **Their criteria for selecting negative control sgRNAs would be made better if they biased their selection for the seed sequence. 18nt sgRNAs are enough to allow for dCas9 binding, and the criteria of only selecting <5 nt mismatches out of 23 allows for such an sgRNA if they all occur on the 3' end. The authors should consider biasing their screen of control sgRNAs by including this information going forward (i.e. mismatches in the 1st nt next to the PAM sequence is more important than the 2nd, and so forth).**

Response: we thank the reviewer for this suggestion. This is indeed a problem in our design, although as we responded previously it cause only

minimal off-target effect (the possibility that all 5 mismatches occur at 3' end is very low). To address this, we revised our software at GitHub to bias negative control sgRNA design towards the seed region as suggested here. We stated this point in the GitHub README file of our integrated software. Meanwhile, we redesigned the negative control sgRNAs using the revised integrated software for several other model bacterial genomes (Supplementary Data 12).

Reviewer 1-7

• **This is an intriguing find that these genes are actually not essential. The authors should clarify their methods here a bit more; which specific strain these mutants were made in (I'm assuming WT MG1655). And were these cells able to grow in both solid agar and liquid culture? Or did all growth occur only on plates including the colonies that were sequenced. Regardless, this additional information strengthens the paper significantly.**

Response: we thank the reviewer for these suggestions. MG1655 strain is used in this experiment. The mutant can be cultivated in liquid culture. We revised the manuscript in the Methods part to clarify this (Line 715-717).

“*E. coli* K12 MG1655 was used in this assay and the obtained mutants were cultivated in liquid LB culture to confirm their growth.”

Reviewer 3-2

• **This addition of lines 636-639 should be removed. Claims of unpublished data in text present more questions than answers, while adding little to the reader's understanding.**

Response: we thank the reviewer for this suggestion. We deleted this statement in the manuscript. Besides, as we responded to Reviewer 3-2, our screening results are consistent with several previously reported gene deletions (or mutations) by our group (*pcnB*) or others (Supplementary Table 6).

Reviewer 3-3

• **Change “5” to “five”**

Reviewer 3-4

• **Change “5” to “five”**

Response: we thank the reviewer for this suggestion. We revised these issue across the manuscript.

Reviewer 3-5 We look forward to more data (Line 320-21) on the 22 genes determined to be essential by the CRISPRi method but not by deletion (although please see update to the Keio above). In particular does complementation with the gene product allow deletion? It may be a good time to note that the genetic backgrounds of the Keio collection and the strain used in this manuscript are not identical.

• I agree that the minor differences between MG1655 and BW25113 are probably negligible for creating the differences observed here. But the authors could strengthen their manuscript by further discussing the 22 genes identified as essential. The authors don't really answer this question raised by this reviewer. Given that they claim their approach can be better than knockout approaches for determining essential genes, especially for shorter genes, this could be addressed by asking questions about these 22 genes. Are they shorter than the other essential genes, and therefore less likely to be identified by gene knockouts? Are they duplicated genes in the genome?

Response: we thank the reviewer for this critical and crucial suggestion. We suggest here that CRISPRi is expected to perform better for determining gene function with short coding length, in contrast to transposon-based method such as Tn-seq. This is not the case when arrayed gene deletion library was used as the reference. Indeed, we compared the length of the 22 potentially false negative essential genes with other Keio essential genes, and found no significant difference (MWU test, two-tailed, $P = 0.142$). And these genes are not duplicated in the genome. We suspect that the difference between our result and arrayed gene knockout like Keio collection is mainly due to the polar effect of marker cassette on neighbor genes during gene knockout. We did not perform complementation assisted gene deletion. We note that we only challenged the essential gene list provided by Keio collection, rather than systematically providing a comprehensively new one. This is beyond the scope of this work. We added these discussions in the manuscript (Line 344-352).

“We firstly checked the number of sgRNAs belonging to these 22 genes and found that most of them have more than 10 sgRNAs (Supplementary Table 4). The distribution exhibits no significant difference compared with that of all protein-coding genes ($P = 0.649$ by two-tailed Mann–Whitney U-test (MWU test)), suggesting no bias derived from sgRNA design is introduced when performing hit gene calling. Similar conclusion can be made for the absence of gene length bias in these potential false positive essential genes ($P = 0.142$ by two-tailed MWU test). None of them are duplicated in the *E. coli* chromosome. Moreover, functional analysis indicates that most of them are not related to paramount biological processes (Supplementary Table 4).”

Interestingly, our results are highly consistent with a very recent preprint paper on BioRxiv (doi: <https://doi.org/10.1101/308916>) developing very similar strategy as our work (posted on April 26). Because they did not provided any supplementary data such as genome-wide gene fitness in rich media on BioRxiv, we failed to perform systematic comparison. They mentioned that there existed totally 53 potentially false negative essential genes identified by CRISPRi. Among 7 potentially false

negative essential genes they mentioned in the main text (*alsK*, *bcsB*, *chpS*, *entD*, *mazE*, *yafN* and *yefM*. Further, *alsK*, *bcsB* and *entD* were confirmed by knockout.), 6 of them are also identified by our work (Supplementary Table S4) except for *yafN*, while this gene is not annotated as essential by Keio collection data. They mentioned it probably due to the difference reference they used (Keio plus EcoGene database).

Reviewer #3 (Remarks to the Author):

Revisions of NCOMMS-17-00870A-Z

Previously NCOMMS-17-32460

Summary

The revised manuscript has addressed many of our concerns, and only a few questions remain.

Major concerns not addressed in this revision

- (#3) Although we appreciate the more thorough explanation of the fitness analysis and depth/growth limitations now in the manuscript, there are some remaining concerns with this analysis.

- Agree that the analysis presented here (not growth over time (γ) but comparison of selection vs. non-selected growth) has some precedent in the RNAi field (Kampmann et al. 2013, PNAS). The “differential growth metric” delta in Kampmann (see Methods) is the same as the one presented here, except that it includes a scaling factor for the number of doublings. Is there a specific reason why this scaling factor is not included in the presented analysis? It seems appropriate that the measured changes in abundance be scaled to the number of generations. If the essentiality experiment described here had been carried out for different number of generations, how would the calculated fitness of each gene (and therefore designation of essentiality) have changed over time?

Response: we thank the reviewer for this suggestion. We agree that using scaling factor (doublings) to normalize the data (growth defect measured by NGS) is very useful for comparison between experiments conducted by different doublings. However, due to the resolution limitation issue we mentioned (doubling vs. NGS depth) for this method, it might be confusing to perform the normalization when NGS depth is the resolution-limiting factor of the method, such as in essential gene identification experiments (Line 845-871). In such scenario, normalized growth defect per generation is expected to be much smaller than its actual value. For example, if such normalization is applied, essential gene knockdown leads to $-6/15 \sim -0.4$, although its true value should be ~ -1 . Even though, we note that, in all other experiments of this work, such treatment is proper because the doublings control the detection resolution

there. To avoid the potential confuse across different experiments caused by this issue, we eliminate the usage of scaling factor in this paper. To clarify this, we add a comment in Methods part to state that when NGS depth is not the limiting issue for method resolution, normalization by generation in experiment is helpful to make the final data comparable across different experiments (Line 863-866).

“We note that when NGS depth is not the limiting issue for method resolution, normalization by generation in experiment, as suggested previously⁶⁷, is helpful to make the final data (gene fitness) comparable across different experiments with varieties of doublings.”

○ **(Minor) Do the authors note or anticipate any cases where knockdown of an essential gene may lead to cell lysis not just cessation of growth? Might there be biological reasons why more sequencing depth won't help recovery of those strains?**

Response: we thank the reviewer for this interesting suggestion. We recognize that the deletions of some genes, for example, antitoxin in TA system, result in cell lysis. Such processes (if genetically encoded, is known as programmed cell death, PCD, Bayles K W. Nat. Rev. Microbiol., 2014) are actually found to be involved in bacterial stress response in a collective manner, such as within biofilm. It is reasonable to anticipate that such processes exist after repressing relevant genes during pooled screening. It may contribute to the release and sharing of metabolite between cells in mixed culture, resulting in false negative (as shown in this paper) for phenotype such as auxotrophy.

However, we do not think it is related to the method resolution due to limited NGS depth. In the experiment setting here, we only recovered the intact cells by centrifugation and wash before plasmid extraction. No matter gene repression leads to growth cessation or cell lysis, the results are all the same that the relevant sgRNA plasmids are not recovered in the NGS library.

Actually, because extreme growth defect phenotype such as complete growth cessation (e.g. caused by essential gene deletion) is rare, in most cases of pooled screening for functional genomics studies, NGS depth is not a problem, where nearly no mutants are completely washed out during the screening. Even for essential gene study by CRISPRi pooled screening, we suggest that strictly inducible dCas9 expression can be used to restrict essential gene screening to 5~6 doublings, which are enough for NGS to dissect essential genes. We admit that our experiment setting for essential gene screening is a drawback in this work, which

causes the confusion when NGS depth limits method resolution. We add comment in the relevant sections to clarify this (Line 866-869).

“We also suggest that strictly inducible CRISPRi system is probably better for essential gene screening than protocol used in this work. By restricting the screening procedure into 5~6 doublings, 100× NGS can reliably dissect essential genes and discriminate from those with only growth impairment.”

● (#5 & Reviewer 1 #7) **Claims about essentiality based on a lack of CRISPRi phenotype.**

○ **The manuscript now contains a Cas9-facilitated recombineering approach to test gene essentiality by introducing frameshift mutations early in the ORF. Orthogonal approaches to CRISPRi are important (as the manuscript now notes) for definitively assigning essentiality, but this particular assay should include a report of efficiency in order to be useful for the reader. The experiment shows that 3 out of 5 genes could be disrupted using this approach, while 2 could not. Was the efficiency of disruption comparable to that of a known non-essential gene? If it is significantly lower, the resulting strain is more likely to be an extragenic suppressor than a bona fide “knockout”.**

Response: we thank the reviewer for these important suggestions. We note that ssDNA based recombination efficiency is generally lower than that of dsDNA template with long homologous arm used in conventional gene disruption. Even though, we usually observed tens of colonies on selective agar plates after overnight cultivation following electroporation. PCR and Sanger sequencing suggested that most of these colonies contained designed frameshift mutations. We also confirmed it by using liquid culture to re-grow the mutants. This information is supplemented in Methods section (Line 712-717).

“Usually tens of colonies were obtained on selective agar plates for the three successfully disrupted genes (*chpS*, *gpsA* and *yhhQ*) after overnight cultivation following electroporation. PCR and Sanger sequencing suggested that most of these colonies contained designed frameshift mutations. *E. coli* K12 MG1655 was used in this assay and the obtained mutants were cultivated in liquid LB culture to confirm their growth.”

○ **If the authors wish to settle the record on the essentiality of these 22 genes, a complementation and deletion/disruption experiment would be necessary. If it is not in their interests to do this (as it would surely be a lot of work!) they should refrain from strong statements on the reliability of their approach. An appropriate framework (and more useful to the reader) for determining gene essentiality would be that the Keio collection has one set of errors, and this CRISPRi approach has a non-overlapping set of errors. The authors’ argument that CRISPRi is faster and less time/resources intensive is a good one, so there is no need to overstate their case.**

Response: we thank the reviewer for these important suggestions. Indeed, resettling the essential gene records in *E. coli* is beyond the scope of this

work. We highly appreciated the suggested framework raised by the reviewer regarding gene essentiality by different methods. To address this issue, we added a new tag in supplementary dataset 11 (essential gene suggested by Keio collection, used as gold standard in this work) reporting all overlap essential genes between Keio collection and our work, as these genes are highly confidential to be essential. Besides, we also adjusted our statement about using CRISPRi to assign gene essentiality (Line 545-553).

“It is worthy noted that as a convenient and high-throughput method, CRISPRi method can either capture a big fraction of known essential genes (Figure 3a, Supplementary Data 11) or suggest the potent false positive (negative) targets in other arrayed gene knockout library for further investigation. Different methods such as CRISPRi, Tn-seq and arrayed knockout probably capture different aspects of biology regarding gene essentiality or other functionalities. In this line, a combinatorial approach of different methods may further increase the reliability of prokaryotic functional genomics screening, akin to that which was demonstrated by a recent paper⁴⁸ in which CRISPR/Cas9 and RNAi screenings were integrated in a human cell line.”

• Recently a new Tn-seq dataset for E. coli (in the same background as the Keio collection, BW25113) has been published including a thorough analysis of gene essentiality (Goodall doi: 10.1128/mBio.02096-17). This is a more appropriate reference set for comparing the efficacy of CRISPRi to that of transposon mutagenesis, as it has higher density of insertions, etc. This manuscript would benefit from including this newer dataset in the analysis, at least in Figures 2 and 3.

Response: we thank the reviewer for this very important suggestion. We carried out performance comparison analysis using the data from this work and similar processing method. This transposon library has an unprecedented density where 901,383 unique transposon insertions were identified, around 16-fold of our sgRNA library. As expected, increased library density strengthened the ability of Tn-seq to identify essential genes compared with that of Wetmore et al dataset (152,018 unique transposon insertions), with a comparable performance of our CRISPRi method reported here (AUC: CRISPRi 0.952 vs. Goodall data 0.950). Tn-seq using such an unprecedented-density library even outperforms CRISPRi in very low false positive rate range (see plot blow, Figure 3b).

In contrast, the trend that CRISPRi works better coping with shorter genes is kept with this new dataset (see below, Figure 3c). When testing on genes with length < 400 bp, the performances of both Tn-seq datasets decrease more rapidly than that of CRISPRi (AUC loss: CRISPRi, 0.033; Tn-seq, Goodall, 0.052; Tn-seq, Wetmore, 0.105).

These results suggest that due to the biased distribution of transposon insertions, much bigger library size than CRISPRi is needed to achieve similar performance for gene-phenotype association, at least in this case. We revised our manuscript to add this new comparison (Line 297-320). We also revised one main conclusion of our paper across the manuscript that CRISPRi outperforms Tn-seq in essential gene identification by making this judgment to more restricted conditions (when gene length is short or library size is similar).

“To compare the performance of our approach with the state-of-the-art method in the microbial high-throughput functional genomics field, Tn-seq, we adopted our method, a dataset from a benchmark Tn-seq study representing the largest application of this method so far²⁵ and a widely used *E. coli* essential gene set constructed by the transposon insertion footprinting method²⁸ as three binary classifiers (see Methods).

During the peer review process of our paper, an elegant work was reported using Tn-seq to dissect essential genes in *E. coli* based on a transposon insertion library with an unprecedented size²⁹ (901,383 unique transposon insertions, around 16-fold of our sgRNA library and 6-fold of the library reported by Wetmore et al.). We also trained the dataset from this work as a binary classifier similarly. Taking advantage of the essential gene set of the Keio collection as the gold standard, we studied the performances of these four classifiers based on the receiver operating characteristic curve (ROC) approach (Figure 3b). The results indicated that the CRISPRi screening achieved performance generally comparable to that of Tn-seq method with a 16-fold larger library size (area under the curve (AUC)-ROC value: CRISPRi, 0.952; Goodall et al, 0.950), despite moderately but significantly poorer performance in the low false positive rate range. The performance of Tn-seq decreased significantly as the decrease of library size (Wetmore et al. AUC-ROC, 0.878), followed by genetic footprinting strategy (AUC-ROC, 0.821). We hypothesize that the performance of CRISPRi should be better coping with shorter genes as compared with the transposon insertion-based methods, as transposon insertion suffers from more severe bias problem. To test this, we recruited 702 protein-coding genes shorter than 400 bp, including 45 Keio essential genes, and retested the method performances (Figure 3c). Indeed, the AUC of CRISPRi was maintained better (0.919) whereas those for Tn-seq decreased to 0.898 (Goodall et al.²⁹) and 0.773 (Wetmore et al.²⁵), supporting our hypothesis.”

Because Goodall et al did not present the raw data of each unique transposon insertion; we were unable to plot the transposon insertion density map across the *E. coli* chromosome, such as that in Figure 2b. However, we note that in their paper, such plot is available as Figure 1A, suggesting the biased density distribution of random transposon insertions. We adopted this plot as below, where the innermost circle (blue) corresponds to the frequency and location of transposon insertion sequences mapped successfully to the BW25113 genome after identification of a transposon sequence.

We also note that very recently (April 26th), a new preprint paper was posted on BioRxiv (doi: <https://doi.org/10.1101/308916>) developing very similar strategy as our work. They constructed a random sgRNA library of roughly 90,000 members and used pooled CRISPRi screening to identify essential or phage-resistance-related genes. Interestingly, our results (essential gene identification) are highly consistent with theirs. Because they did not provided any supplementary data on BioRxiv such as genome-wide gene fitness in rich media, we failed to perform systematic comparison. They mentioned that there existed totally 53 potentially false negative essential genes identified by CRISPRi. Among 7 potentially false negative essential genes they mentioned in the main text (*alsK*, *bcsB*, *chpS*, *entD*, *mazE*, *yafN* and *yefM*. Further, *alsK*, *bcsB* and *entD* were confirmed by knockout.), 6 of them are also identified by our work (Supplementary Table 4) except for *yafN*, while this gene is not annotated as essential by Keio collection data. They mentioned it probably due to the difference reference they used (Keio plus EcoGene database). These results suggested that our screenings are highly reproducible regarding cellular fitness consequences after gene knockdown by CRISPRi (Line 364-377).

“Besides, we also note that during the peer review process of this paper, François Rousset and Lun Cui et al. posted an elegant preprint paper, reporting a very similar strategy as our work. They constructed a random sgRNA library of roughly 90,000 members and used pooled CRISPRi screening to identify essential or phage-resistance-related genes. Interestingly, our results (essential gene identification) are highly consistent with theirs. They mentioned that there existed totally 53 potentially false negative essential genes identified by CRISPRi. Among 7 potentially false

negative essential genes they mentioned in the main text (*alsK*, *bcsB*, *chpS*, *entD*, *mazE*, *yafN* and *yefM*). Further, *alsK*, *bcsB* and *entD* were confirmed by knockout. No supplementary data was provided on BioRxiv, disabling systematic comparison), 6 of them are also identified by our work (Supplementary Table 4) except for *yafN*, while this gene is not annotated as essential by Keio collection data. This is probably due to the difference essential gene reference they used (Keio plus EcoGene database). These results suggested that our screenings are highly reproducible regarding cellular fitness consequences after gene knockdown by CRISPRi.”

Minor concerns not addressed

- **Additional references for gene essentiality in *E. coli***

- **Kato & Hashimoto 2007 (doi:10.1038/msb4100174) for essentiality in MG1655 (supplementary table II)**

Response: we thank the reviewer for this very important suggestion. We referred to this essential gene list constructed in K12 MG1655 strain, and compared the false negative essential gene (Supplementary Table 4) found in our study. Indeed, 13/22 of them are supported by the work of Kato et al. One comment is added in the manuscript (Line 355-357).

“Indeed, 13 of these 22 genes were also reported to be non-essential by an independent work using complementation assisted gene deletion in K12 MG1655 strain³⁴ (Supplementary Table 4).”

Major concerns addressed in this revision

- **(#1) The addition of single-cell data on the efficacy of the CRISPRi system is very helpful.**

- **How were these sgRNAs chosen? Could not find the associated figure...**

Response: we thank the reviewer for this kind reminder. These sgRNAs were actually selected randomly across the coding region of *sfGFP* gene. They are named after the position of the first guanine of PAM in coding region (e.g. sgRNA_36, etc). We clarify this in the Methods section and the figure caption part of Supplementary Fig. 23.

- **(#2) Comparison of knockdowns to deletions**

- **Data not shown, but related to hits from industrial chemical resistance screen. This is acceptable.**

- **(#4) Appreciate the response to our concern about “false negatives” in the screen for auxotrophic genes. The supplementary note and reference to BioRxiv doi: 10.1101/129668 is likely sufficient.**

- **(#6) We applaud this new analysis on reverse polarity, and how it is discussed in the manuscript.**

- **(#7) Okay.**

- (#8) **New biology and discussion about screen for resistance to industrial chemicals is sufficient.**